# Targeting the Holy Triangle of Quorum Sensing, Biofilm Formation, and Antibiotic Resistance in Pathogenic Bacteria

**DOI:** 10.3390/microorganisms10061239

**Published:** 2022-06-16

**Authors:** Ronit Vogt Sionov, Doron Steinberg

**Affiliations:** The Biofilm Research Laboratory, The Institute of Biomedical and Oral Research, The Faculty of Dental Medicine, Hadassah Medical School, The Hebrew University, Jerusalem 9112102, Israel; dorons@ekmd.huji.ac.il

**Keywords:** antibiotic resistance, antibiotic sensitization, biofilm, biofilm inhibitors, efflux pump inhibitors, ESKAPE bacteria, quorum sensing, quorum sensing inhibitors

## Abstract

Chronic and recurrent bacterial infections are frequently associated with the formation of biofilms on biotic or abiotic materials that are composed of mono- or multi-species cultures of bacteria/fungi embedded in an extracellular matrix produced by the microorganisms. Biofilm formation is, among others, regulated by quorum sensing (QS) which is an interbacterial communication system usually composed of two-component systems (TCSs) of secreted autoinducer compounds that activate signal transduction pathways through interaction with their respective receptors. Embedded in the biofilms, the bacteria are protected from environmental stress stimuli, and they often show reduced responses to antibiotics, making it difficult to eradicate the bacterial infection. Besides reduced penetration of antibiotics through the intricate structure of the biofilms, the sessile biofilm-embedded bacteria show reduced metabolic activity making them intrinsically less sensitive to antibiotics. Moreover, they frequently express elevated levels of efflux pumps that extrude antibiotics, thereby reducing their intracellular levels. Some efflux pumps are involved in the secretion of QS compounds and biofilm-related materials, besides being important for removing toxic substances from the bacteria. Some efflux pump inhibitors (EPIs) have been shown to both prevent biofilm formation and sensitize the bacteria to antibiotics, suggesting a relationship between these processes. Additionally, QS inhibitors or quenchers may affect antibiotic susceptibility. Thus, targeting elements that regulate QS and biofilm formation might be a promising approach to combat antibiotic-resistant biofilm-related bacterial infections.

## 1. Introduction

The discovery of compounds with antibacterial activities has paved the way to rescue the lives of patients with serious infectious diseases. However, the rapid development of antibiotic-resistant bacterial strains has often led to treatment failure. Another medical challenge is biofilm-associated bacterial infections that are often difficult to treat due to the reduced antibiotic sensitivity of the sessile biofilm-embedded bacteria together with diminished penetrability of antibiotics through the extracellular matrix composed of extracellular polymeric substances (EPS) and other materials produced by the bacteria [1,2]. Biofilm-forming microorganisms are estimated to cause 65–80% of human infections [3,4]. Biofilms are communities of bacteria that are embedded in a hydrated, predominantly anionic matrix of bacterial exopolymers such as polysaccharides that have trapped other components from the bacteria or the surroundings including proteins, nucleic acids, lipids, teichoic acids, and various other organic molecules [1,2]. The production of EPS functions as an anchorage site for the adherence of additional bacteria. The microbes communicate with each other through quorum sensing (QS), which regulates the metabolic activity of the cells, promotes biofilm formation, and increases virulence [4]. Based on the central role of QS in the regulation of bacterial biofilm and virulence, several strategies have been developed to target this signaling system [4,5,6,7,8,9,10,11,12].

The biofilms can be formed on both biotic surfaces such as connective tissue, mucus, epithelium, endothelium, intestine, cardiac valves, bone marrow, and the skin [3,4,13,14], and abiotic surfaces such as prostheses, implants, stents, and catheters [4,14,15]. Biofilm-associated diseases include otitis media, chronic rhinosinusitis, pharyngitis, laryngitis, pneumonia, bacterial vaginosis, infective endocarditis, mastitis, atherosclerosis, osteomyelitis, meningitis, urinary tract infections, kidney infections, skin infections, and inflammatory bowel diseases [13,14,15]. Biofilms formed on biological tissues are a major etiological cause of chronic and recurrent infections. In addition, biofilm formation by oral cariogenic bacteria is associated with tooth decay and gingivitis [16]. The sessile biofilm-associated bacteria have been shown to be up to 100–1000 times more tolerant to antibiotics in comparison to the same bacteria in the planktonic state [15,17,18,19,20,21]. Thus, targeting biofilms would be a prominent approach to overcoming the antibiotic resistance of biofilm-associated infections.

Common bacteria involved in severe biofilm-associated infections include the pathogens of the “ESKAPE” group (*Enterococcus* spp., *Staphylococcus aureus*, *Staphylococcus epidermidis*, *Klebsiella* spp., *Acinetobacter baumannii*, *Pseudomonas aeruginosa*, *Enterobacter* spp.), which cause a variety of infections such as skin and soft tissue infections of wounds, bacteremia, urinary tract infections, meningitis, and pneumonia [22,23,24,25,26,27,28,29]. The “ESKAPE” acronym is derived from the ability of these pathogens to “escape” from antimicrobial therapy and the defense mechanisms of the immune system. These bacteria are common causes of life-threatening nosocomial infections, especially in cystic fibrosis patients, critically ill, and immunocompromised individuals [25,27]. The bacteria can adhere to both biotic and abiotic surfaces and form biofilms that are difficult to eradicate. In addition, the bacteria frequently develop resistance to existing antibiotics, which urges the development of new therapeutic strategies.

This review deals with various aspects of the interrelationship between antibiotic resistance, QS, and biofilms with a specific emphasis on pathogenic bacteria of the “ESKAPE” group. The first part describes various mechanisms involved in antibiotic resistance. The second part describes quorum sensing and various two-component systems (TCSs) affecting antibiotic resistance. The third part discusses various factors including TCSs that affect biofilm formation and the impact of biofilms on antibiotic resistance. In the last section of the review, some strategies that have been developed to break the vicious communication between quorum sensing, biofilm, and antibiotic resistance are described. Due to the enormous number of publications describing these issues, we have restricted our review to selected examples, and apologize for omitting others. The general concept is emphasized.

## 2. Antibiotic Resistance Mechanisms

Although the introduction of antibiotics into the clinics is indispensable for the medical treatment of severe infections, their frequent uses have led to the spread of antibiotic-resistant bacterial strains that lead to treatment failure. There are multiple mechanisms that are involved in the acquisition of antibiotic resistance. Among them, drug resistance can be caused by: (***i***) the acquisition of various antibiotic-resistant genes via horizontal gene transfer; (***ii***) decreased membrane permeability; (***iii***) increased production of degrading enzymes that cleave and thus inactivate the antibiotics; (***iv***) increased production of antibiotic modification enzymes that inactivate the antibiotics; (***v***) alterations of the target that disable the binding of the antibiotics; (***vi***) overexpression of efflux pumps that lead to rapid extrusion of the drugs with consequent low intracellular drug concentration; (***vii***) expression of regulatory small RNAs (sRNAs); (***viii***) methyltransferases that methylate 16S and 23S rRNA, thus altering the antibiotic binding site with reduced drug affinity; (***ix***) mutations in rRNAs; (***x***) ribosomal protection; (***xi***) changes in the metabolic state of the bacteria; (***xii***) biofilm formation; (***xiii***) elevated nutrient sequestering mechanisms; (***xiv***) induction of antibiotic tolerance; (***xv***) appearance of persister cells [20,25,28,30,31,32,33,34,35,36,37,38] (Table 1).

Some organisms show intrinsic resistance to given antibiotics, while in others the resistance mechanism can be acquired, and even induced by the antibiotic itself resulting in adaptive resistance [39,40]. The adaptive resistance is usually transient and reversed after the removal of the triggering environmental factors. Adaptive resistance is a major mechanism of how persister cells evade antibiotics [41]. Due to the high mechanistic versatility of antibiotic resistance, it is not possible to include all of them in this review, and the readers are referred to comprehensive reviews elsewhere [28,30,31,35,39,42]. We will below describe in brief the major antibiotic resistance mechanisms with selected examples. The involvement of biofilm in antibiotic resistance will be discussed in Section 4.2.

**Table 1 microorganisms-10-01239-t001:** Various antibiotic resistance mechanisms in Gram-positive and Gram-negative bacteria.

Resistance Mechanism	Examples	References
**Reduced** **drug uptake**	- Reduced expression of outer membrane porins (OMPs) in Gram-negative bacteria (e.g., *Escherichia coli*, *Klebsiella pneumoniae*, *Acinetobacter baumannii*) causes resistance to β-lactams, sulbactam, imipenem, panipenem, and ertapenem.	[32,34,43,44,45]
**Antibiotic** **degrading** **enzymes**	- Group 1 β-lactamases or cephalosporinases hydrolyze cephalosporins (e.g., *Escherichia coli*, *Klebsiella pneumoniae*, *Pseudomonas aeruginosa)*. - Group 2 β-lactamases or penicillinases cleave the β-lactam ring of penicillin (e.g., *Escherichia coli*, *Pseudomonas aeruginosa*, *Klebsiella pneumoniae*, *Staphylococcus aureus*). - Group 3 metallo-β-lactamases or carbapenemases hydrolyze carbapenem antibiotics (e.g., *Escherichia coli*, *Klebsiella pneumoniae*, *Pseudomonas aeruginosa*, *Acinetobacter baumannii)*. - *ereA-D* erythromycin esterase genes mediate the enzymatic cleavage of the macrolactone ring of the macrolide antibiotics (e.g., *Enterobacteriaceae*, *Escherichia coli*, *Klebsiella pneumoniae*, *Pseudomonas aeruginosa*, *Salmonella enterica*, MRSA).	[25,28,34,46,47]
**Antibiotic** **modifying** **enzymes**	- The aminoglycoside-modifying enzymes (e.g., phosphotransferases, acetyltransferases, and adenylyltransferases) inactivate gentamicin and other aminoglycoside antibiotics by catalyzing hydroxyl/amino group modifications (e.g., *Escherichia coli*, *Acinetobacter baumannii*, *Klebsiella pneumoniae*, *Pseudomonas aeruginosa*, *Salmonella typhimurium*, *Staphylococcus aureus*). - Chloramphenicol acetyltransferase detoxifies chloramphenicol by adding an acetyl group thereby preventing its binding to ribosomes (e.g., *Vibrio cholerae*, *Pseudomonas aeruginosa*, *Staphylococcus aureus*, *Enterococcus faecium*). - Macrolide phosphotransferases inactivate erythromycin, azithromycin, and other macrolide antibiotics (e.g., *Enterobacter*, *Escherichia coli*, *Klebsiella pneumoniae*, *Pseudomonas aeruginosa*). - Tet(X)-mediated flavin-dependent monooxygenase inactivates tetracyclines including the last-resort antibiotic tigecycline by adding a hydroxyl group to the C-11a position, resulting in an unstable compound that undergoes auto-decomposition (e.g., *Enterobacteriaceae*, *Escherichia coli*, *Acinetobacter baumannii*, *Klebsiella pneumoniae*, *Pseudomonas aeruginosa*).	[25,28,48,49,50]
**Proteases and Peptidases**	- The protease SepA of *Staphylococcus epidermidis* degrades antimicrobial peptides produced by neutrophils. - Various membrane proteases (e.g., FtsH and HtpX) of *Pseudomonas aeruginosa* protect the bacteria from aminoglycoside antibiotics. - D-stereospecific peptidases (e.g., TriF and BogQ) lead to hydrolytic cleavage of the peptide antibiotics polymyxin, vancomycin, and teixobactin (e.g., *Firmicutes*, *Bacillus*, and *Clostridium* species). - The DD-peptidases VanX and VanY catalyze the removal of vancomycin target in peptidoglycans of Gram-positive bacteria (e.g., *Staphylococcus aureus*, *Enterococcus faecium*), resulting in resistance to this antibiotic.	[51,52,53,54]
**Efflux pumps**	- AbcA, a type III ABC transporter in *Staphylococcus aureus*, confers resistance to β-lactams such as methicillin and cefotaxime; the phosphoglycolipid moenomycin; the lipopeptide antibiotic daptomycin. - AbeM in *Acinetobacter baumannii* extrudes fluoroquinolones. - AbeS in *Acinetobacter baumannii* confers resistance to chloramphenicol, ciprofloxacin, and erythromycin. - AcrAB-TolC in *Enterobacter* species confers resistance to β-lactams, fluoroquinolones, tigecycline, chloramphenicol, lincosamides, tetracyclines, fusidic acid, rifampin, and nalidixic acid. - AcrAD-TolC in *Enterobacter* species causes resistance to aminoglycosides, β-lactams, and quinolones. - AdeABC in *Acinetobacter baumannii* extrudes β-lactams, chloramphenicol, fluoroquinolones, tetracycline, tigecycline, macrolides, and aminoglycosides. - AdeFGH in *Acinetobacter baumannii* provides resistance to fluoroquinolones, chloramphenicol, trimethoprim, clindamycin, and to a lesser extent tetracyclines, tigecycline, and sulfamethoxazole. - AdeIJK in *Acinetobacter baumannii* provides resistance to β-lactams, fluoroquinolones, tetracyclines, tigecycline, lincosamides, rifampin, chloramphenicol, cotrimoxazole, novobiocin, and fusidic acid. - AmvA in *Acinetobacter baumannii* extrudes chlorhexidine, benzalkonium chloride, and polyamines. - EmrAB-TolC efflux pump of *Enterobacter* species confers resistance to nalidixic acid, thiolactomycin, nitroxoline, and hydrophobic proton uncouplers (e.g., carbonyl-cyanide m-chlorophenylhydrazone (CCCP)). - EmrD in *Escherichia coli* extrudes benzalkonium chloride and sodium dodecylsulfate. - KexD efflux pump of *Klebsiella pneumoniae* extrudes macrolides and tetracycline. - KpnEF efflux pump of *Klebsiella pneumoniae* provides resistance to several antimicrobial compounds including benzalkonium chloride, colistin, erythromycin, rifampin, tetracycline, chlorhexidine, triclosan, and bile salts. - KpnGH efflux pump of *Klebsiella pneumoniae* confers resistance not only to multiple antibiotics including azithromycin, ciprofloxacin, erythromycin, gentamicin, and chlorhexidine, but also protects the bacteria from oxidative and nitrosactive stress stimuli. - MacAB-TolC efflux pump of *Klebsiella pneumoniae* confers resistance to eravacycline. - MdfA in *Acinetobacter baumannii* extrudes ciprofloxacin and chloramphenicol. - MdtABC-TolC in *Enterobacter* species confers resistance to novobiocin and quinolones. - MdtEF-TolC in *Enterobacter* species causes resistance to erythromycin and bile acids. - Mef/Mel efflux pumps extrude macrolide antibiotics in *Streptococcus pneumoniae*. - MepA in *Staphylococcus aureus* pumps out fluoroquinolones, tetracyclines, and quaternary ammonium compounds (QACs). - MexAB-OprM in *Pseudomonas aeruginosa* is responsible for resistance to carbapenems, fluoroquinolones, and aminoglycosides. It is also involved in invasiveness and virulence. - MexCD-OprJ in *Pseudomonas aeruginosa* is responsible for the extrusion of quinolones, erythromycin, and cephalosporins. - MexEF-OprN in *Pseudomonas aeruginosa* confers resistance to chloramphenicol, fluoroquinolones, tetracyclines, and trimethoprim. - MexHI-OpmD in *Pseudomonas aeruginosa* confers resistance to vanadium, norfloxacin, and acriflavine. - MexXY-OprM in *Pseudomonas aeruginosa* confers resistance to aminoglycosides, fluoroquinolones and cefepime. - MsrA in *Staphylococcus epidermidis* extrudes macrolide antibiotics. - NorA-C in *Staphylococcus aureus* extrude fluoroquinolones (ciprofloxacin and norfloxacin). - OpxAB-TolC in *Escherichia coli* and *Klebsiella pneumoniae* is associated with resistance to olaquindox, chloramphenicol, quinolones, tigecycline, and nitrofurantoin. - PmrA in *Streptococcus pneumoniae* confers resistance to fluoroquinolones. - QacAB in *Staphylococcus aureus* extrudes quaternary ammonium compounds (QACs), biguanidines, and diamidines. - TetA and TetB efflux pumps extrude tetracyclines in several bacterial species. - YejABEF ABC transporter in *Salmonella enterica* serovar Typhimurium confers resistance to protamine, melittin, polymyxin B, and human defensin-1 and 2.	[55,56,57,58,59,60,61,62,63,64,65,66,67,68,69,70,71,72,73,74,75,76,77]
**Reduced** **affinity of** **targets to the** **antibiotics**	- The mobile genetic element staphylococcal chromosomal cassette (*SCCmec*) in MRSA carries the *mecA* and *mecC* genes encoding for the penicillin-binding protein (PBP) variant PBP2a with low affinity for β-lactams. - Mutations in 23S rRNA and ribosomal proteins L4 and L22 in *Streptococcus pneumoniae* confer resistance to macrolide antibiotics. - A mutation in S10 ribosomal in *Klebsiella pneumoniae* confers resistance to tigecycline. - Mutations in gyrase *gyrA* and topoisomerase IV subunit *parC* (e.g., *Escherichia coli*, *Acinetobacter baumannii*) cause resistance to quinolones such as ciprofloxacin, ofloxacin, levofloxacin, and norfloxacin.	[78,79,80,81,82,83,84,85]
**Modification** **of the targets**	- Mobile colistin resistance (*mcr*) (e.g., *mcr-1*) in Gram-negative bacteria (e.g., *Escherichia coli*, *Klebsiella pneumoniae*, *Pseudomonas aeruginosa, Acinetobacter baumannii*) encodes for a phosphoethanolamine transferase that adds phosphoethanolamine to lipid A of LPS, thereby reducing the affinity of polymyxins to LPS. - The ArnT enzyme in Gram-negative bacteria (e.g., *Escherichia coli*, *Pseudomonas aeruginosa*, *Salmonella spp.*) adds 4-amino-4-deoxy-L-arabinose (L-Ara4N) to the phosphate group of lipid A, thus conferring resistance to polymyxin. - *Erm* genes methylate 23S ribosomal RNA, resulting in a decreased drug-binding affinity of macrolide antibiotics (e.g., *Staphylococcus* and *Streptococcus* spp., *Escherichia coli*). The *ermB* gene in *Staphylococcus aureus* product confers cross-resistance to lincosamides and streptogramin B. - *Crf* genes (e.g., *Enterococcus faecalis*, *Staphylococcus aureus*, *Clostridium difficile*) methylate 23S ribosomal RNA and confer resistance to chloramphenicol, clindamycin, linezolid, pleuromutilins, streptogramin A, and macrolide antibiotics. - PagP (e.g., *Escherichia coli*, *Yersinia enterocolitica*) transfers palmitate to lipid A, which contributes to resistance to antimicrobial peptides. - The *vanHAX* operon (e.g., *Staphylococcus aureus*, *Enterococcus faecium*) is responsible for the substitution of D-alanyl-D-lactate for the D-alanyl–D-alanine dipeptide, resulting in a 1000-fold lower affinity for vancomycin.	[79,86,87,88,89,90,91,92,93]
**Target** **protection**	- The quinolone resistance protein QnrA interacts with *Escherichia coli* topoisomerase IV and gyrase, thus conferring resistance to fluoroquinolones. - The mutant QnrB1 in *Escherichia coli* showed a 10-fold higher affinity to gyrase B (GyrB) than gyrase A (GyrA). - In *Escherichia coli*, sub-MICs of ciprofloxacin or nalidixic acid interfered with the interaction between QnrB1 and GyrA, while having no effect on the interaction between QnrB1 and GyrB. - QnrB19 interacts with *Salmonella Typhimurium* DNA gyrase and confers resistance to norfloxacin and ciprofloxacin.	[94,95,96,97]
**Ribosomal** **protection**	- Ribosomal protection proteins of the *tet* family dislodge tetracycline antibiotics from the 30S ribosomal unit, resulting in tetracycline resistance (both Gram-positive and Gram-negative bacteria). - The ABC-F proteins (e.g., MsrE of *Pseudomonas aeruginosa*; VgaA of *Staphylococcus aureus* and *Staphylococcus heamolyticus*; LsaA and OptrA of *Enterococcus faecalis*) confer resistance to ribosomal-acting antibiotics via a ribosomal protection mechanism by interacting with the ribosome and displacing the bound drug.	[98,99,100,101,102,103]
**Biofilm-** **embedded** **bacteria**	- Reduced penetration of antibiotics. - Sequestration of tobramycin and other positive charged antibiotics by the negatively charged polysaccharides of the EPS. - Presence of sessile bacteria with low metabolic activity. - Presence of persister cells exhibiting antibiotic tolerance. - Increased expression of efflux pumps. - Increased horizontal transfer of antibiotic-resistant genes. - Sequestration of nutrients. - Increased mutation frequency. - Evasion of host defense mechanisms.	[1,13,14,20,104,105,106]

### 2.1. Acquisition of Various Antibiotic-Resistant Genes via Horizontal Gene Transfer

Bacteria show high genetic plasticity that enables the individual bacteria to develop different phenotypes in an ever-changing environment and to promote adaptive evolution, thus providing the bacteria with fitness traits and survival advantages [39]. The acquisition of antibiotic resistance-conferring genes can occur through horizontal gene transfer, including plasmids, gene cassettes in integrons, and transposons that can capture and disseminate genetic material across bacterial genomes.

The readiness for horizontal gene transfer of mobile genetic elements has led to the terminology “mobilome” [107]. Transposons are transposable elements capable of moving from one position to another within a given genome and are often associated with the dissemination of antimicrobial resistance determinants [42]. Integrons use site-specific recombination to move resistant genes between defined sites [42]. The mobile genetic elements are often present in multiple copies in different locations in the genome and can be transferred to other bacteria through intercellular mechanisms of genetic exchange such as conjugation, mobilization, bacteriophage-mediated transduction, and uptake of extracellular DNA by transformation [42]. For instance, *Acinetobacter baumannii* shows high genetic plasticity with a prominent ability to acquire plasmids [108], transposons [109], and integrons [110], conferring resistance against most classes of antibiotics.

A classic example of horizontal gene transfer resulting in antibiotic resistance is the plasmid-mediated colistin resistance by mobile colistin resistance (*mcr*) in *Enterobacteriaceae* that limits the clinical application of colistin as a last-line drug against bacterial infection [111,112]. Since its detection, several *mcr* genes have been characterized and they have been found in other bacterial species too [86]. Colistin (polymyxin E) acts by binding to negatively charged lipopolysaccharides (LPS) and phospholipids in the outer membrane of Gram-negative bacteria, resulting in increased permeability of the bacterial membrane, and consequent bacterial death [86]. The *mcr* genes encode for a phosphoethanolamine transferase that adds phosphoethanolamine to lipid A of LPS, thereby reducing the affinity of polymyxins to LPS [86].

Another example of horizontal gene transfer is the mobile genetic element staphylococcal chromosomal cassette (*SCCmec*) that carries the *mecA* and *mecC* genes encoding for the penicillin-binding protein (PBP) variant PBP2a with low affinity for β-lactams, and the site-specific recombinase genes *ccrAB* and *ccrC* that mediate the integration and excision of *SCCmec* into and from the chromosome [78]. After accurate excision and integration mediated by the site-specific recombinase genes *ccrAB* and *ccrC*, *SCCmec* is integrated into the staphylococcal chromosome, thus leading to the acquisition of β-lactam antibiotic resistance [78]. *SCCmec* is rapidly transferred between staphylococcal species and might be integrated several times within the same bacterial genome [113]. Various *SCCmec* elements are the underlying cause of the appearance of methicillin-resistant *Staphylococcus aureus* (MRSA) which poses a challenge to hospital infections [78].

### 2.2. Decreased Membrane Permeability

Since most antibiotics target intracellular processes, they need to penetrate the bacterial membrane. One mechanism of drug resistance can be achieved by preventing drug uptake. In Gram-negative bacteria, the outer membrane serves as a physical and functional barrier where lipid A modifications limit the interaction with drugs with a concomitant reduction in drug permeability [114]. The uptake of antibiotics in the Gram-negative bacteria requires, among others, the outer membrane porins (OMPs) which are transmembrane pore-forming proteins with a β-barrel structure that allows the passive transport of hydrophilic compounds including nutrients [32]. OPMs are also important for maintaining membrane integrity [32].

In *Acinetobacter baumannii*, the porin OmpA_Ab_ was found to be required for the uptake of β-lactams and sulbactam [115]. Reduced expression of some outer membrane proteins in *Acinetobacter baumannii* was associated with imipenem resistance [43,44]. Later studies showed that *oprD* and *carO* were downregulated in imipenem-resistant *Acinetobacter baumannii* in comparison to drug-sensitive species, together with an upregulation of an efflux pump [45]. Paradoxically, a Δ*ompA* mutant of *Acinetobacter baumannii* was more sensitive to several antibiotics including aztreonam, nalidixic acid, chloramphenicol, and trimethoprim than the parental wild-type strain [116,117]. The increased susceptibility of the Δ*ompA* mutant to antibiotics despite its involvement in antibiotic uptake might be explained by the increased outer membrane permeability to hydrophobic molecules when *ompA* is lacking [118], and the presence of other porins that contribute to the uptake of the antibiotics. Recently, the trimeric, porin-like DcaP was found to facilitate the permeation of the β-lactamase inhibitor sulbactam into these bacteria [119]. DcaP shows an abundance of positively charged residues which leads to a preferential transport of negatively charged substrates [119]. Besides the negatively charged β-lactamase inhibitors, this porin transports succinate and phthalates [119]. OmpA is also important for biofilm formation on both abiotic and biotic surfaces [120,121]. Compound 62520 inhibits *ompA* expression and prevents biofilm formation in *Acinetobacter baumannii* [122].

In *Escherichia coli*, the outer membrane expresses the two porins OmpF and OmpC [123]. An *ompF*-defective *Escherichia coli* mutant was resistant to several antibiotics including β-lactams, suggesting that OmpF functions as the main route of outer membrane penetration for many antibiotics [32,124]. Similar antibiotic resistance was observed in porin mutants of *Klebsiella pneumoniae* (Δ*ompK35*) [125], *Serratia marcescens* (Δ*ompF*) [126], *Pseudomonas aeruginosa* (Δ*oprD*) [127], and *Enterobacter aerogenes* (*omp36 Gly112Asp* mutant) [128]. In *Klebsiella pneumoniae*, which has developed resistance to ertapenem, the non-selective porins OmpK36 and OmpK35 were found to be reduced, lost, mutated, or truncated [34,129,130,131]. OmpK35 of *Klebsiella pneumoniae* forms large permeable porins with high permeability toward lipophilic (e.g., benzylpenicillin) and large (e.g., cefepime) antibiotics [125]. OmpF of *Serratia marcescens* is important for the penetration of nitrofurantoin and the β-lactams ampicillin and cefoxitin [126]. The *omp36* mutant of *Enterobacter aerogenes* that has a substitution of Gly112Asp in the conserved eyelet L3 region of the porin, confers resistance to β-lactams [128]. Reduced expression of the porin protein OprD in *Pseudomonas aeruginosa* led to reduced drug influx of panipenem [132]. This channel is used by the bacteria to take up basic amino acids. The addition of basic amino acids such as L-lysine reduced the response to panipenem, suggesting a competition for the OprD channel [132].

The major OprF porin of *Pseudomonas aeruginosa*, a homolog to the OmpA of *Enterobacteriaceae*, appears mainly in the closed state, which might explain the low outer membrane permeability of these bacteria in comparison to other bacteria [133]. OprF anchors the outer membrane to the peptidoglycan layer and allows the diffusion of small polar nutrients including polysaccharides [134]. Of note, the absence of OprF in *Pseudomonas aeruginosa* caused an increase in biofilm formation and production of the Pel exopolysaccharide through upregulation of the second messenger bis-(3′-5′)-cyclic dimeric guanosine monophosphate (c-di-GMP) [135]. These authors proposed that the absence of OprF leads to cell envelope stress that activates the SigX regulon that is involved in regulating c-di-GMP levels, which in turn regulates the *pel* and *psl* gene clusters. The *PA1181* and *adcA* (PA4843) genes of the SigX regulon are involved in the increased c-di-GMP levels [135].

### 2.3. Increased Production of Antibiotic Degrading Enzymes

The classical examples of bacteria-produced enzymes that inactivate antibiotics are β-lactamases that cleave the β-lactam ring of penicillin [25,28] and carbapenemases that result in resistance to imipenem, ceftazidime, and ceftriaxone among others [25,28,34,136]. The β-lactam antibiotic methicillin was developed to resist β-lactam-mediated degradation, but rapidly after its introduction into the clinics, resistance to methicillin emerged in *Staphylococcus aureus* due to the *SCCmec* cassette carrying the *mecA* gene encoding for the low penicillin-binding protein PBP2a [78]. A different strategy to overcome resistance caused by β-lactamases is the use of β-lactamase inhibitors such as clavulanic acid, sulbactam, avibactam, and ETX2514 in combination with the β-lactam antibiotics [137].

Resistance to macrolide antibiotics such as erythromycin can emerge by the enzymatic cleavage of the macrolactone ring by erythromycin esterases encoded by the *ereA-D* genes [46,47]. The detoxification of macrolides adds to other mechanisms of macrolide resistance that include decreased intracellular concentration via the efflux pumps such as Mel and Mef [55], the expression of *ermB* gene product that methylates the peptidyl-transferase center of newly synthesized 23S rRNA conferring cross-resistance to lincosamides and streptogramin B (MLS phenotype) [79], mutations in 23S rRNA and ribosomal proteins L4 and L22 [79,80], ribosomal protection, e.g., by MsrE [98] and macrolide phosphotransferase mediated modification [49].

Enzymes of the Tet(X) family are flavin-dependent monooxygenases that inactivate tetracyclines including the last-resort antibiotic tigecycline by adding a hydroxyl group to the C-11a position, resulting in an unstable compound that undergoes auto-decomposition [50,138,139,140,141]. This has led to their nickname “tetracycline destructases” [142].

### 2.4. Increased Production of Antibiotic Modification Enzymes

The aminoglycoside-modifying enzymes (acetyltransferases, nucleotidyltranferases, and phosphotransferases) inactivate gentamicin and other aminoglycoside antibiotics by catalyzing hydroxyl/amino group modifications to the 2-deoxystreptamine nucleus of the sugar moieties [28,48]. Chloramphenicol acetyltransferase (CAT) detoxifies chloramphenicol by adding an acetyl group thereby preventing its binding to ribosomes [25,28].

### 2.5. Alterations of the Target That Disable the Binding of Antibiotics

Methicillin-resistant *Staphylococcus aureus* (MRSA) has acquired a PBP2 variant, PBP2a expressed on the *SCCmec* cassette [78]. This PBP2 variant shows low affinity to penicillin, thus enabling cell wall synthesis even in the presence of high concentrations of β-lactam drugs including methicillin [78]. Several variants of the *SCCmec* cassette have been observed [143].

Vancomycin, a glycopeptide antibiotic, acts by binding to the terminal D-alanyl–D-alanine dipeptide of peptidoglycan precursors, thereby interfering with bacterial wall synthesis. Acquired resistance to vancomycin is caused by the substitution of D-alanyl-D-lactate for the D-alanyl–D-alanine dipeptide, resulting in a 1000-fold lower affinity for vancomycin [87]. This modification is mediated by genes of the *vanHAX* operon. The *vanHAX* operon is regulated at the transcriptional level by the two-component VanR/VanS regulatory system in response to vancomycin [88]. In this case, vancomycin activates the membrane sensory kinase VanS, which, in turn, phosphorylates the transcription regulator VanR that drives the expression of the *vanHAX* operon [88,144] (Figure 1A). The *van* gene cluster has been found in human pathogens such as *Enterococcus faecalis*, *Enterococcus faecium,* and *Staphylococcus aureus* [88].

The zinc-dependent D, D-carboxypeptidases VanX, and VanY act by hydrolyzing the dipeptide (D-Ala-D-Ala) and pentapeptide (UDP-MurNac-L-Ala-D-Glu-L-Lys-D-Ala-D-Ala), respectively, and confer vancomycin resistance in *Enterococci* by eliminating the substrate D-Ala-D-Ala [54,145,146].

### 2.6. Overexpression of Efflux Pumps

A frequent reason for drug resistance is the elevated expression of various efflux pumps that extrude the drugs, thereby reducing the intracellular concentration of the antibiotics below the required minimum inhibitory concentration (MIC) [28]. Efflux pumps, in general, regulate the intracellular environment by extruding toxic substrates including secondary metabolites, QS molecules, dyes, biocides, bile acids, hormones, host defense molecules, fatty acids, detergents, heavy metals, organic pollutants, and antibiotics [31,77,147,148,149,150]. In addition, some efflux pumps have a role in the colonization and the persistence of bacteria in the host [151]. Efflux pumps may affect virulence and biofilm formation by excreting extracellular matrix proteins and QS molecules that coordinate biofilm formation, and by affecting surface adhesion [151,152,153,154,155,156,157,158].

The efflux pumps can be categorized into different families based on the amino acid sequence identity, the energy source required to drive export, and the substrate specificities. The major efflux pump families include the resistance-nodulation-cell division family (RND), the major facilitator superfamily (MFS), the multidrug and toxic compound extrusion family (MATE), the small multidrug resistance family (SMR), ATP-binding cassette family (ABC), and the proteobacterial antimicrobial compound efflux family (PACE) [77,152,159,160]. The ABC superfamily belongs to the primary active transporters that use ATP hydrolysis as the energy source, while the other efflux family members are secondary active transporters (symporters, antiporters, and uniporters) that use energy from proton and/or sodium gradient [77,161,162]. The RDN superfamily is only found in Gram-negative bacteria, while the other efflux pump families are found in both Gram-negative and Gram-positive bacteria [163]. Efflux pumps are either single-component transporters catalyzing the drug efflux across the inner cytoplasmic membrane, or multiple-component systems composed of an inner membrane transporter, periplasmic adaptor, and an outer membrane channel [163,164]. The three components in the latter type of efflux pumps (usually belonging to the RDN family) function together to promote the efflux across both the inner and outer membrane of Gram-negative bacteria [163]. Examples of RDN efflux pumps are the AcrAB-TolC of *Escherichia coli* and *Salmonella typhimurium*, and MexAB-OprM and MexXY-OprM of *Pseudomonas aeruginosa* [163]. EmrE of *Escherichia coli* and QacC of *Staphylococcus aureus* belong to the SMR family, while NorA and QacA of *Staphylococcus aureus* and PmrA of *Streptococcus pneumoniae* belong to the MFS family [163,165]. PmpM of *Pseudomonas aeruginosa* and MepA of *Staphylococcus aureus* are examples of efflux pumps belonging to the MATE family, and AbcA of *Staphylococcus aureus* and LmrA of *Lactococcus lactis* belong to the ABC superfamily [163,165].

#### 2.6.1. Inducible Efflux Pumps

The activities of many regulators of the efflux pumps are frequently affected by the substrates that will be transported by the regulated efflux pump [166,167,168,169,170]. These regulators usually contain a drug-binding pocket within the ligand-binding domain, and the binding of the drug to these regulators modulates their transcriptional repressor/activator activities [171,172,173,174]. The best-understood example of the regulation of a gene encoding the regulation of a drug exporter is the control of *tetA* expression by the specific repressor protein TetR [172,175]. Tetracycline binds to TetR, resulting in the transcription of the *tetA* efflux pump [172,175]. The *Staphylococcus aureus* multidrug transporter QacA is transcriptionally repressed by QacR, which interacts with similar substrates as QacA including chlorhexidine digluconate, benzalkonium chloride, and cetylpyridinium chloride [176]. Upon exposure to these compounds, QacR is released from the *qacA* promoter, resulting in the upregulation of QacA [176]. In *Escherichia coli*, EmrR is a negative regulator of the gene encoding the macrolide efflux pump *emrAB*, the repression of which is relieved upon binding of substrates such as tetrachlorosalicylanilide to EmrR [177,178]. Mutations in *emrR* in *Salmonella typhi* and *Salmonella enterica* cause an upregulation of *emrAB* with consequently reduced susceptibility to ciprofloxacin and other antibiotics [179,180]. Norfloxacin induces the expression of the *norA* efflux pump in *Staphylococcus aureus* [181]. NorA expression is regulated by the ArlRS QS system [182], NorR [183], NorG [184], and MgrA [185] (Figure 1B). Mupirocin induces the expression of the efflux pumps NorA and MepA, resulting in resistance induction to norfloxacin and chlorhexidine [186].

The macrolide erythromycin induced the expression of the *mefE*-*mel* efflux pumps in *Streptococcus pneumoniae* by specific interactions of the macrolide C-5 saccharide with the ribosome that alleviate transcriptional attenuation of *mefE*-*mel* [61]. Transcriptional attenuation occurs when the secondary structure of the leader sequence of the transcript terminates transcription in a rho-independent manner [61]. Additionally, certain antimicrobial peptides such as LL-37 activate the transcription of *mefE*-*mel*, resulting in the resistance to erythromycin [187].

The MexXY-OprM efflux pump in *Pseudomonas aeruginosa* can be induced by ribosome-targeting antibiotics such as chloramphenicol, tetracycline, macrolides, and aminoglycosides [188,189,190] (Figure 2). Mutations in the *fmt* gene that encodes for methionyl-tRNA formyltransferase, or the *folD* gene, a component of the folate biosynthesis pathway, led to impaired protein synthesis and upregulation of *mexXY* [191]. Additionally, mutations in the ribosomal proteins L1 (encoded by *rplA*) and L25 (encoded by *rplY*) resulted in an upregulation of *mexXY*, further supporting a functional link between *mexXY* transcription and ribosome dysfunction [192,193]. Stalling of ribosomes at the PA5471 leader peptide (PA5471.1) leads to the transcription of PA5471 that upregulates the expression of *mexXY* through releasing the repressive action of MexZ [194,195]. *mexXY* expression is also regulated by MexR [196] and the QS systems ParRS [66] and AmgRS [197]. MexZ is frequently mutated in aminoglycoside-resistant *Pseudomonas aeruginosa* clinical isolates [198,199,200]. Calcium and magnesium ions could antagonize aminoglycoside efflux through MexXY-OprM [65].

Pathogens that survive in the intestine have often developed resistance mechanisms to the hazardous effects of bile acids [201] (Figure 3). One mechanism is the expression of the AcrAB efflux pump in the *Enterobacteriaceae* family including *Escherichia coli*, *Salmonella*, *Shigella*, and *Klebsiella* [202]. Other resistance mechanisms include the production of bile salt hydrolase that deconjugates bile acids and neutralizes its antimicrobial activity [203] and the expression of the signaling protein IreK (PrkC) that maintains cell wall integrity resulting in resistance to bile salts and cell wall active antibiotics such as cephalosporins [204,205]. Bile salts induce the expression of the efflux pumps *emrB* and *qacA* in *Enterococcus faecalis*, resulting in the simultaneous acquisition of resistance to various antibiotics [206]. In *Pseudomonas aeruginosa*, bile salts induced the expression of *mexAB*-*oprM* and some other efflux pumps, resulting in resistance to macrolide antibiotics and polymyxin [207]. Bile salts also activate various QS two-component systems (QS TCSs) resulting in increased bacterial virulence. In *Lactobacillus rhamnosus* GG, bile salts increased the expression of *baeRS*, *phoRP3,* and *vraRS* [208]. In *Escherichia coli*, bile salts led to the upregulation of the *acrAB* efflux pump, the TCSs *basRS,* and *pmrAB*, as well as lipid A modification genes (*arnBCADTEF* and *ugd*), resulting in cross-resistance to polymyxin [209]. The TCSs BcrXRS and LiaFSR were found to contribute to bile salt resistance in *Enterococcus faecium* [210]. The TCS CpxAR conferred bile acid resistance in *Klebsiella pneumoniae* [211].

The metabolite indole that is produced by the degradation of tryptophan by *Escherichia coli* and other gut bacteria was shown to induce the expression of the efflux pumps *acdD* and *mdtABC* in *Escherichia coli* through a mechanism involving the TCSs BaeSR and CpxAR [212] (Figure 4). In this study, the transcriptional induction by CpxAR required BaeSR, while BaeSR could act alone, suggesting that BaeR is the primary regulator, while CpxR enhances the effect of BaeR [212]. The induction of the efflux pump *mdtE* by indole in *Escherichia coli* was mediated by transcriptional regulator GadX [212]. Moreover, indole was shown to act as an intercellular signaling molecule that induces RamA-mediated upregulation of the *acrAB* multidrug efflux pump in *Salmonella enterica*, with the consequent acquisition of drug resistance [213,214]. Indole is excreted from *Escherichia coli* via the AcrEF-TolC efflux pump [215].

#### 2.6.2. Mechanisms Resulting in Constitutive Overexpression of Efflux Pump

Besides being induced by antibiotics and various other toxic compounds for the bacteria, the expression of the efflux pump is regulated by QS (see Section 3), various stress stimuli (e.g., membrane disruption, protein misfolding), changes in metabolic state, and when the bacteria are embedded in a biofilm (see Section 4) [31,64,77,216]. Moreover, additional factors can result in the constitutive overexpression of efflux pumps, including (***i***) mutations in the local repressor gene; (***ii***) mutations in a global regulatory gene; (***iii***) mutations in the promoter region of the efflux gene; (***iv***) insertion elements upstream of the efflux pump gene [164,165]. Due to the multiple regulatory mechanisms, only selected examples will be highlighted here.

In *Klebsiella pneumoniae*, resistance to tigecycline can be caused by mutations in *ramR*, *acrR*, and *rpsJ* [217,218]. RamR represses the transcription of *ramA* [219], which regulates the multidrug efflux pump AcrAB-TolC [220]. Transformation of *ramR* mutant strains of *Klebsiella pneumonia* with the wild-type *ramR* gene restored susceptibility to tigecycline [219]. *ramR* mutations in a *Salmonella enterica* serovar Typhimurium strain led to overexpression of *ramA* and consequent overproduction of the AcrAB efflux pump [221]. A 2-nucleotide deletion in the putative RamR binding site of the *ramA* promoter was found to confer resistance to fluoroquinolones [221].

The AcrAB-TolC efflux pump is also regulated by the stress-response regulators MarA, RarA, SoxS, and RobA [149,222,223,224] (Figure 5). In *tolC* mutant bacteria, the two QS systems for sensing extracytoplasmic stress BaeRS and CpxARP were upregulated along with the upregulation of MarA, SoxS, and RobA [149]. RarA also regulates the expression of the *oqxAB* efflux genes and the porin *ompF* [222].

Insertion sequence (IS) elements that disrupt the function of regulatory proteins can upregulate the expression of *acrAB*, *adeABC*, and *kpgABC* efflux pump genes in *Escherichia coli*, *Acinetobacter baumannii*, and *Klebsiella pneumoniae*, respectively, resulting in tigecycline resistance [225,226,227]. IS1 elements were found to disrupt the function of AcrR, a repressor of *acrAB* in *Escherichia coli* [225]. Fluoroquinonolone resistance in a *Salmonella enterica* serovar Typhimurium strain was found to be due to an activation insertion sequence (IS1 or IS10) integrated upstream of the *acrEF* operon that encodes for the *acrEF* efflux pump [228].

#### 2.6.3. Major Efflux Pumps in *Pseudomonas aeruginosa*

*Pseudomonas aeruginosa* contains a large number of efflux pumps, with four potent RND-type multidrug resistance efflux pumps (Mex) capable of eliminating toxic compounds from the periplasm and cytoplasm. These efflux pumps (MexAB-OprM, MexCD-OprJ, MexEF-OprN, and MexXY-OprM) have overlapping spectra of antibiotic substrates and confer resistance to carbapenems, fluoroquinolones, and/or aminoglycosides [25]. The MexAB and MexCD are located in the inner membrane, while the OprM and OprJ are in the outer membrane [64]. The *mexAB*-*oprM* operon is repressed by MexR [196] and NalD [229], while activated by BrlR [230] and CpxR [231] (Figure 6). The *mexCD*-*oprJ* operon is repressed by NfxB [232]; and the *mexEF*-*oprN* operon is repressed by NfxC, while activated by the MexT transcriptional activator [233]. Mutation in MexR or NalC results in upregulation of *mexAB*-*oprM*, and resistance to aztreonam [234,235,236].

The efflux pump MexHI-OpmD exports the toxic metabolite anthranilate that serves as a precursor of the autoinducer PQS [237]. *Pseudomonas aeruginosa* lacking a functional MexHI-OpmD pump showed impaired growth due to accumulation of the toxic anthranilate [237]. The MexHI-OpmD efflux pump confers resistance to vanadium, norfloxacin, and acriflavine [237]. However, mutants lacking MexHI-OpmD became less sensitive to tetracycline, chloramphenicol, and rifampicin, and resistant to kanamycin and spectinomycin [237]. Extracellular addition of the autoinducer PQS increased the susceptibility of both the *mexI* and *opmD* mutant strains as well as the wild-type strain to these antibiotics [237]. MexHI-OpmD is upregulated by the endogenous 5-methylphenazine-1-carboxylate which is a substrate of this efflux pump and required for normal *Pseudomonas aeruginosa* biofilm morphogenesis [238]. 5-methylphenazine-1-carboxylate is an intermediate metabolite formed during the conversion of phenazine-1-carboxylic acid to the virulence factor pyocyanin (5-N-methyl-1-hydroxyphenazine) [238]. Pyocyanin upregulates *mexHI*-*opmD* through activation of the redox-responding transcription factor SoxR [239].

#### 2.6.4. Major Efflux Pumps in *Enterobacter* spp.

The AcrAB-TolC tripartite multidrug efflux pump of *Enterobacter* species belongs to the RND superfamily and forms a tripartite complex consisting of an inner membrane pump protein (AcrB) and an outer membrane channel protein (TolC) bridged by a periplasmic adaptor protein (AcrA) [77]. It utilizes the proton motive force as an energy source to extrude the various substrates [56]. This efflux pump is essential for bacterial survival, particularly in the presence of toxic agents. Subinhibitory concentrations of ertapenem induced the expression of the regulator of antibiotic resistance A (*rarA*) that upregulates the expression of *acrAB*-*tolC* [40]. The expression of the *acrAB* and *tolC* genes are upregulated by the AraC-type transcriptional activators MarA, RamA, and SoxS [224,240,241]. *acrAB* is also upregulated by the QS regulator SdiA [242], while repressed by the transcriptional regulators AcrR [243] and Rob [244]. Bile salts and fatty acids bind to the C-terminal part of Rob, inducing a conformational alteration that results in the transcriptional activation of *acrAB* [244] (Figure 5).

The multidrug-resistant operon *marRAB* encodes the repressor *marR*, the activator *marA*, and the repressor *marB* which reduces the rate of *marA* transcription [245,246]. The operon is activated by compounds such as salicylate, chloramphenicol, tetracycline, acetaminophen, and sodium benzoate [247,248]. *marA* was found to be upregulated by the TCS QseBC in *Escherichia coli* through directly binding of QseB to the *marA* promoter [249]. MarA causes a decreased production of the *ompF* porin in *Escherichia coli* by activating the transcription of *micF*, an antisense RNA that binds to *ompF* mRNA, preventing its translation [250]. The OqxAB efflux pump was shown to be regulated by the AraC multidrug-resistant regulators RamA and RarA [222,241,251]. The transcription factor SoxR is oxidized in response to oxidative stress stimuli resulting in the activation of SoxS [252,253,254]. The SoxRS response protects the cells against superoxide toxicity [252], among others by inducing *sodA* [255]. SoxA also induced the expression of the *acrAB*-*tolC* efflux pump in *Klebsiella pneumoniae* with concomitant resistance to tetracycline [255].

#### 2.6.5. Major Efflux Pumps in *Staphylococcus aureus* Contributing to the MRSA and MDRSA Phenotypes

More than 30 efflux genes have been characterized in *Staphylococcus aureus* [57]. Among these, NorA-C, MepA, and MdeA pump out fluoroquinolones and quaternary ammonium compounds (QACs), and the efflux pumps SepA and QacA/B extrude QACs and biguanidines such as chlorhexidine [57]. The *norA* gene is overexpressed in around 50% of *Staphylococcus aureus* strains and contributes to antibiotic-resistant strains [256,257].

The multidrug efflux pump AbcA that confers resistance to β-lactam antibiotics, moenomycin, and daptomycin in *Staphylococcus aureus*, is regulated by the transcription factors NorG, Rot, SarA, SarZ, MgrA, and the QS system AgrBDCA [258,259,260] (Figure 1B). In addition, AbcA is involved in the secretion of the phenol-soluble modulins (PSMs) [71,72], which are cytolytic toxins that lyse erythrocytes and neutrophils and play important roles in *Staphylococcus aureus* infections [51,261,262]. AbcA also affects cell wall autolysis [260]. Subinhibitory concentrations of ampicillin increased the expression of *abcA* and the surface proteins *clfB*, *isdA*, and *sasG* with a concomitant increase in biofilm formation [263].

### 2.7. Involvement of rRNA Methyltransferase in Antibiotic Resistance

Dimethylation of a specific nucleotide residue in the 23S ribosomal RNA by erythromycin resistance methyltransferase (*erm*) protects bacteria from macrolide antibiotics [89]. The majority of the *erm* genes are induced by the macrolide antibiotics, which is likely due to the reduced fitness caused by the ribosomal modification [89,264]. The Cfr methyltransferases methylate 23S ribosomal RNA, thereby preventing the binding of antibiotics to the peptidyl-transferase center [92]. Crf genes have been shown to confer resistance to chloramphenicol, clindamycin, linezolid, pleuromutilins, streptogramin A, and macrolide antibiotics [92,93].

### 2.8. Involvement of DNA Methyltransferase in Antibiotic Resistance

The DNA methyltransferase VchM was found to be required for the sensitivity of *Vibrio cholerae* to aminoglycosides [265]. VchM is an m^5^C DNA methylase that methylates cytosine at 5′-RCCGGY-3′ motifs. The lack of VchM results in increased expression of *groESL-2* chaperone genes and tolerance to aminoglycosides, likely by capturing aminoglycoside-induced misfolded proteins [265].

### 2.9. Involvement of Ribosomal Protection in Antibiotic Resistance

Ribosomal protection proteins (RPPs) are involved in conferring antibiotic resistance toward ribosome-targeting antibiotics [103,266]. The ribosomal protection proteins TetM, TetO, TetS, TetT, TetQ, TetB, and TetW confer resistance to tetracycline antibiotics by releasing the drugs from the 30S ribosomal subunit or by preventing their binding to the ribosome [103]. These RPPs exhibit GTPase activity, bind ribosomes analogously to elongation factors, and displace ribosomal-bound antibiotics [103]. The GTP hydrolysis depends on the binding of the RPP to the ribosome and occurs only after correct codon-anticodon interaction [103]. After the release of the drug, GTP is hydrolyzed and the Tet RRP dissociates from the ribosome, enabling the ribosome to continue the elongation cycle [103].

The ATP-binding cassette (ABC) proteins of the F-subtype (ABC-F) confer resistance to several antibiotics such as lincosamides, pleuromutilins, streptogramin A, and oxazolidinones that target the ribosome peptidyl-transferase center (PTC) of the 50S large ribosomal subunit, and antibiotics such as macrolides and streptogramin B that target the adjacent nascent peptide exit tunnel (NPET) region of the 50S large ribosomal subunit [100,102,267]. The ABC-F proteins are ATPases that confer antibiotic resistance via ribosomal protection mechanism by interacting with the ribosome and displacing the bound drug, thus alleviating the translational inhibition caused by the antibiotics [99,100,268]. Examples are the LsaA and OptrA of *Enterococcus faecalis*, VgaA of *Staphylococcus aureus*, and VgaL of *Listeria monocytogenes* that confer resistance to PTC-binding antibiotics, and the macrolide and streptogramin B resistance (Msr) proteins such as MsrE of *Pseudomonas aeruginosa*, that confer resistance to NPET-binding antibiotics [101,102,268,269].

### 2.10. Involvement of Non-Coding RNAs in Antibiotic Resistance

Bacterial non-coding RNAs (ncRNAs), although not translated into functional proteins, can regulate sensitivity to antibiotics [33,270] as well as biofilm formation [271,272,273,274,275,276], virulence [277,278,279,280,281], and stress responses [280,282,283,284] by modulating gene expression. Some regulatory ncRNAs (rRNAs) reside in the 5’UTR of the regulated gene and sense the presence of the antibiotics by recruiting translating ribosomes onto short upstream open reading frames embedded in the ncRNA. In the presence of translation-inhibiting antibiotics, ribosomes arrest over the upstream open reading frames, altering the RNA structure of the regulator and thus activating the transcription of the resistance gene [33]. The ciprofloxacin stress-induced ncRNA CsiR was found to regulate in a negative manner ciprofloxacin resistance in *Proteus vulgaris* by targeting the efflux pump *emrB* [285]. CsiR-deficient strains were less sensitive to ciprofloxacin than the wild-type strain [285].

A ribosome-dependent riboregulation is involved in controlling the expression of the *Staphylococcus aureus* macrolide resistance methyltransferase *ermC* gene [286]. In the absence of erythromycin, *ermC* expression is repressed because the ribosome-binding site and AUG start codon of the *ermC* mRNA are sequestered in a stem-loop structure [286]. However, in the presence of erythromycin, ribosomes translating the ErmCL leader peptide become stalled, leading to an alternative stem-loop structure in the *ermC* mRNA that exposes the ribosome-binding site and start codon of the *ermC* gene and thus allows ribosome binding and induction of *ermC* expression [286].

Overexpression of the small RNA SprX in *Staphylococcus aureus* increased the sensitivity of the bacteria to the glycopeptide antibiotics teichoplanin and vancomycin [270]. An *spxR* deletion mutant showed reduced sensitivity to these antibiotics [270]. SprX inhibits the expression of the RNA-binding protein SpoVG [270], which is involved in bacterial resistance to methicillin, oxacillin, and glycopeptide antibiotics, among others, through promoting cell wall synthesis and inhibiting cell wall degradation [287,288]. SpoVG positively regulates the two-component system LysSR resulting in the activation of the antiholin *lrgA* and repression of the murein hydrolase *lytN* [270]. Overexpression of LrgA inhibits murein hydrolase activity and reduces the sensitivity of *Staphylococcus aureus* to penicillin [289] (Figure 7).

### 2.11. Involvement of Bacterial Proteases in Antibiotic Resistance

Bacterial-produced proteases (e.g., SepA) can degrade and thus inactivate antimicrobial peptides [51]. Various membrane proteases in *Pseudomonas aeruginosa* could confer resistance to aminoglycoside antibiotics [52]. Deletion mutation of the *ftsH* gene or insertion inactivation of two FtsH protease accessory factors (HflK and HflC) and the cytoplasmic protease HslUV increased the bacterial sensitivity to tobramycin [52]. Additionally, YccA, a modulator of FtsH, and the membrane protease HtpX conferred resistance to aminoglycosides [52]. The expression of the two latter gene products is regulated by the AmgRS two-component system [52]. The authors proposed that the proteases conferred resistance to aminoglycosides through the elimination of membrane-disruptive mistranslation products [52].

## 3. Quorum Sensing

Quorum sensing (QS) is an intercellular signaling mechanism that allows the communication between bacteria in a cell density-dependent manner [290]. The QS signaling system enables the bacteria to modify their gene expression pattern in response to changes in the environmental conditions, such as nutrient starvation, alterations in temperature, pH and osmolarity, oxidative stress, membrane stresses, antibiotics, and other toxic substances. It provides the bacteria with a selective survival advantage under different harsh conditions. Among others, the QS signaling cascade modulates cellular functions such as metabolic activity, extracellular polymeric substance (EPS) production, nutrient acquisition, transfer of genetic material between the cells, motility, biofilm formation, antibiotic resistance, virulence, and the synthesis of secondary metabolites [4,5,290,291,292,293,294,295,296]. In *Pseudomonas aeruginosa*, QS may also increase the resistance to oxidative stress stimuli by increasing the expression of catalase and superoxide dismutase [297].

The QS system usually involves the secretion of small molecules (autoinducers) that act on surface receptors on adjacent bacteria resulting in the induction of signal transduction pathways regulating biofilm formation, virulence, competence, conjugation, antibiotic resistance, motility, and sporulation [290,298,299,300]. These sensor–regulator pairs are called two-component systems (TCS) and involve phosphotransfer and phosphorelay that activates specific transcriptional regulators [290]. Usually, one of the TCS components encodes the autoinducer, while the other is the receptor that responds to the autoinducer. The sensor receptor is often membrane-bound and consists of an N-terminal sensor domain linked to a C-terminal cytoplasmic histidine kinase that autophosphorylates a conserved histidine residue in its own domain upon receptor activation. The sensor domains can, among others, detect changes in pH, temperature, and osmolarity. To ensure the transfer of the phosphoryl group from the sensor receptor kinase to the response regulator, an intermediate histidine phosphotransfer module (Hpt) is often required [290,301,302]. Phosphorylation of the response regulator leads to conformational changes resulting in altered affinity of the effector domain for its target DNA. As a result, the activation of TCS results in extensive alterations in gene expression. To emphasize the complexity of the bacterial QS system, more than 127 TCS members have been identified in *Pseudomonas aeruginosa* [303,304,305], versus 62 in *Escherichia coli* [306], 70 in *Bacillus subtilis* [307], 17 in *Staphylococcus aureus* [308,309,310,311], 15 in *Enterococcus faecalis* [312], and more than 20 in *Acinetobacter baumannii* [313]. 

Usually, the QS molecules in Gram-positive bacteria are oligopeptides such as autoinducing peptides (AIP), while in Gram-negative bacteria they belong to the family of N-acyl-L-homoserine lactones (AHL) [290,314,315,316,317,318]. The activation of the QS signaling system is usually induced when the autoinducer concentrations reach a threshold level, which occurs at higher cell densities. When activated, the QS leads to the large transcriptional alterations of hundreds of genes in the bacterial genome [290].

The transportation of autoinducers into the extracellular space is essential for their function as ligands for the respective receptors. Some autoinducers such as 3-oxohexanoyl homoserine lactone (3-Oxo-C6-HSL) from *Vibrio fischeri* and N-butyryl homoserine lactone (PAI-2) of *Pseudomonas aeruginosa*, are freely diffusible across the bacterial membranes [319,320]. However, other autoinducers, e.g., N-(3-oxododecanoyl) homoserine lactone (3-Oxo-C12-HSL, PAI-1) and 2-heptyl-3-hydroxy-4-quinolone (PQS) of *Pseudomonas aeruginosa* require a trafficking system [320,321]. One trafficking mechanism is the packaging of the autoinducers into membrane vesicles that are released from the bacteria [321]. The autoinducer PAI-1 of *Pseudomonas aeruginosa* is exported by the MexAB-OprM efflux system [320,322,323], while the autoinducers C4-homoserine lactone (C4-HSL) and 2-heptyl-3-hydroxy-4-quinolone (*Pseudomonas* quinolone signal, PQS), as well as the precursor 4-hydroxy-2-heptylquinoline (HHQ), are extruded by the MexEF-OprN efflux pump [324,325]. The overexpression of these efflux pumps in *Pseudomonas aeruginosa* reduces the intracellular autoinducer levels with concomitant diminished QS signaling [322,324]. Loss of MexAB-OprM caused increased QS responses [323].

### 3.1. TCSs in Vibrio Strains

The QS in various *Vibrio* strains including *Vibrio fischeri*, *Vibrio harveyi*, and *Vibrio cholerae* has been extensively studied [290,326,327,328]. The two major autoinducers in *Vibrio cholerae* are (S)-3-hydroxytridecan-4-one (cholera autoinducer-1; CAI-1) produced by CqsA and 4,5-dihydroxy-2,3-pentanedione (autoinducer-2; AI-2) synthesized by LuxS [327,329,330]. CAI-1 and AI-2 act on the membrane-bound receptors CqsR and LuxPQ, respectively, resulting in the dephosphorylation of LuxO, with concomitant reduced expression of the regulatory small RNAs *qrr1-4*, reduced expression of the AphA regulator, but increased expression of the HapR regulator [331,332,333]. AphA activates genes required for biofilm formation and pathogenicity [334], while HapR prevents biofilm formation by repressing the transcriptional activator *vpsT* [335]. HapR induces the expression of genes encoding proteases that promote *Vibrio cholerae* dispersion [336]. A third QS system of *Vibrio cholerae* is mediated by the autoinducer 3,5-dimethylpyrazin-2-ol (DPO), which activates the transcription factor VqmA, resulting in the expression of the small regulatory RNA *vqmR* [328]. VqmR, in turn, represses genes required for biofilm formation [328]. In *Vibrio harveyi* and *Vibrio fischeri*, the autoinducers activate a signal transduction pathway that leads to the activation of the master regulator LuxR that induces bioluminescence and biofilm formation [337,338,339,340].

Homologs of LuxS have been found in many different Gram-positive and Gram-negative bacterial species, suggesting interbacterial communication [341,342,343,344,345]. Moreover, bacteria can respond to autoinducers produced by other bacteria. For instance, *Escherichia coli* and *Salmonella enterica* serovar Typhimurium detect autoinducers produced by other bacteria through the orphan receptor SdiA which is a homolog to LuxR [346,347]. In addition, the dCACHE-domain proteins PctA/TlpQ in *Pseudomonas aeruginosa* function as a receptor for AI-2 [348]. The AI-2 (R-2-methyl-2,3,3,4-tetrahydroxytetrahydrofuran) produced by LuxS in *Escherichia coli* interacts with its receptor LsrB that regulates the cognate signal kinase LsrK and the activity of the transcriptional repressor LsrR [349,350].

Besides function as an AI-2 synthase, LuxS plays a central role in the metabolic activated methyl cycle (AMC), which is involved in the recycling of S-adenosylmethionine (SAM), a major methyl donor of the cell [341]. Thus, inactivation of *luxS* could result in changes in gene expression due to defective signaling, methionine metabolism, or accumulation of intermediates of S-adenosylmethionine metabolism [341].

### 3.2. TCSs in Pseudomonas aeruginosa

The rapid adaption of *Pseudomonas aeruginosa* to environmental stress conditions has been attributed to the high percentage of the genome encoding TCS elements (64 genes encoding response regulators and 63 histidine kinases) as well as 16 atypical kinases [305]. The major QS systems of *Pseudomonas aeruginosa* rely on the LasI/LasR, RhlI/RhlR, PqsABCDE/PqsR (MvfR), and the AmbBCDE/IqsR TCSs involved in the communal response to extracellular signaling autoinducer molecules, such as N-(3-oxododecanoyl) homoserine lactone (3-Oxo-C12-HSL) encoded by LasI, N-butyryl-L-homoserine lactone (C4-HSL) encoded by RhlI, 2-heptyl-3-hydroxy-4-quinolone (PQS) by PqsABCD and PqsH, and 2-(2-hydroxyphenyl)-thiazole-4-carbaldehyde (IQS) by AmbBCDE [296,351,352]. In the Las system, the transcription factor LasR is activated by the autoinducer 3OH-C12-HSL, and then LasR drives LasI expression and triggers the production of exotoxin A, the LasA protease, and the LasB elastase. In the Rhl TCS, the autoinducer C4-HSL increases RhlI expression by the binding and interaction of C4-HSL with RhlR and induces controlled production of the LasB elastase, rhamnolipids, pyocyanin, and cytotoxic lectins that play crucial roles in virulence and biofilm formation and development. The Las, Rhl, Pqs, and Iqs QS systems are hierarchically connected [296,353,354] (Figure 8). The Rhl and Pqs systems are regulated by the LasR/C12-HSL complex at the transcriptional and posttranscriptional levels [353,355,356]. The genes regulated by LasR and RhlR are involved in biofilm formation, bacterial motility, virulence, and antibiotic resistance. PqsE regulates a range of genes involved in biofilm formation, virulence, and antioxidative processes by serving as a link between the Las and Rhl QS systems [357,358,359,360,361]. Under conditions of low phosphate concentrations, the IQS QS system is activated and promotes biofilm formation through the Rhl and Pqs QS systems [362].

A regulatory network between β-lactam resistance, alginate production, QS, and virulence factors has been shown to exist in *Pseudomonas aeruginosa* [363]. In the presence of β-lactam antibiotics and the β-lactam antibiotic resistance master regulator AmpR, the sigma factor AlgT/U upregulates *ampR* gene expression, while AmpR represses *algT/U* gene expression, generating a negative regulatory feedback loop [363]. In the absence of *ampR*, there is an increase in the transcription of the QS components *lasI* and *lasR*, with a concomitant increase in alginate and pyocyanin production, increased LasA staphylolytic protease activity, and elevated expression levels of the two inducible β-lactamase genes, *ampC* and *poxB* [363,364]. AmpR also modulates antibiotic resistance to other antibiotics by regulating the expression of the MexEF-OprN efflux pump [365].

### 3.3. TCSs in Staphylococcus aureus

Virulence factor production in *Staphylococcus aureus* is largely under the control of the accessory gene regulator (*agr*) QS system (agrBDCA), the SaeRS and ArlRS QS systems, as well as the transcriptional regulators SarA and MgrA [308,366,367,368,369,370] (Figure 9). There are four *agr* groups, all of which exhibit bacterial interference [366,370,371]. Each *agr* type synthesizes a cyclic autoinducing peptide (AIP) with a distinct sequence that activates its cognate AgrC receptor and inhibits the activation of others [366,370]. The four *Staphylococcus aureus* AIPs are seven to nine amino acids long and all contain a thiolactone macrocycle, involving a conserved cysteine sulfhydryl group and R-carboxylate, and an N-terminal tail region [366,370].

The P2 operon in the *agr* locus encodes a polycistronic mRNA termed RNAII which is translated into four Agr proteins (AgrA-D) involved in the autoinduction circuit [370]. AgrD is the precursor of AIP which is proteolytically processed by the AgrB peptidase to generate a thiolactone intermediate. The intermediate is exported and further processed into mature AIP pheromone that activates the membrane-bound AgrC receptor histidine kinase, resulting in the phosphorylation and activation of the response regulator AgrA [366,370]. The phosphorylated AgrA activates the P2 promoter resulting in the upregulation of RNAII, thereby providing a feed-forward QS signaling circuit. In addition, ArgA activates the P3 operon encoding RNAIII, which is a pleiotropic regulator factor. RNAIII functions by base-pairing to the 5’-ends of virulence factor mRNAs, suppressing the synthesis of proteins involved in adhesion, while increasing those involved in invasion [372]. A portion of RNAIII is translated into δ-toxin. RNAIII upregulates the expression of alpha-hemolysin [373] and the global regulator *mgrA* [374], while represses the expression of *coa* (coagulase), *spa* (protein A), and the pleiotropic transcription factor *rot* (repressor of toxin), which is responsible for the repression of toxins [375,376].

AgrCA positively regulates various virulence factors such as hemolysins, proteases, nucleases, phenol-soluble modulins (PSMs), leukocidins, toxins, and capsular polysaccharides, that contribute to the invasive phenotype [308]. In addition, AgrCA upregulates the TCSs *arlRS* and *saePQRS* [308]. The activity of AgrA can be modulated by SarA and SarR [377]. SaeRS regulates the expression of various virulence factors including fibronectin-binding proteins, hemolysins, leukocidins, and coagulase that are important for the pathogenicity of *Staphylococcus aureus* [378,379].

Agr also represses the expression of *apsRS* which confers resistance to antimicrobial peptides such as human β-defensin-3, LL37, and bacteriocins (nisin A, nukacin ISK-1) [380]. ApsR regulates the *dlt* operon that adds alanine to teichoic acid in the cell wall and *mprF* (*fmtC*), which adds lysine to phosphatidylglycerol in cell membranes [381,382,383]. This leads to a reduced negative charge of the bacterial surface, and a consequently reduced binding of the positively charged antimicrobial peptides [381,382,383]. Since Agr expression is low during the early phase of bacterial growth, while high in the stationary phase, the susceptibility to antimicrobial peptides changes during cell growth with low susceptibility during the exponential phase and high susceptibility in the stationary phase [380,384].

### 3.4. Involvement of Two-Component Systems in Promoting Antibiotic Resistance

QS may affect antibiotic resistance by altering the expression of efflux pumps and β-lactamases, modulating the membrane structure, and promoting biofilm formation [31,302,385,386,387] (Table 2 and Section 4.1). Vice versa, efflux pumps may be important for QS and biofilm formation. For instance, the QS-regulated biofilm formation in *Burkholderia pseudomallei* was found to rely on the BpeAB-OprB efflux pump [388]. The autoinducers N-octanoyl-homoserine lactone (C8-HSL) and N-decanoyl-homoserine lactone (C10-HSL) of *Burkholderia pseudomallei* induced the expression of BpeAB-OprB, and BpeAB was required for acyl-homoserine lactone (acyl-HSL) production and optimal production of quorum sensing-controlled virulence factors such as siderophore and phospholipase C [388]. In analogy, reduced biofilm formation was observed in *Escherichia*
*coli* mutants that do not express various genes associated with efflux pumps including *emrD*, *emrE*, *emrK*, *acrD*, *acrE*, and *mftE* [158].

The envelope stress responsive AmgRS TCS of *Pseudomonas aeruginosa* is activated following membrane damage caused by Zn ions and aminoglycosides [197,396]. The aminoglycoside-mediated activation of AmgRS results in the induction of *mexXY* and *mexAB*-*oprM* multidrug efflux operons, which confer aminoglycoside resistance [197,397]. The Zn-induced *mexXY* expression relies on the TCS ParRS [396]. Zinc also activates the TCS ColSR that promotes cell envelope-protective modifications, contributing to Zn tolerance [415].

Gram-negative bacteria survive harmful environmental stressors by modifying their outer membrane. This protection is often achieved by modifying the lipid A moiety of LPS. Various environmental stress stimuli can induce the addition of cationic components, such as 4-amino-4-deoxy-L-arabinose (L-Ara4N) and phosphoethanolamine (pEtN) at the lipid A phosphate groups, which contribute to the stabilization of the outer membrane.

The TCS CprRS of *Pseudomonas aeruginosa* triggers the expression of the LPS modification *arnBCADTEF* operon upon exposure to cationic antimicrobial peptides [418]. The *arnBCADTEF* operon mediates the addition of the positively charged arabinosamine to the negatively charged lipopolysaccharides, thereby preventing the binding of cationic antimicrobial peptides to the lipopolysaccharides [418]. Since the membrane permeabilization activity caused by the antimicrobial peptides relay on their binding to LPS, the arabinosamine modification of LPS leads to drug resistance [418]. The *arn* operon is also induced by low Mg^2+^ concentration detected by the TCSs PhoPQ and PmrAB [452], and by cationic peptides that activate the TCS ParRS [449]. Extracellular DNA present in biofilms activates PhoPQ and PmrAB through chelating metal ions, thus inducing antimicrobial peptide resistance [471]. The activation of PhoPQ and PmrAB is also caused by extracellular DNA-mediated acidification of the biofilms [454]. The activation of PhoPQ and PmrAB resulted in resistance to aminoglycoside antibiotics which is caused by both aminoarabinose modification of lipid A and production of spermidine on the bacterial outer membrane which interacts electrostatically with the negative charges of the O-antigen [454,471]. Both modifications likely reduce the entry of the aminoglycoside into the bacteria [454,471]. DNA-enriched biofilms were 8-fold more tolerant to the antimicrobial peptides polymyxin B and colistin, and 64- to 128-fold more tolerant to the aminoglycosides gentamicin and tobramycin [472].

In *Pseudomonas aeruginosa*, the phosphoethanolamine modification is mediated by a pEtN transferase that is regulated by zinc ions via the ColRS TCS [415]. In *Escherichia coli* and *Salmonella enterica*, the pEtN transferase *eptA* is regulated by the PmrAB TCS [473]. Mutation in PmrAB in *Acinetobacter baumannii* showed a 30-fold increase in the transcription of *pmrC*, which encodes the lipid A phosphoethanolamine transferase [474]. In *Salmonella*, PhoP can activate *pagB*, which encodes a palmitoyltransferase that adds palmitic acid to lipid A [475]. In *Salmonella enterica*, the knockout of PhoPQ made the bacteria more susceptible to antimicrobial peptides by making their surface less rigid and more polarized [476].

The Cpx stress response mediated by the two-component sensor histidine kinase CpxA and the cytoplasmic response regulator CpxR confers resistance to aminoglycoside antibiotics [477], hydroxyurea [477], cationic antimicrobial peptides [420,423], β-lactams [211,403], and chlorhexidine [478], besides being implicated in virulence [479] and biofilm formation [480,481] (Figure 10). The Cpx TCS senses periplasmic or inner membrane protein misfolding and accumulation resulting in the upregulation of the periplasmic protease DegP and degradation of the misfolded proteins [482]. In addition, Cpx induces the expression of the protein folding factors *dsbA*, *ppiA*, *cpxP*, and *spy* [482]. The expression of Cpx-regulated genes was shown to be upregulated upon initial adhesion of *Escherichia coli* to abiotic surfaces, a response requiring the outer membrane lipoprotein NlpE [483]. The Cpx regulon is also upregulated by cell wall-acting antibiotics such as β-lactams [484] and cationic antimicrobial peptides [420] besides its induction by copper ions [485] and alterations in pH and osmolarity [211]. Activation of CpxRA by the aromatic metabolite indole stimulates *mar* transcription [423], which subsequently triggers the multidrug resistance cascade. The CpxRA TCS controls the expression of proteoglycan-modifying enzymes such as the LdtD (YcbB) transpeptidase which catalyzes unusual diaminopimelic acid (DAP)-DAP crosslinks [486] and mediates resistance to β-lactams [487]. In *Salmonella*, CpxRA upregulates the expression of the two N-acetylmuramoyl-L-alanine amidases encoded by *amiA* and *amiC*, resulting in bacterial resistance to protamine and the α-helical peptides magainin 2 and melittin [424]. CpxRA, together with the TCS EnvZ/OmpR, regulates the expression of the porins *ompF* and *ompC* in *Escherichia coli* [211,425].

In *Staphylococcus aureus*, the TCS GraRS was found to regulate resistance to cationic antimicrobial peptides and vancomycin [439]. The membranal receptor GraS senses cationic antimicrobial peptides, resulting in its autophosphorylation on histidine [439]. The GraS then phosphorylates the GraR regulator that upregulates the efflux pump *vraFG* that can pump out cationic antimicrobial peptides as well as vancomycin [439,442]. VraFG can also affect the expression of *mprF*, which attaches lysine to membrane phospholipids and *dltABCD* which adds D-alanine to cell wall teichoic acids, two processes that result in an increase in the net surface positive charges [439,488].

## 4. Biofilms

Biofilm formation is one of the main causes of the persistence of pathogenic bacteria associated with severe infections and outbreaks in hospitals. It is a universal phenomenon among microorganisms and is an important virulence factor that is responsible for the colonization of living tissues or medical devices and causes treatment failure due to decreased susceptibility to antimicrobial drugs and resistance to host defense mechanisms. The biofilms appear as complex three-dimensional structures where the bacteria are embedded in a network of extracellular polymeric matrices (EPS) composed of proteins, polysaccharides, alginates, lipids, teichoic acids, extracellular DNA (eDNA), and other organic compounds secreted by the bacteria or absorbed from the surrounding environment. The EPS can account for over 90% of the biofilm biomass [2,489]. The EPS functions as a barrier and boundary between the microbial community and the external environment, and it plays a central role in bacterial attachment. Additionally, eDNA released from the lysis of a subpopulation of bacterial cells is involved in the attachment, aggregation, and stabilization of the biofilms [490]. eDNA may even facilitate adhesion to highly hydrophobic surfaces [491]. Some autolysins are important for the release of eDNA and the consequent biofilm formation [490,492,493,494,495]. The biofilms have a high water content that allows the flow of nutrients required for bacterial survival within the biofilms [4].

Biofilm formation is characterized by four major stages that act in a cyclic mode [2,8,496,497,498]: (1) an initial reversible attachment to a biotic or abiotic surface, followed by an irreversible binding to the surface; (2) maturation stage where replication of the bacteria forms a microcolony, accompanied by the production of EPS and other components of the extracellular matrix surrounding the microcolony; (3) adhesion of other bacteria to the glycocalyx composed of EPS and other components, thus increasing the complexity and depth of the biofilm that can reach multiple layers of more than 100 μm; (4) release of some of the bacteria from the mature biofilm, leading to the dispersion of the bacteria to other locations thus generating infection with potentially new biofilm formation. During biofilm maturation, canals are formed in the biofilm structure. These allow gradient-based passage of nutrients and signaling molecules, favoring organized agglomeration and differentiation of cells based on their metabolic state [2].

The initial attachment can be active or passive depending on microbial factors such as motility and expression of adhesins and is affected by the kind of surface [499]. The glycocalyx, which is composed of glycoproteins and polysaccharides, contributes to the maturation of the biofilms through electrostatic and hydrogen bonds between the EPS and the surface [499]. Not only the complex polysaccharides but also extracellular DNA contribute to the structure of the matrix and enable cell–cell and cell–surface interactions [500]. The bacteria might also produce factors that induce tissue damage, thereby favoring their subsequent adhesion [501]. The EPS is important for the development of a microenvironment that allows microbial cell–cell interactions and communication, and some of its components may serve as a reservoir of metabolic substances, nutrients, and energy for the biofilm-embedded microbes [2]. Enzymes that can degrade the EPS polymers play an important role in the biofilm life cycle. They provide carbon and energy during starvation and cause biofilm degradation during detachment and dispersal [496,502]. The bacteria in the outer biofilm layers display active metabolism, while those in the deeper layers of the biofilm where hypoxia prevails, show reduced metabolism and have entered a sessile, dormant state [4].

### 4.1. Regulation of Biofilm Formation

Biofilm formation is induced by different factors including changes in the environmental conditions including oxidative stress, alterations in nutritional and metabolic cues, low pH, starvation, heavy metals, host-derived signals, toxic compounds, QS signals, and subinhibitory concentrations of antimicrobials [20,503,504,505,506,507,508,509,510,511,512]. For instance, biofilm formation of *Salmonella typhimurium* is induced by acidic stress and bile salts under oxygen-limiting conditions in the stomach and the small intestine [513]. Biofilm formation is affected by the properties of the biotic or abiotic surfaces [4,499]. Within the biofilms, the microbes coordinate their behavior for promoting growth and producing EPS [489,514].

#### 4.1.1. Induction of Biofilm Formation by Low Antibiotic Concentrations

Various antibiotics at subinhibitory concentrations can induce biofilm formation [263,510,512,515,516,517,518,519,520]. The cell wall antibiotic-induced biofilm formation of *Enterococcus faecalis* was associated with increased cell lysis, increased extracellular DNA levels, and an increased density of bacteria within the biofilm [510]. In *Staphylococcus aureus*, the β-lactam antibiotics induce extracellular DNA release that was dependent on the autolysin Atl [515]. The mupirocin-induced biofilm formation of *Staphylococcus aureus* was dependent on the upregulation of the CidA holin that was associated with the increased production of extracellular DNA [520]. Hoffman et al. [516] observed that aminoglycoside antibiotics could induce biofilm formation of *Pseudomonas aeruginosa* through a mechanism that depends on the aminoglycoside response regulator (*arr*). The *arr* gene encodes for an inner membrane phosphodiesterase whose substrate is cyclic di-guanosine monophosphate (c-di-GMP), a second messenger that regulates cell-surface adhesiveness [516,517] (see Section 4.1.3). Chloramphenicol and erythromycin induce the expression of the capsular exopolysaccharide in *Acinetobacter baumannii* through a mechanism that depends on the TCS BfmRS [518].

Azithromycin, on the other hand, reduces biofilm formation of *Pseudomonas aeruginosa* and inhibits QS-regulated virulence factors such as autoinducer production, pyocyanin production, and swarming [521,522,523]. Azithromycin downregulates the expression of *gacA* [521], which mediates the switch between the motile and biofilm lifestyles of *Pseudomonas aeruginosa* [524]. The involvement of GacA in biofilm formation will be further discussed in Section 4.4.

#### 4.1.2. Involvement of Two-Component Systems in Biofilm Formation

The QS cell-to-cell communication among bacteria facilitates the formation of specialized biofilm structures and causes alterations in the expression of surface molecules such as adhesins that promote adhesion [4,5,301,335,503,507,525,526,527,528,529] (Table 3). The QS-regulated biofilm formation affects both bacterial pathogenesis and resistance to antibiotics [526,527,530,531]. The dependency of biofilm induction on cell density varies between the bacterial species. For instance, in *Vibrio cholerae*, the biofilm-related genes are induced at low cell density, whereas biofilm-related genes in *Pseudomonas aeruginosa* are expressed at high cell density [326,505,532].

LuxS, which is responsible for AI-2 synthesis, is one of the first autoinducer synthetases that was demonstrated to affect biofilm formation [530]. It might either reduce or promote biofilm formation depending on the bacterial species [344,530,533,534,535,536,537] (Table 3). For instance, in *Vibrio* strains AI-2 increases biofilm formation and motility [290,338]. In *Staphylococcus epidermidis*, the *luxS* mutant formed a thicker and more compact biofilm compared to the wild-type strain [533]. LuxS appears to repress biofilm formation in these bacteria through transcriptional regulation of the *ica* gene locus that is responsible for PIA exopolysaccharide production [533]. Similarly, in *Staphylococcus aureus*, LuxS seems to prevent biofilm formation by activating the transcription of *icaR*, a repressor of the *ica* operon [538]. In *Salmonella typhimurium*, AI-2 induces the transcription of the *lsrACDBFGE* operon which, among others, encodes for the *lsr* ATP-binding (ABC) transport system involved in the uptake of AI-2 [539]. LuxS is also responsible for virulence gene expression in *Salmonella typhimurium* [540], and in the absence of *luxS* and AI-2, biofilm formation was impaired in these bacteria [536]. Exogenously added AI-2 increased *Escherichia coli* biofilm formation among others by upregulating adhesin antigen 43 and curli fibers [537,541,542].

**Table 3 microorganisms-10-01239-t003:** Examples of regulatory factors including two-component systems (TCSs) affecting biofilm formation.

Biofilm Formation Regulating Factors	Function	Species	Reference
**Agr**	- Agr is a QS TCS of *Staphylococcus* species that leads to the spread of infection by dispersion of the biofilm and increased production of exoproteins and murein hydrolases at high cell density. - Loss of *agr* activity is associated with enhanced biofilm formation and decreased autolysis rates.	*Staphylococcus aureus*	[370]
**AlgD-A**	- AlgD-A is responsible for alginate synthesis, which is involved in adherence and early biofilm formation.	*Pseudomonas aeruginosa*	[509,543,544]
**AlsSD**	- AlsSD is an α-acetolactate synthase/decarboxylase that is active during staphylococcal biofilm development. - AlsSD prevents cell death by antagonizing CidABC, thus affecting the biofilm biomass. The *cidC*-encoded pyruvate:menaquinone oxidoreductase contributes to cell death through acetate-mediated acidification of the bacterial cytoplasm, while AlsSD involved in acetoin biosynthesis, promotes cell survival by consuming protons, thereby increasing cytoplasmic pH.	*Staphylococcus aureus*	[545,546]
**ArgR**	- Deletion of ArgR reduces biofilm formation of MRSA through the arginine catabolic pathway. - *argR* is upregulated 3.9-fold in a biofilm compared to exponential-phase planktonic cells.	*Staphylococcus aureus*	[547,548]
**ArlRS**	- ArlRS is important for *ica*-dependent biofilm formation. - ArlRS regulates PNAG synthesis by relieving IcaR-mediated repression of the *icaADBC* operon. - An *arl* mutant showed deficient PNAG production and reduced capacity to colonize implanted catheters. - ArlRS represses the expression of *rbf*, a positive regulator of biofilm formation. - ArlRS promotes the expression of *aur*, thereby inhibiting protein-mediated, *ica*-independent, biofilm formation.	*Staphylococcus aureus*	[549,550,551,552]
**AtlA/AtlE**	- AtlA is important for the initial adhesion to a surface. - AtlE is an autolysin gene involved in bacterial lysis resulting in the release of eDNA important for biofilm formation.	*Staphylococcus aureus*, *Staphylococcus epidermidis*	[492,493,553]
**BasSR**	- BasSR is a TCS important for biofilm formation. - BasSR is involved in the expression of biofilm-related genes including the glucan biosynthesis protein *opgC*, the cellulose synthase subunit *bcsA*, the major subunit of curli fibers *csgA*, the basic unit of flagellar filament structural protein *fliC*, the flagellar motor complex unit *motA*, and the type-1 fimbrial subunit *fimA*. - BasSR is induced by Fe and Zn ions.	*Escherichia coli*	[554]
**BfiRS**	- Inactivation of *bfiS* arrests biofilm formation at the irreversible attachment stage and reduces virulence. - Deletion of *bfiS* results in elevated *rsmYZ* levels in biofilm cells. - BfiSR regulates biofilm formation via transcription of *cafA* (RNase G). Inactivation of *cafA* results in increased *rsmZ* levels and arrested biofilm formation.	*Pseudomonas aeruginosa*	[555,556,557]
**BfmRS (RtsAB)**	- BfmS is the sensor kinase that acts on the transcriptional regulator BfmR. - BfmRS increases the expression of capsular exopolysaccharides in *Acinetobacter baumannii*, resulting in enhanced biofilm formation and resistance to killing by host complement. - Sublethal concentrations of the antibiotics chloramphenicol and erythromycin enhance capsular exopolysaccharide production in *Acinetobacter baumannii*, which depends on BfmRS. - BmfRS regulates biofilm maturation of *Pseudomonas aeruginosa*.	*Acinetobacter baumannii*, *Pseudomonas aeruginosa*	[518,558,559,560,561]
**cAMP-CRP**	- cAMP-CRP is stimulated by reduced metabolic energy. - In the absence of cAMP, cAMP receptor protein (CRP) is in an “off” state, which binds DNA nonspecifically and weakly. Upon cAMP binding, CRP undergoes an allosteric transition and is activated to the “on” state, which binds DNA specifically and strongly via its C-terminal domain. - cAMP is produced by adenylate cyclase (CyaA) in *Escherichia coli* and deletion of *cyaA* results in reduced extracellular matrix production and biofilm formation. - CRP positively regulates *csgD* transcription, leading to curli and cellulose production in *Escherichia coli*. - cAMP-CRP in *Klebsiella pneumoniae* is required for fimbria production and biofilm formation, but it inhibits the capsular polysaccharide biosynthesis. - *crp* mutant *Klebsiella pneumoniae* strains could not express MrkA, the major subunit of the type 3 fimbrial shaft.	*Escherichia coli*, *Klebsiella pneumoniae*	[562,563,564,565]
**CidABC**	- CidABC enhances murein hydrolase activity with pro-death functions. - CidABC facilitates DNA release and biofilm formation.	*Staphylococcus aureus*	[495,546,566]
**c-di-GMP**	- High concentrations of c-di-GMP promote alginate and Pel polysaccharide production and biofilm formation in *Pseudomonas aeruginosa*. - A high level of c-di-GMP is associated with reduced OprD expression in the presence of imipenem. - c-di-GMP contributes to biofilm-associated antimicrobial resistance via upregulation of the transcriptional regulator BrlR. - The diguanylate cyclase YdeH is involved in c-di-GMP production.	*Pseudomonas aeruginosa*	[508,517,567,568,569,570,571]
**CodY**	- CodY is a transcriptional repressor that is activated by binding GTP. Depletion of GTP during stringent responses causes de-repression of CodY-regulated genes. - *codY* mutants showed reduced biofilm formation and increased protease activity.	*Staphylococcus aureus*	[572,573,574]
**CqsA**	- CqsA is an autoinducer synthase that reduces biofilm formation through HapR-mediated repression of the *Vibrio* polysaccharide synthesis (*vps*) operon.	*Vibrio harveyi*	[336]
**CpxRA**	- CpxRA responds to extracellular stress stimuli. - CpxA is the membrane sensor kinase/phosphatase and CpxR is the response regulator. - CpxR plays a role in biofilm formation via induction of curli *csgAB* and fimbrial *stdAC*, and repression of *sdiA* transcription.	*Escherichia coli*, *Salmonella**enteritidis*	[575]
**CreBC (BlrAB)**	- CreBC is a TCS activated by the β-lactam cefoxitin that inhibits PBP4, and when PBP4 is mutated. - CreBC promotes biofilm formation, among others through its effector CreD. - The cefoxitin-mediated inhibition of biofilm formation is enhanced in *creD* and *creBC* mutants.	*Pseudomonas aeruginosa*	[426,576]
**CsgD**	- CsgD is a master regulator of biofilm formation that activates the production of curli fimbriae, EPS, and several genes including *adrA* encoding a diguanylate cyclase involved in the synthesis of cyclic di-GMP. - *csgD* translation is negatively controlled by the RprA sRNA in *Escherichia coli*. - CsgD of *Salmonella enterica* serovar Typhimurium is required for the expression of *csgBA*, which encodes curli fimbriae.	*Escherichia coli*, *Salmonella enterica*	[577,578,579,580]
**DltA**	- A *dltA* mutant that leads to a lack of D-alanine modification of the cell wall teichoic acid, shows impaired biofilm formation.	*Staphylococcus aureus*	[581]
**FsrBDC**	- FsrBDC is a QS-dependent regulatory system involved in biofilm formation through the production of gelatinase. Gelatinase is essential for the release of extracellular DNA required for biofilm formation.	*Enterococcus faecalis*	[582,583,584]
**GacSA**	- The histidine kinase GacS and its cognate response regulator GacA regulate the expression of the two small RNAs *rsmY* and *rsmZ* that antagonize RsmA, thereby promoting biofilm formation in *Pseudomonas aeruginosa*. - The GacSA TCS is regulated by RetS, LadS, and PA1611 in *Pseudomonas aeruginosa*. - Disruption of *gacA* caused a 10-fold reduction in biofilm formation by *Pseudomonas aeruginosa*. - GacSA increases biofilm formation in *Acinetobacter baumannii* by regulating the *csu* operon involved in pilus biosynthesis.	*Pseudomonas aeruginosa*, *Acinetobacter baumannii*	[507,559,585,586,587,588]
**GraRS**	- A Δ*graRS* mutant showed reduced staphyloxanthin production, retarded coagulation, weaker hemolysis on blood agar plates, and a decreased biofilm formation. - Expression of the virulence-associated genes *coa*, *hla*, *hlb*, *agrA*, and *mgrA* were downregulated in a Δ*graRS* mutant. - A CA-MRSA with a transposon insertion in *graS* showed increased autolysis and defective biofilm formation. - GraRS activates the *dlt* operon responsible for the D-alanylation of teichoic acid. - GraRS induces the expression of the autolysin gene *atlA*.	*Staphylococcus aureus*	[441,589]
**HapR**	- HapR, a major regulator of QS, represses biofilm formation by regulating the transcription of a series of genes that ultimately causes a reduction in cellular c-di-GMP levels. - HapR represses the expression of the biofilm transcriptional activator *vpsT*. - HapR activates *hapA*, which encodes for a hemagglutinin/protease that promotes detachment of *Vibrio cholerae* from epithelial cells.	*Vibrio cholerae*	[335,504]
**LadS**	- LadS is a calcium-responsive kinase that stimulates GacA activity, thereby promoting biofilm formation by GacA-mediated upregulation of the two small non-coding RNAs *rsmY* and *rsmZ*.	*Pseudomonas aeruginosa*	[507,585,590]
**LasR/LasI**	- LasR/LasI is a TCS system involved in virulence and biofilm formation. - LasI produces the autoinducer 3OC_12_-HSL that binds to LasR activating several genes including the TCS component *rhlR*. - A *lasI* mutant formed flat, undifferentiated biofilms that were sensitive to the biocide sodium dodecyl sulfate. - LasR regulates the expression of more than 300 genes, including the *psl* exopolysaccharide.	*Pseudomonas aeruginosa*	[296,505,509,591]
**LecA**	- LecA is a cytotoxic lectin and adhesin that binds to hydrophobic galactosides and contributes to biofilm formation.	*Pseudomonas aeruginosa*	[592]
**LrgAB**	- LrgAB inhibits murein hydrolase activity and promotes penicillin tolerance. - A *lrgAB* mutant exhibited increased biofilm formation and matrix-associated extracellular DNA.	*Staphylococcus aureus*	[289,593]
**LuxS**	- LuxS is involved in the synthesis of the QS autoinducer-2 (AI-2). - In *Escherichia coli*, AI-2 stimulates biofilm formation and affects biofilm architecture through activating the quorum sensing regulators *mqsR* and *B3022*. B3022 positively regulates the expression of *qseBC*, *flhD*, *fliA*, and *motA*. - In *Staphylococcus aureus*, the LuxS/AI-2 system inhibits PIA-dependent biofilm formation by repressing the expression of *rbf* and activation of *icaR*. - In *Streptococcus pneumoniae*, LuxS is important for early biofilm formation. LuxS induces the expression of the autolysin *lytA* that is implicated in biofilm formation. - LuxS promotes iron-dependent biofilm formation in *Streptococcus pneumoniae*. - A *luxS* mutant of *Klebsiella pneumoniae* was able to form mature biofilms in the intestine but showed reduced capacity in developing microcolonies. - A *luxS* mutant of a *Salmonella* serovar produced thinner biofilms.	*Salmonella* species, *Vibrio* species, *Escherichia coli*, *Staphylococcus aureus*, *Streptococcus pneumoniae*, *Klebsiella pneumoniae*	[525,530,538,541,594,595,596]
**LytA**	- The lack of the autolysin gene *lytA* resulted in impaired biofilm formation.	*Streptococcus pneumoniae*	[597]
**LytSR**	- LytSR senses decreases in membrane potential. - LytSR induces the expression of *lrgAB* that inhibits murein hydrolase activity. - A *lytS* mutant formed a more adherent biofilm.	*Staphylococcus aureus*	[598,599]
**MgrA**	- MgrA acts downstream to ArlRS to control the expression of various virulence genes. - MgrA regulates PNAG synthesis. - MgrA can directly bind to the promoter of the *ica* operon, enhancing the expression of *icaA* and *PIA*. - A *mgrA* mutant was deficient in PNAG production and showed reduced capacity to colonize catheters. - MgrA represses the expression of *rbf,* a positive regulator of biofilm formation. - MgrA promotes the expression of *aur*, thereby inhibiting protein-mediated biofilm formation. - The *mgrA* mRNA is stabilized by RNAIII of the *arg* QS system.	*Staphylococcus aureus*	[374,398,549,550,552,600]
**MifRS**	- MifR is involved in regulating the maturation stage of biofilm formation. - *mifR* deficient mutants fail to form microcolonies.	*Pseudomonas aeruginosa*	[557,601]
**QseBC**	- The QseC sensor kinase—QseB response regulator TCS affects carbon metabolism, flagellar motion, and promotes biofilm formation by upregulating the biofilm-associated genes *bcsA*, *csgA*, *fliC*, *motA*, *wcaF*, and *fimA*. - QseBC confers antibiotic resistance by upregulating the transcription of the efflux pump-associated genes *marA*, *acrA*, *acrB*, *acrD*, *emrD*, and *mdtH*. - The QseCB TCS is regulated by MqsR.	*Escherichia coli*, *Salmonella* Typhimurium	[249,541,602,603,604]
**PA1161**	- PA1161 is a hybrid sensor kinase that promotes biofilm formation by increasing *rmsY* and *rmsZ* expression, and by interacting with RetS, resulting in its inactivation. - PA1161 represses T3SS genes and swarming motility.	*Pseudomonas aeruginosa*	[605]
**PilSR**	- PilSR, which is composed of the PilS sensor and the PilR regulator, controls the expression of *pilA*, the major subunit of type IV pilus. - PilSR regulates the transcription of *fleSR*, and thus positively affects twitching and swimming motilities.	*Pseudomonas aeruginosa*	[606,607]
**(p)ppGpp**	- (p)ppGpp is induced by amino acid starvation. - RelA is involved in the synthesis of ppGpp, while the SpoT hydrolase reduces the ppGpp levels. - ppGpp positively regulates the ribosome modulation factor *rmf* in *Escherichia coli.* - SpoT-mediated reduction of ppGpp levels results in derepression of the poly-GlcNAc biosynthesis machinery PgaA in *Escherichia coli*. - A ppGpp-deficient strain of *Acinetobacter baumannii* showed a significant reduction in *csu* operon expression, which is important for pilus biosynthesis during early biofilm formation. - In *Staphylococcus aureus*, (p)ppGpp can be synthesized by Rel upon amino acid deprivation or by the two alarmone synthetases RelP and RelQ under cell wall stress. RelP and RelQ increase biofilm formation in response to subinhibitory concentrations of vancomycin. - (p)ppGpp contributes to antibiotic tolerance in *Staphylococcus aureus* where it activates ROS-detoxifying systems.	*Escherichia coli*, *Acinetobacter baumannii*, *Staphylococcus aureus*	[570,608,609,610,611,612,613]
**PprAB**	- PprAB regulates the expression of the *cupE1* gene involved in fimbria assembly. - Activation of PprAB triggers a hyper-biofilm phenotype characterized by the expression of BapA adhesin, a type 1 secretion system (T1SS) substrate, CupE CU fimbriae, Flp type IVb pili, and eDNA.	*Pseudomonas aeruginosa*	[614,615]
**Rbf**	- Rbf promotes biofilm formation. - Rbf induces the expression of *sarX* that negatively regulates the expression of *icaR*, thereby activating the *icaADBC* operon. - Rbf represses sarR transcription, resulting in activation of the *icaADBC* operon.	*Staphylococcus aureus*, *Staphylococcus epidermidis*	[616,617,618]
**RcsCDB**	- RcsC is a hybrid sensor kinase that upon activation undergoes autophosphorylation, then transfers the phosphoryl group to the histidine phosphotransferase RcsD, which, in turn, transfers the phosphoryl group to the response regulator RcsB. - The RcsC sensor kinase is required for induction of *pgaABCD* and normal biofilm development in *Escherichia coli*. - RcsC is activated during growth on a solid surface, by zinc ions and by osmotic shock in *Escherichia coli*. - The Rcs phosphorelay is activated by β-lactam antibiotics in *Escherichia coli*, resulting in the survival of the bacteria in the presence of the antibiotics. - In *Salmonella enterica* serovar Typhimurium, phosphorylated RcsB inhibits biofilm development by repressing *csgD*, which is mediated by the accumulation of the small non-coding RNA *rprA*. The RcsCDB phosphorelay is negatively regulated by IgaA, which interacts with RcsD. - RcsB positively regulates the *cupD* gene involved in cell-surface fimbria assembly that is required for biofilm formation in *Pseudomonas aeruginosa*.	*Escherichia coli*, *Salmonella enterica* serovar Typhimurium, *Pseudomonas aeruginosa*	[484,619,620,621,622,623,624]
**RetS**	- The RetS sensor inhibits biofilm formation by repressing GacA activity. - RetS is required for type III secretion system (T3SS) activation and represses exopolysaccharide production and biofilm formation. - A *retS* mutant expresses high levels of c-di-GMP and forms biofilms. - The *retS* mutation led to repression of the type III secretion system, while upregulation of the type VI secretion system was mediated by the diguanylate cyclase WspR.	*Pseudomonas aeruginosa*	[524,625]
**RhlR/RhlI**	- RhlR/RhlI is a TCS system that regulates virulence and biofilm formation. - RhlR binds the autoinducer C4-HSL and an alternative autoinducer to activate genes involved in the synthesis of virulence-associated QS factors and biofilm formation.	*Pseudomonas aeruginosa*	[509,626,627]
**RocS1A1R**	- RocS1A1R is a TCS that consists of two response regulators RocA1 and RocR, and the sensor RocS1. - RocA1 positively regulates the expression of the *cupB* and *cupD* genes involved in the assembly of cell-surface fimbriae required for biofilm formation. - RocR downregulates the expression of *cupB* and *cupD*, resulting in reduced biofilm formation. - RocR is a phosphodiesterase that leads to the hydrolysis of c-di-GMP into 5’pGpG.	*Pseudomonas aeruginosa*	[628,629]
**Rot**	- Rot is a DNA-binding transcriptional regulator that represses the transcription of secreted proteases, thus increasing biofilm mass.	*Staphylococcus aureus*	[630]
**RpoS**	- The expression of the RpoS/σ^S^ sigma subunit of RNA polymerase (RNAP) is induced during entry into the stationary phase and in response to stress conditions. - During rapid growth, the translation of RpoS is inhibited and the RpoS protein is degraded by the ClpXP protease. - RpoS regulates biofilm formation both in a positive and negative manner. - At low temperatures, RpoS contributes to the expression of the exopolysaccharide colanic acid. - An *Escherichia coli rpoS* mutant was defective in the formation of mature biofilms, while another *rpoS* mutant showed increased biofilm formation during the exponential growth phase. - RpoS together with the transcription factor MlrA induce the transcription of *csgD*. CsgD promotes biofilm formation by controlling the expression of curli fibers and the diguanylate cyclase *adrA*, which indirectly activates cellulose production. - *rpoS* transcription is negatively regulated by ArcBA, but positively by (p)ppGpp and cAMP. - The three sRNAs ArcZ, DsrA, and RprA activate the translation of *rpoS* mRNA by enabling ribosome entry to the Shine-Dalgarno sequence. The sRNA/*rpoS* mRNA complexes are stabilized by the RNA chaperone Hfq. - RpoS induces antibiotic tolerance in *Pseudomonas aeruginosa*, especially during the stationary phase.	*Escherichia coli*, *Pseudomonas aeruginosa*	[578,631,632,633,634,635,636,637]
**SadARS**	- SadARS is a three-component system that consists of a histidine kinase and two response regulators. - Nonpolar mutations in any of the *sadARS* genes result in biofilms with an altered mature structure without affecting growth, early biofilm formation, swimming, or twitching motility. - The mutant biofilms show reduced formation of water channels.	*Pseudomonas aeruginosa*	[638]
**SaeRS**	- SaeRS is a TCS that cooperates with Rot to activate the expression of the staphylococcal superantigen-like exoproteins. - A Δ*saeRS* strain exhibited enhanced biofilm formation that was related to increased autolysis and consequent increased extracellular DNA production. The Δ*saeRS* mutant showed increased expression of the autolysins *atlE* and *aae*, without any alterations in *icaA* expression.	*Staphylococcus aureus*	[639,640]
**SagS**	- SagS is a sensor regulator that contains a periplasmic sensory HmsP, and phosphorelay HisKA and Rec domains. - SagS promotes biofilm formation and drug tolerance. - SagS mediates the switch between the planktonic to the biofilm mode of growth. - A Δ*sagS* mutant showed unstructured biofilms and was more sensitive to antimicrobial drugs such as tobramycin. - SagS contributes to biofilm formation via hierarchical phosphotransfer-based signaling to the TCS BfiSR which is required for biofilm formation. - SagS activates the c-di-GMP-responsive transcriptional regulator BrlR, which mediates biofilm tolerance to antimicrobial agents by upregulating the efflux pumps MexAB-OprM, MexEF-OprN, and the ABC transport system PA1874-77.	*Pseudomonas aeruginosa*	[230,555,568,641,642,643,644]
**SarA**	- SarA is a transcription factor that influences virulence, metabolism, biofilm formation, and resistance to some antibiotics. - SarA upregulates the synthesis of fibronectin- and fibrinogen-binding proteins, hemolysins, enterotoxins, toxic shock syndrome toxin 1, oxidative stresses (*sodM* and *trxB*), and genes involved in biofilm formation (e.g., *icaRA* and *bap*), and represses expression of proteases (*ssp* and *aur*), protein A (*spa*), and collagen-binding proteins (*cna*). - The reduced capacity of a *sarA* mutant to form a biofilm involves increased expression of extracellular proteases.	*Staphylococcus aureus*	[645,646,647,648]
**SarX**	- SarX is a SarA homolog that activates *spa* transcription. - SarX promotes biofilm formation by regulating *icaADBC* transcription and PIA production. - *sarX* transcription is upregulated in the stationary phase. - *sarX* is upregulated by MgrA and Rbf.	*Staphylococcus aureus*, *Staphylococcus epidermidis*	[616,649,650,651]
**SdiA**	- SdiA is a QS receptor that detects acyl homoserine lactone (AHL) type autoinducers. - SdiA inhibits biofilm formation and adhesion to epithelial cells. - SdiA activates cell division by activating the *ftsQAZ* operon. - A *sdiA Escherichia coli* mutant formed thicker biofilms compared to the wild-type strain. - The addition of AHL to *Escherichia coli* reduces biofilm formation and represses the expression of *csgD*, *csgA*, and *fimA*. - The two-component systems ArcA, CpxR, OmpR, RcsB, and TorR repress *sdiA* transcription.	*Escherichia coli*, *Salmonella* spp.	[652,653,654]
**SigB**	- SigB is a sigma factor that inhibits the *agr* system, thus increasing biofilm formation. - A *sigB* mutant showed elevated RNAIII levels and increased protease activity.	*Staphylococcus aureus*	[655]
**SrrAB**	- An *srr* mutant showed reduced ability to colonize catheters.	*Staphylococcus aureus*	[549]
**TcaR/IcaR**	- TcaR and IcaR are negative regulators of the *ica* operon, thus preventing *ica*-dependent biofilm formation. - IcaR is the dominant repressor of the *ica* operon, and TcaR enhances the IcaR-mediated repression. - A Δ*tcaRAB* strain shows increased resistance to teicoplanin and methicillin.	*Staphylococcus aureus*	[656,657,658]
**VraSR**	- VraSR is a TCS that responds to cell wall stress. - A Δ*vraSR* mutant showed impaired biofilm formation with a higher ratio of dead cells within the biofilm.	*Staphylococcus epidermidis*	[467]

#### 4.1.3. Role of Cyclic di-GMP (c-di-GMP) in Biofilm Formation

Various bacterial species, especially Gram-negative bacteria such as *Pseudomonas aeruginosa*, *Salmonella enterica*, *Escherichia coli*, and *Vibrio cholerae*, use the secondary metabolite c-di-GMP signaling to regulate biofilm formation [659,660,661,662]. c-di-GMP regulates the transition from planktonic to biofilm state [662]. In general, low intracellular c-di-GMP levels are associated with motility and a planktonic lifestyle, whereas high c-di-GMP levels are associated with biofilm formation and sessility [507,517].

c-di-GMP modulates several aspects of biofilm formation, including flagella rotation, type IV pili retraction, EPS production, surface adhesin expression, resistance to antimicrobial drugs and other stress responses, secondary metabolite production, and biofilm dispersion [517,659] (Figure 11; and Section 4.4). In addition, c-di-GMP affects many other bacterial functions such as cell division, type III secretion, RNA modulation, stress responses, and virulence [663]. Changes in c-di-GMP levels are sensed by c-di-GMP receptor proteins and riboswitch RNAs which propagate the downstream signaling cascade [664,665]. In *Pseudomonas aeruginosa*, c-di-GMP is required for the synthesis of the extracellular polysaccharides Psl and Pel [666], affects the expression of the extracellular matrix adhesion protein CdrA that interacts with Psl and Pel [666,667], and modulates the production of alginate [668].

The c-di-GMP level is controlled by enzymes that synthesize c-di-GMP (e.g., diguanylate cyclases (DGCs) carrying a GGDEF active site motif) and enzymes that degrade the molecule (e.g., c-di-GMP phosphodiesterases (PDEs) carrying either EAL or HD-GP domains) [662]. The *Pseudomonas aeruginosa* genome encodes one of the highest numbers of DGCs and PDEs: 18 GGDEF, 5 EAL, 16 GGDEF/EAL, and 3 HD-GYP predicted proteins [517]. Upon contact of *Pseudomonas aeruginosa* with a surface, the membrane-bound receptor, WspA, becomes activated and triggers c-di-GMP production by the DGC WspR [669]. c-di-GMP, in turn, downregulates the flagellum motility machinery, thereby forcing the bacteria into a sessile growth mode [669].

#### 4.1.4. Role of Non-Coding RNAs (ncRNAs) or Small Regulatory RNA (sRNA) in Regulating Biofilm Formation

The multicellular adhesive McaS sRNA of *Escherichia coli* interacts with mRNAs encoding master transcription regulators of curli and flagella synthesis, resulting in their respective downregulation and upregulation [276]. In *Escherichia coli*, McaS activates the synthesis of the exopolysaccharide β-1,6 N-acetyl-D-glucosamine (PGA) by binding and blocking the global RNA-binding protein CsrA, which negatively regulates *pgaA* translation and positively controls the flagellar master *flhDC* operon expression [276,578]. Ectopic McaS expression leads to the induction of diguanylate cyclases that are repressed by CsrA [276].

The bacterial 3’UTR-derived non-coding RNA RibS was found to affect the biofilm formation of *Salmonella enterica* serovar Typhi [271]. RibS is formed by RNase III-mediated cleavage of the 3’UTR of riboflavin synthase subunit alpha mRNA RibE [271]. Overexpression of RibS promotes biofilm formation by increasing the expression of the cyclopropane fatty acid synthase gene *cfa* [271]. The expression of *cfa* is also regulated by the sRNAs RydC, ArrS, and CpxQ [670]. Both RydC and ArrS mask the RNaseE cleavage site in the *cfa* mRNA 5’-UTR, resulting in the upregulation of *cfa* [670]. CpxQ binds to a different site of the *cfa* mRNA 5’UTR, resulting in the repression of *cfa* [670].

The σ^S^-dependent sRNA SdsR was shown to regulate biofilm formation in *Salmonella* by activating *csgD* and curli expression and downregulating the *ompD* porin [578]. In *Escherichia coli*, SdsR inhibits *flhDC* expression [578].

The small non-coding RNA RsaE affects extracellular matrix composition in *Staphylococcus epidermidis* biofilm [671]. RsaE interacts with the antiholin *lrgA* mRNA, thereby facilitating bacterial lysis and eDNA release [671]. Moreover, RsaE augments PIA-mediated biofilm matrix production [671]. The long non-coding RNA IcaZ prevents *icaR* translation, thereby relieving the repression of *icaADBC* transcription with a consequent increase in PIA production [672].

The small non-coding RsmW RNA in *Pseudomonas aeruginosa* upregulates biofilm formation by binding to the RNA-binding regulator RsmA [673]. RsmA represses the type VI secretion system, exopolysaccharide production, and biofilm formation [674,675]. The involvement of RsmY and RmsZ in regulating RsmA will be discussed in Section 4.4.

### 4.2. Biofilm Formation by Vibrio cholerae

The extracellular matrix of *Vibrio cholerae*, the causative agent of cholera, is composed of *Vibrio* polysaccharides (VPS) and three major matrix proteins, RbmA, RbmC, and Bap1 involved in cell–cell and cell–surface adhesion [676]. Several transcriptional regulators and alternative sigma factors control the biofilm formation in *Vibrio cholerae*. Among them, VpsR and VpsT are considered master regulators of biofilm formation which positively regulate VPS production [677,678]. Both VpsR and VpsT respond to c-di-GMP by inducing the transcriptional switch toward biofilm-regulated genes [678,679,680].

### 4.3. Biofilm Formation by Escherichia coli

The QseBC TCS of *Escherichia coli* was found to promote biofilm formation in dairy cows that suffered from mastitis by upregulating the transcription of the biofilm-associated genes *bcsA*, *csgA*, *fliC*, *motA*, *wcaF*, and *fimA*, and conferred antibiotic resistance by upregulating the transcription of the efflux pump-associated genes *marA*, *acrA*, *acrB*, *acrD*, *emrD*, and *mdtH* [249]. The BasSR TCS promotes biofilm formation in avian pathogenic *Escherichia coli* by upregulating the expression of biofilm- and virulence-related genes, including *ais*, *entC*, *opgC*, *gtcE*, and *fepA* [554]. Moreover, the CpxRA TCS, which senses membrane stress and recognizes misfolded proteins, is required for proper biofilm formation, adherence, motility, and the production of type 1 fimbriae [681].

### 4.4. Biofilm Formation by Pseudomonas aeruginosa

The biofilms of *Pseudomonas aeruginosa* are characterized by exopolysaccharides (EPS) such as Psl, Pel, and alginate intermingled with proteins, rhamnolipids, membrane vesicles, and eDNA [682,683]. The rhamnolipid biosurfactants are required for the initial stage of microcolony formation, the formation of the mushroom-shaped microcolonies, the maintenance of channels between multicellular structures, and contribute to biofilm dispersal [682,683]. The intercalation of the redox active virulence factor pyocyanin with eDNA contributes to the stabilization of the *Pseudomonas aeruginosa* biofilm architecture [684]. Moreover, the flagellum is considered a central component of the biofilm formation of *Pseudomonas aeruginosa* since it provides the mobility needed to actively approach a surface [301]. The type IV pili play a role in surface attachment and colonization [301]. The Cup fimbriae were also found to contribute to the attachment to both abiotic and biotic surfaces, besides providing protection against the recognition by the immune system [301].

The production of the virulence factors rhamnolipids and pyocyanin, and the release of eDNA and membrane vesicles, as well as biofilm development, are controlled by the QS two-component system [296,514]. The QS also elicits a global metabolic rewiring in *Pseudomonas aeruginosa* which is caused by a combination of direct transcriptional alteration of metabolism-associated genes mediated by the QS system and indirectly by global metabolic readjustment as a result of the general and QS-dependent stress responses [685].

In *Pseudomonas aeruginosa*, small cationic polyamines such as putrescine as well as its biosynthetic precursors L-arginine and agmatine ((4-aminobutyl)guanidine), promote biofilm formation and confer resistance to environmental stress stimuli [508,517,686]. Both putrescine and L-arginine increase the intracellular c-di-GMP levels [686]. One of the diguanylate cyclases in *Pseudomonas aeruginosa* is SiaD which is co-transcribed with *siaA*/*siaB*/*siaC* from the *siaABCD* operon. The *siaABCD* encodes a signaling network regulating biofilm and aggregate formation by modulating the enzymatic activity of SiaD [687]. SiaC interacts with SiaD to promote its diguanylate cyclase activity, thus promoting c-di-GMP synthesis [687]. The interaction of SiaC with SiaD is facilitated by dephosphorylation mediated by the inner membrane-associated Ser/Thr SiaA phosphatase, while it is prevented when phosphorylated by the SiaB protein kinase [687]. SiaA is activated by external stress stimuli such as sodium dodecyl sulfate [687].

Biofilm formation in *Pseudomonas aeruginosa* is, among others, regulated by the two QS systems LasI/LasR and RhlR/RhlI that are expressed at both early and late biofilm phases [688,689]. A *lasI* mutant forms flat and undifferentiated biofilms that were sensitive to sodium dodecyl sulfate [505]. *Pseudomonas aeruginosa rhlA* mutants that are deficient in the synthesis of biosurfactants were not capable of forming microcolonies in the initial phase of biofilm formation and were defective in migration-dependent development of mushroom-shaped multicellular structures in the later phase of biofilm formation [682]. The Rhl system regulates the expression of pyocyanin and rhamnolipids that are responsible for the deposition of extracellular DNA which is an important component in the early and late biofilm developmental stages. Additionally, the PqsE thioesterase, which is part of the 2-alkyl-4-quinolone biosynthesis gene cluster pqsABCDE, plays a role in the production of pyocyanin, rhamnolipids, and lectin A [690].

The GacSA TCS in *Pseudomonas aeruginosa* increases the expression of the exopolysaccharides Pel and Psl by upregulating the expression of two small regulatory RNAs, *rsmY* and *rsmZ* [301]. RsmY and RsmZ sequester the translational repressor RsmA thus allowing translation of target mRNAs such as the exopolysaccharides *pel* and *psl*, required for biofilm formation [301,691] (Figure 12A). RsmA increases the expression of the type III secretion system (T3SS) while repressing the expression of the type VI secretion system (T6SS) [625,692]. The activity of GacS is activated by the hybrid histidine kinase LadS [585], while antagonized by the hybrid sensor kinase RetS [524]. The activity of RetS is repressed by the hybrid sensor kinase PA1611, which positively regulates biofilm formation by upregulating *rsmY* and *rsmZ* [605]. The expression level of PA1611 is increased during the transition between acute and chronic infection and mediates the transition to the biofilm state [605]. A *retS* mutant showed elevated levels of c-di-GMP, increased biofilm formation, and a shift in type III and type VI secretion systems that were dependent on c-di-GMP [625]. The *siaABCD* responsible for c-di-GMP production is repressed by RsmA [693], thereby generating a feedback loop (Figure 12A). GacSA also positively regulates the transcription of the QS components *lasR* and *rhlR* in *Pseudomonas aeruginosa* [296]. A Δ*gacA* mutant showed a 10-fold reduction in the ability to form biofilms [588]. The Δ*gacS* strain was hypermotile, produced a reduced amount of acyl-homoserine lactones, showed impaired biofilm maturation, and was more sensitive to certain antibiotics (e.g., tobramycin, ceftriaxone, oxacillin, piperacillin, and rifampicin) in comparison to the parental strain [694].

Other TCSs involved in the biofilm formation of *Pseudomonas aeruginosa* include SagS, BfiRS, BfmRS, and MifRS [301,556,557]. SagS, BfiRS, BfmRS, and MifRS are sequentially phosphorylated during biofilm formation [557] (Figure 12B). Inactivation of either of these components arrests biofilm formation at distinct developmental stages [502,555,556,557]. Biofilms formed by Δ*sagS* and Δ*bfiS* mutants are arrested at the irreversible attachment stage, while biofilms formed by Δ*bfmR* and Δ*mifR* mutants are arrested at the maturation-1 and maturation-2 stages of biofilm development, respectively [502,555,556,557]. SagS enables the switch from the planktonic state to the sessile, biofilm state, through activation of the TCS BfiRS [555,556,641]. BfiRS represses the *rsmYZ* expression levels which seem to be a necessary step for future maturation of *Pseudomonas aeruginosa* biofilm [556]. SagS is also required for the biofilm-associated resistance to antibiotics through BrlR-mediated upregulation of efflux pumps such as *mexAB*-*oprM* and *mexEF*-*oprN* [569,644]. The BrlR expression was dependent on sufficient levels of c-di-GMP [569]. BmfRS regulates biofilm maturation [561], while MifRS modulates microcolony formation [695]. The loss of biofilm biomass in the Δ*bmfR* mutant might be related to increased cell death [696].

The fimbrial *cupB* and *cupC* genes are regulated by the *roc1* locus that encodes one sensor kinase (RocS1) and two response regulators (RocA1 and RocR) [301]. This cluster has also been termed *sadARS* and has been documented to positively regulate biofilm formation while repressing type III secretion genes [638]. Nonpolar mutations in any of the *sadARS* genes result in biofilms with an altered mature structure, but without defects in growth or early biofilm formation, swimming, or twitching motility [638]. Mutations in type III secretion genes resulted in strains with enhanced biofilm formation [638]. Type III secretion genes including *pcrV* are negatively regulated by c-di-GMP [697]. Of note, RocR contains an EAL motif with a phosphodiesterase activity that degrades c-di-GMP [629]. RocA2 was found to repress the expression of the *mexAB*-*oprM* efflux pump [698]. The RocS1A1R system seems to be counterintuitive by providing a signal cascade that promotes biofilm formation via Cup fimbriae and simultaneously increases the sensitivity to antibiotics by repressing a multidrug efflux pump. This could be associated with the fitness cost of this efflux pump in a biofilm environment [301].

### 4.5. Biofilm Formation by Staphylococcus Species

The biofilms of *Staphylococcus aureus* can be classified into *ica*-dependent and *ica*-independent biofilms. The *ica*-dependent biofilms are associated with exopolysaccharide intercellular adhesin/poly-N-acetylglucosamine (PIA/PNAG; product of the *ica* operon) involved in intercellular adhesion [699]. These polysaccharides are encoded by the *icaADBC* operon. The ArcC-type transcriptional regulator Rbf activates the *icaADBC* operon and PIA production in *Staphylococcus epidermidis* by preventing the transcription of the SarR repressor [617]. The ArlRS TCS regulates PNAG synthesis in *Staphylococcus aureus* by repressing *icaR*, a transcriptional repressor of the *icaADBC* operon [549]. The *arl* mutant of *Staphylococcus aureus* showed reduced catheter colonization [549].

The *ica*-independent biofilms have a proteinaceous matrix and are more frequently found in MRSA isolates [700,701]. These biofilms involve fibrinogen and fibronectin-binding proteins FnBPA and FnBPB, which are LPXTG-containing proteins anchored to peptidoglycan [700,702,703]. Loss of sortase (*sarA*), which anchors LPXTG-containing proteins to peptidoglycan, reduced the MRSA biofilm phenotype [700]. Other proteins involved in *ica*-independent biofilm formation include *Staphylococcus aureus* surface proteins C and G (SasC and SasG), extracellular adherence protein (Eap), biofilm-associated protein (Bap), fibronectin-binding proteins (FnBPs), clumping factor B (ClfB), and Staphylococcus protein A (Spa) [704,705,706,707,708,709].

In *Staphylococcus aureus*, the accessory gene regulator (*arg*) QS system is involved in regulating biofilm formation [370]. Silencing the *agr* system strengthens biofilm formation [533]. The increased biofilm thickness of the *agr* mutant is thought to be due to the inability of the cells to detach from the mature biofilm [533]. AgrB, which is associated with the secretion of virulence factors, promotes biofilm dispersion through increased production of the Aur metalloproteinase and the SplABCDEF serine proteases [528,710]. The Arg-mediated detachment of bacteria from the biofilm restored sensitivity to rifampicin [710]. Another reason for the increased biofilm formation of *agr* mutants might be the AgrA-mediated regulation of the phenol-soluble modulin (PSM) operon. PSMs have surfactant-like properties that are involved in biofilm dispersion [711]. Other systems in *Staphylococcus aureus* regulating virulence genes and biofilm formation include the TCSs ArlRS, SaeRS, SrrAB, and GraRS [367,368,441,549,712] (Table 3).

During the biofilm formation of *Staphylococcus aureus*, a small subpopulation undergoes lysis to provide extracellular DNA that glues together the extracellular matrix [593,713,714]. This process is regulated by the holin-like CidABC and the anti-holin-like LrgAB systems [495,593,715]. CidA, which regulates the activities of murein hydrolases, promotes genomic DNA release and biofilm formation [495,715]. LrgAB, which is regulated by the LytSR TCS [289], inhibits CidA-mediated lysis [593]. CidA and LrgA are membrane-associated proteins that form heteromers with disulfide bonds formed between cysteine residues [716]. An *lrgAB* mutant exhibited increased biofilm formation and matrix-associated extracellular DNA [593]. The murein hydrolase AtlA is important for the initial adhesion of *Staphylococcus aureus* to a surface [553]. AtlA is also required for cell division, cell wall turnover, and bacterial lysis [553].

### 4.6. Biofilm Formation by Klebsiella pneumoniae

In *Klebsiella pneumoniae*, some virulence-related genes are involved in biofilm formation. These include the *cps* gene cluster responsible for capsule formation [717], the *fimA* and *mrkA* genes involved in type 1 and type 3 fimbriae formations [718], *wbbM*, which encodes an enzyme involved in the biosynthesis of the O-antigen of LPS [719], *wzm* involved in the transport of LPS [719], *luxS* of the type 2 QS regulatory system [525], and the *pgaABCD* operon responsible for the synthesis and translocation of the poly-β-1,6-N-acetyl-D-glucosamine (PGA) adhesin [720]. The RcsAB TCS regulates the biosynthesis of capsular polysaccharides by upregulating the gene *galF* [721]. Type 1 fimbriae mediate adherence to many types of epithelial cells, while type 3 fimbriae can bind to the extracellular matrix of urinary and respiratory tissues [722]. The *luxS* mutant could form mature biofilms but had a reduced ability to develop microcolonies, especially during the early stages of biofilm formation [525]. PgaC is involved in bile salt-induced biofilm formation [720]. The OxyR transcription factor upregulates the defense mechanisms against oxidative stress, bile salt, and acid stresses, and is important for biofilm formation and the production of types 1 and 3 fimbriae [723].

### 4.7. Antibiotic Resistance of Biofilm-Embedded Bacteria

Biofilm formation by pathogens is a major contributor to antibiotic resistance and treatment failure [20,724,725,726,727,728]. The EPS surrounding the biofilm-embedded bacteria protects the cells from adverse and disruptive environmental conditions. The biofilms can increase microbial tolerance to dehydration [729], radiation [729], extreme temperature and pH [729], osmotic stress, nutrient deficiency, metal toxicity, and antibiotics [8,724,730,731,732].

Bacteria embedded in biofilms are often more resistant to antimicrobial agents than planktonic growing cells [153,503,724,727,731,732,733,734]. There are several explanations for this phenomenon. The biofilms are highly persistent and protected from the immune system as well as adverse conditions such as antibiotics owing to their specialized structures [13,14,104]. Within the biofilm, the bacteria adapt to environmental anoxia and nutrient limitations, showing an altered metabolism and altered gene expression profile, concomitant with lower metabolic activity, reduced cell proliferation, and increased nutrient sequestration [1,13,14]. In addition, biofilm growth is associated with an increased level of mutations [14], and the proximity of the biofilm-embedded cells facilitates the horizontal transfer of resistance genes between the bacteria [20,727,735]. The secretion of β-lactamase from a biofilm-embedded bacteria into the biofilm matrix can prevent the β-lactam antibiotics to act on a neighboring cell even if the latter does not produce the enzyme [727]. Thus, antibiotic resistance can be passively conferred.

Bacterial metabolites can also affect the response to antibiotics [727]. For instance, indole produced by the metabolism of tryptophan can shift the antibiotic sensitivity profiles of neighboring organisms [727]. Indole induces the expression of drug transporters such as *acrD*, *acrE*, *cusB*, *emrK*, *mdtA*, *mdtE*, and *yceL* in *Escherichia coli* [212]. The induction of *acrD*, and *mdtA* by indole is mediated by the BaeSR and CpxAR TCSs [212]. *Escherichia coli*-produced indole protects *Salmonella* Typhimurium from ciprofloxacin by activating OxyR-regulated genes that confer protection from oxidative stress [736]. Another example of cross-species adaptive resistance is the interaction between *Staphylococcus aureus* and *Candida albicans* hyphae, which leads to resistance of *Staphylococcus aureus* to miconazole and vancomycin [737,738].

#### 4.7.1. Prevention of Antibiotic Penetration through the Biofilm

The EPS enwrapping the bacteria in the biofilm are less penetrable for many antibiotic drugs (e.g., ampicillin), although some can diffuse through the biofilm matrix (e.g., ciprofloxacin) [733,739]. The negatively charged polysaccharides (especially the Pel polysaccharide and alginate of *Pseudomonas aeruginosa*) can effectively sequestrate the positively charged aminoglycoside-class of antibiotics such as tobramycin, thus preventing them from penetrating the deeper layers of the biofilm [739,740]. This property of EPS also makes the biofilm-embedded bacteria tolerant to metals such as zinc, copper, and lead [741].

Additionally, the extracellular DNA (eDNA) released to the extracellular matrix of the biofilms has been shown to neutralize the activity of antimicrobial drugs such as tobramycin and antimicrobial peptides, through its cationic chelating properties [454,742]. The eDNA through chelating cations such as magnesium ions forms a cation-limited environment that results in the induction of the PhoPQ and PmrAB TCSs in *Pseudomonas aeruginosa* [472]. These TCSs regulate cationic antimicrobial peptide resistance by upregulating the PA3552-PA3559 operon [472]. The DNA-induced expression of PA3552-PA3559 results in up to a 2560-fold increase in the resistance to cationic antimicrobial peptides and a 640-fold increase in the resistance to aminoglycosides [472]. Wilton et al. [454] observed that the aminoglycoside resistance is caused by aminoarabinose modification of lipid A and the production of spermidine on the outer membrane of the bacteria, both contributing to reduced uptake of aminoglycosides by the bacteria. The addition of L-arginine or sodium bicarbonate that neutralizes the acidic environment caused by eDNA could sensitize *Pseudomonas aeruginosa* to aminoglycosides [454].

The release of eDNA was found to be mediated by the Twin-arginine translocation Tat factor, which acts downstream to the PQS TCS in *Pseudomonas aeruginosa* [743]. *tat* mutants of *Pseudomonas aeruginosa* exhibit reduced eDNA release, defective biofilm architecture, and enhanced susceptibility to tobramycin [743]. In addition, the *tat* mutants showed reduced production of pyocyanin, rhamnolipid, and membrane vesicles that were associated with deficient expression of Rieske iron-sulfur subunit of the cytochrome bc1 complex involved in electron transfer and energy transduction [743].

#### 4.7.2. Antibiotic Tolerance Due to Low Metabolic State of Biofilm-Associated Bacteria

Since the biofilm is a multilayered community of bacteria, nutrient and oxygen gradients are formed from the outer part to the inner part of the biofilms, resulting in metabolically versatile bacterial communities [2,744]. Different bacterial subpopulations showing distinct metabolic activities evolve within the biofilm depending on their spatial localization [745]. Both nutrient sparsity and hypoxia result in a decreased metabolic activity and growth rate of the bacteria in the biofilm core [746]. The low metabolic activities and slow growth rates of the sessile biofilm-associated bacteria and persister cells make them tolerant to antibiotics that rely on cell division (e.g., ciprofloxacin, tobramycin, tetracycline, penicillin) [610,747,748,749,750,751]. Additionally, conditions with nutrient limitations induce antibiotic tolerance in bacteria, which is associated with reduced levels of oxidative stress and depends on the SOS response [752,753]. Some of the dormant bacterial variants develop into antibiotic-tolerant persister cells that can regain cell proliferation when the therapy is withdrawn [41]. The persister subpopulation is, among others, controlled by stress signaling pathways such as the general stress response or the SOS response, in conjunction with the second messenger (p)ppGpp and the toxin–antitoxin modules [41]. The MazEF toxin–antitoxin system is thought to be responsible for biofilm-associated antibiotic resistance and the reduced metabolism resulting in the appearance of persister cells [754,755]. MazF is an endoribonuclease that cleaves single-stranded ACA sequences with consequent translational inhibition, and MazE is an antitoxin that antagonizes MazF [756]. The persistent cells are responsible for chronic infections as they tolerate antibiotics and escape the immune system.

#### 4.7.3. Antibiotic Tolerance Due to Altered Chemical Microenvironment within the Biofilm

The high-affinity quinol oxidase cytochrome bd encoded by the *cydABX* operon which is induced under hypoxic conditions and exhibits high oxygen affinity is expressed at elevated levels within *Escherichia coli* biofilms and plays a role in maintaining the biofilm structure [744]. It is required for aerobic respiration under hypoxic conditions [744], and bacteria lacking cytochrome bd showed reduced expression of EPS and increased sensitivity to exogenous stress stimuli such as oxidative and nitrosactive stress [744]. The latter protection might be related to its quinol peroxidase activity [757]. Further studies show that cytochrome bd promotes antibiotic tolerance in *Escherichia coli* biofilms by promoting the proton-mediated efflux of noxious chemicals through the RND tripartite export proteins [758]. Deletion of cytochrome bd increased the susceptibility of biofilm-embedded *Escherichia coli* to several antibiotics including aminoglycosides, β-lactams, and fluoroquinolones [758].

#### 4.7.4. Activation of Protective Stress Responses

Environmental stress stimuli such as decreased nutrition, lack of oxygen, and lower pH induce stress-response genes including sigma factors that protect the bacteria from antibiotics, host immune factors, and environmental toxins [14,15,759]. For instance, biofilm-associated *Pseudomonas aeruginosa* expresses the gene *ndvB* which encodes a glycosyltransferase that catalyzes the synthesis of periplasmic β-(1→3)-cyclic glucans. The glucans are thought to promote aminoglycoside resistance by sequestering the antibiotics (e.g., tobramycin) in the periplasm away from their cellular target [18]. The expression of *ndvB* in biofilms is dependent on the stationary-phase sigma factor RpoS [759] which is the master regulator of general stress responses [760]. Blocking the alternative sigma factor RpoN (σ^54^), which regulates many virulence factors, increased the susceptibility of *Pseudomonas aeruginosa* to β-lactam antibiotics [761].

#### 4.7.5. Altered Expression of Antibiotic-Resistant Genes in Biofilm-Embedded Bacteria

When *Pseudomonas aeruginosa* biofilms are exposed to β-lactam antibiotics or colistin, various resistance mechanisms are induced such as increased expression of β-lactamase [762,763] and the production of modified lipopolysaccharides that make the bacteria resistant to colistin and other polymyxin antibiotics [764]. Polymyxins are cationic antimicrobial peptides that target Gram-negative bacteria through electrostatic interactions with lipid A and core phosphates of LPS. The resistance to these antibiotics is caused by modification of the lipopolysaccharides through the addition of L-4-aminoarabinose (L-Ara4N) and phosphoethanolamine (PEtN) [764]. These modification systems are regulated by complex networks of two-component systems that sense magnesium, iron, zinc, cationic antimicrobial peptides, and pH [765] (Table 2).

Zhang et al. [766] observed that *tssC1*, which is implicated in type VI secretion (T6S), is upregulated in *Pseudomonas aeruginosa* biofilms. The upregulation of *tssC1* was important for the induction of biofilm-associated antibiotic resistance to tobramycin, gentamicin, and ciprofloxacin [766].

Dale et al. [767] studied genetic determinants that are responsible for biofilm-associated antibiotic resistance in *Enterococcus faecalis* and observed a role for components of the quorum-sensing system (*fsrA*, *fsrC*), the virulence-associated protease *gelE*, and two glycosyltransferase (GTF) genes (*epaI*, *epaOX*). FsrC is a QS histidine kinase that phosphorylates the response regulator FsrA, resulting in the expression of *gelE* [582]. The polysaccharide production by EpaI and EpaOX is thought to make the biofilm less penetrable for antibiotics [767].

*Acinetobacter baumannii* biofilms often show high antibiotic resistance [768,769]. Proteomic analysis of the biofilm-embedded *Acinetobacter baumannii* in comparison to those in the planktonic state showed, among others, upregulation of a membrane Fe transport protein, a sensor histidine/response regulator, diguanylate cyclase, and bacterial antibiotic resistance-related proteins such as β-lactamase PER-1 and aminoglycoside acetyltransferase type I [770].

#### 4.7.6. Increased Efflux Pump Expression in Biofilm-Embedded Bacteria

Another reason for biofilm-associated drug resistance is the increased expression of various efflux pumps in biofilm-associated bacteria [734,771,772]. Bacteria in biofilms show higher horizontal gene transmission than planktonic bacteria [773]. The gene transfer of antibiotic-resistant genes from resistant to susceptible bacterial species within the biofilms leads to antibiotic resistance.

In *Escherichia coli* biofilm, RapA was found to cause antibiotic resistance to β-lactams, norfloxacin, chloramphenicol, and gentamicin through upregulating the *yhcQ* multidrug resistance pump [734]. A *rapA* mutant formed biofilms with reduced content of polysaccharides [734]. These biofilms showed better penetration of antibiotics which may explain the increased sensitivity to antibiotics [734].

The expression of RND efflux pumps and genes involved in type III secretion were upregulated in antibiotic-resistant biofilms of *Pseudomonas aeruginosa* that have developed in the presence of azithromycin [771]. The MexAB-OprM and MexCD-OprJ efflux pumps, but not the type III secretion system, appeared to be integral to biofilm formation in the presence of azithromycin [771].

The PA1874-1877 cluster of genes in *Pseudomonas aeruginosa* encodes for an efflux pump of the ATP-binding cassette (ABC) transporter complex family that is involved in the biofilm-specific antibiotic resistance [772]. Deletion of the genes encoding this pump, resulted in an increase in sensitivity to tobramycin, gentamicin, and ciprofloxacin, especially when the mutant strain was growing in a biofilm [772]. This efflux pump is expressed at a higher level in biofilm cells in comparison to planktonic cells [772].

### 4.8. The Relationship between Biofilm Formation and Efflux Pumps

There are several lines of evidence that efflux pumps can affect biofilm formation and contribute to bacterial colonization and persistence [151,152,719,728,774,775,776]. Efflux pumps can affect biofilm formation directly by mediating the efflux of molecules required for biofilm formation (e.g., EPS) and biofilm-regulatory QS molecules, and indirectly by regulation of transcription factors involved in biofilm formation and bacterial adhesin expression [775].

Efflux pumps are frequently upregulated in biofilms [153,734,771,772,777,778], and deletion of efflux pumps might impair biofilm formation [153,155,158,779,780]. The expression of the *norB*, *norC*, and *mdeA* efflux pump genes were upregulated in *Staphylococcus aureus* during biofilm growth [777]. Deletion of *acrB* and *tolC* in *Salmonella* Typhimurium impaired avian gut colonization [780]. Impaired biofilm formation was observed in *Escherichia coli* upon deletion of *acrB*, *acrD*, *acrE*, *emrD*, *emrE*, *emrK*, or *mdtE* [153,158]. The reduced biofilm formation of an AcrAB-TolC defective *Salmonella* Typhimurium strain was related to the repression of curli biosynthesis [774].

The MexAB-OprM efflux pump of *Pseudomonas aeruginosa* was shown to play an important role in biofilm formation [779]. A Δ*mexAB-oprM* deletion strain was compromised in its capacity to invade and transmigrate across a monolayer of Madin–Darby canine kidney (MDCK) epithelial cells and could not kill mice, in contrast to wild-type bacteria which were highly invasive and caused fatal infection [779]. The defect in the Δ*mexAB-oprM* strain could be complemented by the addition of culture supernatant from MDCK cells infected with wild-type bacteria, suggesting that the efflux pump exports virulence determinants that contribute to bacterial virulence [779]. Another study showed that the MexAB-OprM efflux pump is involved in the secretion of AHL autoinducers [322]. Similarly, the AdeFGH and AcrAB efflux pumps of *Acinetobacter baumannii* have been shown to affect biofilm and pellicle formation [781,782].

A positive correlation has been observed between the expression of the AcrAB efflux pump in drug-resistant *Klebsiella pneumoniae* clinical isolates and biofilm formation [719]. Other studies have found a positive correlation between the AcrAB efflux channel and *Klebsiella pneumoniae* virulence [783,784]. The EefABXC efflux pump of *Klebsiella pneumoniae* contributes to the colonization of the bacteria in the digestive tract [785]. In addition to contributing to antibiotic resistance, this efflux pump confers acid tolerance to inorganic acids [785]. The *eef* promoter can be induced by an acidic environment and by hyperosmolarity [785]. An AcrB knockout strain of *Klebsiella pneumoniae* showed a reduced capacity to cause pneumonia in a murine model [783].

Studies using efflux pump inhibitors (EPIs) showed that these compounds not only sensitize bacteria to antibiotics but also impact biofilm formation [153,154,774,786,787]. The EPIs thioridazine and Phe-Arg-βNaphtylamide (PAβN) reduced biofilm formation in *Escherichia coli*, *Klebsiella pneumoniae*, and *Staphylococcus aureus* [153]. The MexAB-OprM-specific EPI D13-9001 reduced the invasiveness of *Pseudomonas aeruginosa* [788]. These findings support a role for efflux pumps in biofilm formation. Alternatively, it could be that the EPIs affect a common nodule of both processes. For instance, the endocannabinoid anandamide inhibits both drug efflux and biofilm formation of antibiotic-sensitive and antibiotic-resistant *Staphylococcus aureus* species [789]. Anandamide induces immediate membrane depolarization [789], an effect that has implications on both drug export and biofilm formation.

## 5. Targeting Quorum Sensing and Biofilms as a Strategy to Overcome Antibiotic Resistance

The multiple resistance mechanisms that have evolved in the bacteria to protect them from commonly used antibiotics have urged the need for alternative treatment therapies and the development of drugs that act on new scaffolds of targets, with the hope that these approaches could hit the Achilles’ heel of the microorganisms. These drugs can be used as a single agent or in combination with other drugs including antibiotics. The multi-purposing drug therapy aims to target different nodal points, that together will eliminate the bacteria prior to the development of resistance mechanisms. For instance, the β-lactamase inhibitor tazobactam enhanced the synergy between the β-lactam antibiotic piperacillin and daptomycin [790].

Several strategies have been developed to overcome drug resistance. These include (***i***) use of multiple antibiotics to overcome drug resistance [791]. This approach has the advantages of targeting multiple targets in the same bacteria, increasing drug potency by synergistic effects, and suppressing resistance evolution. (***ii***) Use of drugs that target the resistance mechanism(s), thus sensitizing the bacteria to the respective antibiotics [791,792]. (***iii***) Use of drugs that target QS [10,531,793,794]. This approach aims to reduce bacterial virulence, diminish biofilm formation, and increase the susceptibility of the bacteria to antibiotics and human defense mechanisms. (***iv***) Use of drugs with anti-biofilm properties [795]. These drugs should preferentially also disrupt preformed biofilms. The release of bacteria from the biofilms will make them prone to both antibiotics and human immune defense mechanisms. (***v***) Use of drugs that target essential cell division mechanisms such as the Z-ring division protein FtsZ, which will increase the susceptibility to antibiotics [796]. (***vi***) Use of drugs that target cell wall teichoic acid that will interfere with biofilm formation and increase the response to antibiotics [797,798].

Anti-virulence agents represent a promising alternative to the use of antibiotics, as these compounds suppress the production of factors involved in bacterial pathogenicity, without affecting their replication. Thus, it is less likely that therapy resistance might occur. When reducing the ability of the pathogens to colonize and invade host tissue, the drugs will enable the host immune defense mechanisms to eradicate the infection [799].

### 5.1. Antibiotic Adjuvants

Drugs that can sensitize resistant bacteria to antibiotics are termed antibiotic adjuvants [800]. An adjuvant is a compound that usually does not exert an antimicrobial activity by itself but can potentiate the antibiotic activity. Such adjuvants include antibiotic resistance enzyme inhibitors, efflux pump inhibitors, membrane permeabilizers, compounds leading to outer membrane disruption or inner membrane depolarization, anti-virulence compounds, QS, and biofilm inhibitors [10,531,791,792,793,794,800,801,802]. Classical examples of inhibitors of enzymes that cause antibiotic resistance are the FDA-approved β-lactamase inhibitors clavulanic acid, sulbactam, tazobactam, avibactam, and vaborbactam, which sensitize β-lactamase-expressing bacteria to penicillin antibiotics such as amoxicillin [792]. Membrane permeabilizers include polymyxin B, colistin, cationic antimicrobial peptides, glycine peptides, and caragenins [792]. The non-psychotropic phytocannabinoid cannabidiol (CBD) was found to be highly potent against Gram-positive bacteria through disruption of the bacterial membrane [803,804,805]. Interestingly, the combined treatment of CBD with polymyxin B showed a synergistic effect against some Gram-negative bacteria including *Acinetobacter baumannii*, *Klebsiella pneumoniae*, and *Escherichia coli* [804,806,807].

#### Repurposing Clinically Approved Drugs as Antibiotic Adjuvants

Many efforts have been invested in finding clinically approved drugs that can function as antibiotic adjuvants. The advantage of this approach is the well-established knowledge of the pharmacokinetics and toxicology of these compounds [808]. Ejim et al. [809] screened a collection of 1057 FDA-approved drugs to identify compounds that could augment the activity of the antibiotic minocycline. Among these, 69 non-antibiotic compounds including anti-inflammatory, anti-histamine, anti-spasmodic, psychotropic, and anti-hypertensive drugs, exhibited synergy with minocycline against *Staphylococcus aureus*, *Escherichia coli,* and *Pseudomonas aeruginosa* [809]. Among them, the acetaldehyde dehydrogenase inhibitor disulfiram synergized with minocycline to inhibit the growth of *Staphylococcus aureus*. Additionally, the DOPA decarboxylase inhibitor benserazide used for Parkinson’s disease, the serotonin 5-HT receptor antagonist tegaserod used for irritable bowel disease, and the opoid receptor agonist loperamide (Imodium) used to treat diarrhea were found to sensitize *Pseudomonas aeruginosa* to minocycline [809]. Of note, minocycline has been found to act as an inhibitor of the AcrAB-TolC efflux pump expressed in *Enterobacter* species [810].

Other studies have shown that the antiretroviral zidovudine in combination with carbapenems had a synergistic activity against New Delhi Metallo-β-lactamase (NDM-1) *Enterobacteriaceae* strains [811] and the anti-helminthic niclosamine had anti-biofilm and antibacterial activities against *Staphylococcus aureus* [812,813]. Niclosamine has also been shown to have a synergistic effect with colistin on colistin-resistant *Acinetobacter baumannii*, *Klebsiella pneumoniae*, *Pseudomonas aeruginosa*, *Escherichia coli*, and *Enterobacter cloacae* [814,815]. Similarly, the related anti-helminthic drug oxyclozanide could restore colistin sensitivity in drug-resistant *Acinetobacter baumannii*, *Pseudomonas aeruginosa*, *Klebsiella pneumoniae*, *Escherichia coli*, and *Enterobacter cloacae* [816,817].

### 5.2. Quorum Sensing Inhibitors and Quenchers

Due to the tight regulation of biofilm and antibiotic resistance by the QS system, efforts have been made to find QS inhibitors or quenchers that can overcome antibiotic resistance and combat biofilm formation [5,7,8,9,10,11,12,818,819,820,821] (Table 4). QS inhibitors interfere with the action of one or more of the components of the QS system, thereby reducing the virulence and biofilm formation of the bacteria with consequently increased sensitivity to antibiotics. The concept quorum quenchers are often used for enzymes that promote the degradation of the autoinducers, thereby interrupting the QS cascade at the autoinducer receptor interaction. Histidine kinase inhibitors that target TCSs are attractive since these systems are unique to bacteria [822]. Additionally, the catalytic ATP domain of the kinases is highly conserved among the different TCSs and across bacterial species, thereby broadening the spectrum of responsive bacteria [822]. 

The idea of using QS inhibitors in treating biofilm-associated infections came from the observation that certain sea-weed plants (e.g., the Australian macroalga *Delisea pulchra*) never become covered with bacteria, which was found to be due to the production of halogenated furanones which have QS inhibition activities [925,926].

Many secondary plant metabolites have been shown to have anti-quorum and anti-biofilm activity, which may increase the sensitivity of bacteria to antibiotics [509,927,928]. These compounds are part of the plant defense system to avoid microbial infections. These metabolites usually do not have a direct antibacterial activity, but through targeting biofilm formation and QS, they are involved in protecting the plants from infective diseases through mechanisms that act in synergy with antimicrobials [927]. Halogenated thiophenones are examples of QS inhibitors that can control biofilm formation [875]. Baicalin hydrate and cinnamaldehyde target the acyl-homoserine lactone-based QS system present in *Pseudomonas aeruginosa* [531], and hamamelitannin targets the peptide-based system of *Staphylococcus aureus* [531]. Baicalin increased the susceptibility of *Pseudomonas aeruginosa* to tobramycin [531].

Inhibition of the alkyl-quinolone (AQ)-responding PqsR QS component of *Pseudomonas aeruginosa* has been attractive since PqsR activates the *pqsABCDE* operon encoding the autoinducer enzymes PqsABCD and PqsE, which regulates the expression of the cytotoxic galactophilic lectin protein LecA [361,929,930,931] involved in biofilm development [592], and the PhzA1 enzyme involved in the biosynthesis of the virulence factor pyocyanin [743]. Ilangovan et al. [351] developed 2-alkyl-4(3H)-quinazolinone analogs with C7 or C9 alkyl side chains (e.g., 3-NH2-7Cl-C9-QZN) that bind to PqsR and antagonize the PqsR-mediated QS signaling cascade. These inhibitors were found to inhibit *Pseudomonas aeruginosa* virulence at microMolar concentrations [351].

Another approach to attenuate the QS signaling and *Pseudomonas aeruginosa* virulence is to use enzymes (e.g., the Hod 2,4-dioxygenase) that inactivate PQS [932]. Treating *Pseudomonas aeruginosa* with the Hod enzyme reduced the expression of *pqsA* and the virulence determinants lectin A, pyocyanin, and rhamnolipids [932]. However, the proteolytic cleavage of Hod by extracellular proteases together with the competitive inhibition by the PQS precursor 2-heptyl-4(1H)-quinolone reduced the efficiency of Hod as a QS quenching agent [932].

Researchers have also developed methylated or halogenated derivates of the QS precursor anthranilate (e.g., methyl anthranilate and 2-amino-4-chlorobenzoic acid) which inhibit PQS biosynthesis probably by competing with anthranilate for the active site of the PqsA enzyme [882,933]. Methyl anthranilate inhibited the production of PQS and reduced elastase production [882]. However, high concentrations (milliMolars) are required for therapeutic effect [882]. 2-amino-4-chlorobenzoic acid and 2-amino-6-fluorobenzoic acid inhibited HAQ biosynthesis, disrupted PqsR (MvfR)-dependent gene expression, and inhibited osmoprotection [933]. Small molecule inhibitors of PqsD have also been developed [934]. PqsD is a key enzyme in the biosynthesis of 2-heptyl-4-hydroxyquinoline (HHQ) and PQS. The use of the PqsD inhibitor (2-nitrophenyl)(phenyl)-methanol reduced biofilm formation by *Pseudomonas aeruginosa* [934]. Grossman et al. [896] have synthesized a new series of thiazole-containing quinazolinones (e.g., 6-chloro-3((2-pentylthiazol-4-yl)methyl)quinazolin-4(3H)-one) capable of binding to PqsR and inhibiting the production of pyocyanin.

In a *Pseudomonas aeruginosa* infection model in *Caenorhabditis elegans*, the lignans Sesamin and Sesamolin isolated from the *Sesamum indicum* (L.) plant were found to prevent the infection of pre-infected worms through attenuation of QS-regulated virulence factors of the bacteria [818]. Both the lignans exerted anti-QS activity at 75 μg/mL without affecting the bacterial growth. Sesamin and Sesamolin decreased the production of virulence factors such as pyocyanin, proteases, elastase, and chitinase [818]. Additionally, the biofilm constituents of *Pseudomonas aeruginosa* including alginate, exopolysaccharides, and rhamnolipids were affected by the lignans [818]. The lignans acted on the LasR QS system, with minimum effect on the Rhl system [818].

Tryptophan-containing peptides with antibacterial activities impaired QS and biofilm development in multidrug-resistant *Pseudomonas aeruginosa* and increased the susceptibility to ceftazidime and piperacillin [820]. The tryptophan-containing peptides reduced the production of QS-regulated virulence factors by downregulating the gene expression of both the Las and Rhl QS systems [820]. Biofilm formation was inhibited by the tryptophan-containing peptides that were associated with extracellular polysaccharide production inhibition by downregulating *pelA*, *algD*, and *pslA* transcription [820]. In addition, the peptides reduced the expression of the efflux pump genes *oprM*, *mexX*, and *mexA* [820].

Garlic extracts have been shown to block QS in *Pseudomonas aeruginosa* [935] and sensitize the biofilm-embedded bacteria to tobramycin [936]. The garlic extract also provoked a higher degree of inflammation and improved the clearance of the bacteria in an infection model in mice [936]. Olive (*Olea europaea)* leaf extract reduced the expression of *lasI*, *lasR*, *rhlI*, and *rhlR*, with concomitant suppression of virulence and biofilm formation by *Pseudomonas aeruginosa* [937]. Additionally, ginseng (*Panax ginseng*) extract was found to have anti-infective activity against *Pseudomonas aeruginosa* by inhibiting QS [938]. The ginseng extract did not affect bacterial viability and even enhanced the extracellular protein and alginate production [938]. However, it suppressed the production of LasA and LasB and downregulated the synthesis of the AHL molecules [938]. The anti-inflammatory agent itaconic acid (methylenesuccinic acid), which is a metabolite induced during the activation of immune cells [939], increased the anti-biofilm activity of tobramycin on *Pseudomonas aeruginosa*, likely by facilitating the transport of tobramycin through the biofilm [940].

### 5.3. Inhibition of Biofilm Formation

Inhibition of biofilm formation and eradication of established biofilms are major goals to overcome antibiotic resistance and infectious diseases [795,941]. In the previous section, we discussed the inhibition of biofilm formation and sensitization to antibiotics by using QS inhibitors or QS quenchers. However, there are also compounds that prevent biofilm formation whose action mechanisms do not involve the QS system (Table 5). They may act by preventing the production of EPS, adhesion molecules, or pili formation. Efflux pump inhibitors, which will be discussed in Section 5.4, might affect biofilm formation through simultaneous inhibition of other membrane transport systems. Compounds affecting the biosynthesis of cell wall teichoic acid and lipoteichoic acid will be discussed in Section 5.5. Another approach is the inhibition of the transpeptidase Sortase A (SrtA), which is involved in the covalent attachment of adhesive matrix molecules to the peptidoglycan cell wall in Gram-positive bacteria [942], resulting in the inhibition of biofilm formation [858,943,944,945,946,947,948,949].

### 5.4. Inhibition of Efflux Pumps

Another approach to restore the antibacterial activity of antibiotics is to use efflux pump inhibitors (EPIs) that increase the intracellular level of the drugs [162,801,991,992,993,994] (Table 6). Different mechanisms are involved in this activity. Some efflux pump inhibitors (e.g., IITR08027 and anandamide) dispatch the protein gradient required for efflux pump activities [789,995], while others (Phe-Arg-β-naphthylamide (PAβN), phenothiazines, and 1-(1-naphthylmethyl)-piperazine) bind to an efflux pump component such as AcrB [810,996]. Many of the efflux pump inhibitors also prevent biofilm formation as discussed in Section 4.8. Many of the efflux inhibitors have not been approved for clinical uses due to their cytotoxicity, which is seemingly caused by the simultaneous targeting of various membrane-spanning transporters in humans and the need for relatively high doses to achieve the effect [792]. For instance, reserpine causes neurotoxicity via the inhibition of the mammalian ABC-system P-glycoprotein [792]. MP-601205 is an EPI that has been used in clinical trials. It was delivered as an aerosol in patients with ventilation-associated pneumonia or respiratory infections in patients with cystic fibrosis [997]. However, due to tolerability issues, these trials were discontinued [998].

Thiazolidinedione derivatives that target NorA in *Staphylococcus aureus* and sensitize the bacteria to fluoroquinolones [1034] have also been shown to prevent biofilm formation in mixed *Streptococcus mutans*-*Candida albicans* cultures and *Candida albicans* monocultures [982,1035]. Additionally, these compounds exert anti-QS activity against *Vibrio harveyi* by decreasing the DNA-binding activity of LuxR [984]. Additionally, these compounds exerted anti-QS activities against *Pseudomonas aeruginosa* by targeting the LasI quorum-sensing signal synthase [981] and PhsZ, a key enzyme in the biosynthesis of the virulent factor pyocyanin [1036].

The antipsychotic phenothiazine drug chlorpromazine has been shown to potentiate the activities of many antibiotics at subinhibitory concentrations including nalidixic acid, norfloxacin, ciprofloxacin, chloramphenicol, tetracycline, rifampicin, and streptomycin [1019,1040]. Prochlorperazine and trans(E)-flupentixol prevent drug efflux in *Staphylococcus aureus* species by reducing the proton motive force with a concomitant reduction in the transmembrane potential [1041]. Chlorpromazine and amitriptyline were found to be substrates and inhibitors of the AcrB multidrug efflux pump of *Salmonella typhimurium* and *Escherichia coli* [996]. Some clinically approved antimicrobial drugs have been shown to inhibit efflux pumps, including ketoconazole and minocycline [787,810]. Additionally, raloxifene used for treating osteoporosis and pyrvinium used as an anthelmintics were shown to inhibit the NorA efflux pump of *Staphylococcus aureus* and sensitize the bacteria to ciprofloxacin [1024]. The antidiabetic drug metformin prevents drug efflux in *Staphylococcus aureus* and increases their susceptibility to various antibiotics [1015]. These drugs are potential antibiotic adjuvants for the treatment of drug-resistant bacteria.

### 5.5. Targeting Cell Wall Teichoic Acid Synthesis

There are several lines of evidence that the presence of wall teichoic acid (WTA) in the Gram-positive *Staphylococcus aureus* contributes to β-lactam resistance [798,1042,1043,1044]. Moreover, WTA has been shown to protect *Staphylococcus aureus* against the cytotoxic effects of some unsaturated fatty acids [921,1045]. WTA plays an important role in bacterial cell wall processes and integrity and is required for proper cell division, biofilm formation, host colonization, and endovascular infection [1043,1046,1047,1048,1049]. WTA acts as a scaffold for PBP2a and is required for the proper localization of PBP4, thereby contributing to β-lactam resistance [1042]. WTA is also involved in keeping the autolysins/cell wall amidases Sle1, Atl, and LytN in the cross-wall region of the cell wall during cell division of *Staphylococcus aureus* [1050,1051,1052,1053]. Since Sle1 confers β-lactam resistance and promotes biofilm formation in MRSA [1054,1055], WTA may indirectly contribute to β-lactam resistance and biofilm formation via its action on the autolysin.

WTA are anionic polymers composed of repeating units of N-acetylglucosaminyl-ribitol phosphate that are modified by D-alanylation, α-O-GlcNAcylation, and β-O-GlcNAcylation [1056]. WTA are synthesized on a bactoprenol lipid carrier inside the bacterial cell before being transported to the cell surface where they are covalently linked to proteoglycans [1049,1057]. Deletion of tarO, which catalyzes the first step of WTA synthesis, or deletion of *tarS*, which attaches β-O-GlcNAc (β-O-N-acetyl-D-glucosamine) residues to WTA, sensitized MRSA to β-lactam antibiotics [798,1042]. Deletion of the 2-epimerase *mnaA*, which interconverts UDP-GlcNAc and UDP-ManNAc to modulate substrate levels of TarO and TarA, caused complete loss of WTA synthesis and β-lactam hypersensitivity in MRSA and methicillin-resistant *Staphylococcus epidermidis* [1058]. Furthermore, alanine-modified teichoic acids contribute to antimicrobial peptide resistance [1059]. The four proteins DltA-D are essential for the incorporation of D-alanine residues into WTA [1059]. Genetic inactivation of the Dlt system or chemical inhibition (D-alanylacyl-sulfamoyl-adenosine) of D-alanylation of teichoic acid sensitized MRSA to the β-lactams oxacillin and imiprenem [1060] and the cationic antimicrobial peptides nisin and gallidermin [1061]. Upregulation of the *dlt* operon in a *Staphylococcus aureus pitA6* mutant led to daptomycin tolerance [1062].

Several compounds have been tested for their ability to interfere with WTA synthesis and the consequences on antibiotic susceptibility (Table 7). Blocking the synthesis of WTA by inhibiting the first enzyme of the pathway, TarO, by tunicamycin, sensitized MRSA to β-lactams, even in the presence of PBP2a [1043]. The blockage of WTA synthesis led to defects in septation and cell separation [1043]. Ticlopidine, which targets the N-acetylglucosamine-1-phosphate transferase encoded by tarO, increased the sensitivity of MRSA to cefuroxime [798].

The small molecule 1835F03, which inhibits WTA biosynthesis by targeting the TarGH ABC transporter involved in the translocation of WTA across the cell membrane, showed potent antibacterial activity towards several *Staphylococcus aureus* stains including the Newman MRSA strain [1063]. On the basis of this compound, the TarG inhibitor targocil was developed [1044,1063,1064]. Targocil treatment inhibited bacterial growth, induced osmotic stress with subsequent swelling of the cells, and downregulated the expression of numerous virulence factors [1065]. Targocil was also found to block the translocation of the major autolysin Atl across the membrane, resulting in a significant decrease in autolysis [1053]. Compounds targeting TarG could increase the susceptibility of MRSA to β-lactams such as imipenem and oxacillin [1044]. Subinhibitory concentrations of beta-lactam antibiotics could suppress the development of targocil-resistant mutants [1043].

**Table 7 microorganisms-10-01239-t007:** Examples of compounds targeting wall teichoic acid (WTA) synthesis.

Compound	Effects on Bacteria	References
**Clomiphene**	- Clomiphene is a fertility drug that was shown to target the undecaprenyl diphosphate synthase (UppS). Undecaprenyl diphosphate synthase catalyzes the synthesis of a polyisoprenoid essential for peptidoglycan and WTA synthesis. - Clomiphene sensitizes MRSA to various β-lactam antibiotics (e.g., ampicillin, cloxacillin, piperacillin, cefuroxime), as well as bacitracin, which inhibits the dephosphorylation of undecaprenyl diphosphate.	[1066]
**HSGN-94 and HSGN-189**	- HSGN-94 and HSGN-189 inhibit lipoteichoic acid synthesis and prevent biofilm formation by MRSA and vancomycin-resistant *Enterococcus faecalis*.	[1067]
**Targocil**	- Targocil targets TarG, which together with TarH forms the WTA flippase involved in the export of WTA through the membrane to the cell wall. - Targocil inhibits bacterial growth, induces cell wall stress, and downregulates virulence factors in *Staphylococcus aureus.* - Targocil does not sensitize MRSA strains to beta-lactams; but subinhibitory concentrations of beta-lactams suppress the development of targocil-resistant mutants.	[1065]
**Tarocin A and Tarocin B**	- The two tarocins inhibit TarO, the first step in WTA biosynthesis. - The tarocins increase the susceptibility of MRSA to β-lactam antibiotics.	[1068]
**Ticlopidine**	- Ticlopidine is an antiplatelet drug that was found to inhibit WTA synthesis supposedly through inhibition of the TarO enzyme. - Ticlopidine sensitizes MRSA to cefuroxime.	[798]
**Tunicamycin**	- The WTA synthesis inhibitor tunicamycin increases the susceptibility of MRSA to various β-lactam antibiotics but did not affect the susceptibility to vancomycin cycloserine, ciprofloxacin, chloramphenicol, or moenomycin. - Tunicamycin inhibits biofilm formation of *Staphylococcus aureus* and *Listeria monocytogenes*.	[1043,1058,1069]

### 5.6. Inactivation of PBP2a as an Approach to Sensitize MRSA to β-Lactams

The penicillin-binding proteins (PBPs) catalyze in separate domains the transglycosylase and transpeptidase activities involved in the biosynthesis of the cell wall peptidoglycans. β-lactam antibiotics inactivate the transpeptidase activity through irreversible acylation of an active site serine [62]. The PBP2a variant shows low affinity to β-lactams, and therefore continues to catalyze the DD-transpeptidation reaction necessary to complete the cell wall in the presence of these antibiotics [62,1070]. Since PBP2a lacks transglycosylase activity, intact cell wall synthesis requires the presence of the regular PBP2 [62,1070]. β-Lactam resistance is also achieved by elevated expression of PBP4 [1071] and the cell-division proteins FtsA, FtsW, and FtsZ [1072]. FtsZ is required for the proper localization of PBP2 to the division site where it mediates localized peptidoglycan synthesis prior to daughter cell separation [1073].

Otero et al. [1074] identified an allosteric binding domain 60 Å from the DD-transpeptidase active site of PBP2a. The binding of the β-lactam ceftaroline to the allosteric binding domain stimulated the allosteric opening of the active site, enabling a second β-lactam molecule to inactivate the PBP2a [1074]. The ability of compounds to inactivate PBP2a by allostery and thus sensitize the bacteria to antibiotics makes this protein the Achilles’ heel of MRSA [1075].

García-Fernández et al. [1076] observed that the scaffold protein flotillin facilitates oligomerization of PBP2a in functional membrane microdomains (FMMs) and drugs such as the cholesterol-lowering statin zaragozic acid, that disrupt the FMMs, prevent PBP2a oligomerization and sensitize MRSA to β-lactam antibiotics such as methicillin, oxacillin, flucoxacillin, nafcillin, and dicloxacillin. This finding can explain the observed beneficial clinical effects of statins in microbial infections [1077,1078,1079,1080]. Thus, repurposing the cholesterol-reducing drugs of the statin family can be used for overcoming β-lactam resistance to MRSA [1076].

### 5.7. Targeting Cell Division Proteins to Sensitize MRSA to β-Lactams

There are several lines of evidence that targeting bacteria cell division components could sensitize drug-resistant bacteria (e.g., MRSA) to antibiotics (e.g., β-lactams) [796,1081,1082,1083]. In particular, inhibition of the FtsZ cell division protein that forms a contractile ring structure termed the Z-ring at the future division sites has attracted attention [1081,1082,1084]. FtsZ mutant MRSA strains and an MRSA strain treated with FtsZ antisense displayed attenuated virulence and increased susceptibility to β-lactam antibiotics [1072,1083].

Several compounds have been developed to target FtsZ (Table 8). The FtsZ inhibitor PC190723 increases the susceptibility of MRSA to β-lactams [1083]. The effect of FtsZ inhibition is likely due to the delocalization of both FtsZ and PBP2 [1083]. Similarly, Ferrer-González et al. [1085] observed that the FtsZ inhibitor TXA707 leads to disruption of septum formation concomitant with mislocalization of PBPs from midcell to nonproductive peripheral sites [1085]. TXA707 acts synergistically with β-lactam antibiotics with a high affinity for PBP2 [1085]. This research group further observed that in the absence of TXA707, PBP1, PBP2, PBP3, and PBP4 colocalize with FtsZ at the septum of MRSA, while PBP2a localizes to distinct foci at the cell periphery [1086]. TXA707 disrupts septum formation resulting in FtsZ relocalization away from the midcell still interacting with PBP1 and PBP3 [1086]. The interaction of FtsZ with PBP2 and PBP4 was disrupted by TXA707 [1086]. When oxacillin was combined with TXA707, both PBP2 and PBP2a localized to malformed septum-like structures, which might be one mechanism how TXA707 renders MRSA susceptible to β-lactams [1086]. The combined treatment of the TXA707 prodrug TXA709 with the third-generation cephalosporine cefdinir could cure systemic and tissue infections in mice [1087].

## 6. Conclusions

In this review, we have presented data showing various mechanisms involved in antibiotic resistance of various ESKAPE pathogens with a specific emphasis on the triangle interconnection between quorum sensing, biofilm formation, and antibiotic resistance. A myriad of synthetic and natural compounds has been studied for their ability to interfere with one or more of these processes with the aim to eradicate biofilm and sensitize the bacteria to antibiotics. Altogether, these studies show that it is sufficient to target one or few nodal processes to overcome antibiotic resistance. A combination of these adjunctive compounds together with conventional antibiotics might be necessary to optimize their efficacy which is expected to improve the treatment regime against drug-resistant bacterial infections.

## Figures and Tables

**Figure 1 microorganisms-10-01239-f001:**
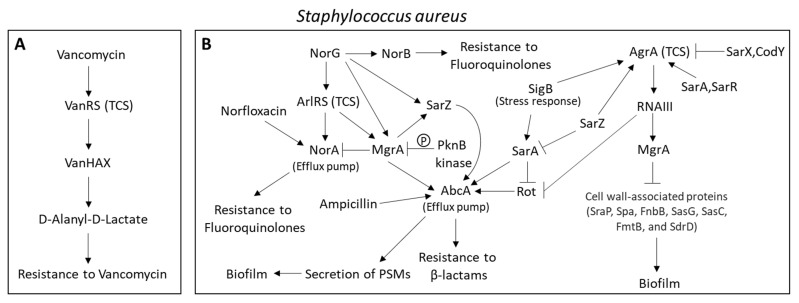
(**A**). **Induction of vancomycin resistance in *Staphylococcus aureus* by vancomycin**. Vancomycin activates the TCS VanRS, which induces the expression of the *vanHAX* operon responsible for the synthesis of the D-alanyl-D-lactate dipeptide. The D-alanyl–D-lactate dipeptide shows a 1000-fold lower affinity to vancomycin compared to the regular D-alanyl–D-alanine dipeptide, thereby conferring vancomycin resistance. (**B**). **Examples of regulatory mechanisms involved in antibiotic resistance and biofilm formation in *Staphylococcus aureus***. The expressions of the efflux pumps NorA and AbcA, which confer antibiotic resistance to fluoroquinolones and β-lactams, respectively, are induced by their respective substrates norfloxacin and ampicillin. Additionally, their expression levels are influenced by the ArlRS and Agr TCSs, which both affect the global transcriptional regulator MgrA (also known as NorR). Phosphorylation of MgrA by the PknB serine/theonine kinase leads to increased transcription of *norA*. The ArlRS and Agr TCSs are regulated by several transcriptional factors (e.g., NorG, Rot, SarA, SarR, SarZ, SigB) as illustrated in the figure. This network of regulatory factors also affects biofilm formation, where AbcA promotes biofilm formation by exporting phenol-soluble modulins (PSMs), and the Agr QS prevents biofilm formation. The Agr QS is explained in more detail in Section 3.3.

**Figure 2 microorganisms-10-01239-f002:**
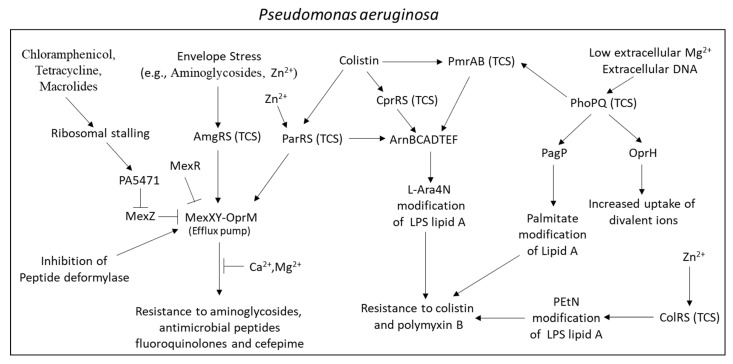
**Induction of antibiotic resistance in *Pseudomonas aeruginosa* by antibiotics, zinc ions, and low concentrations of extracellular magnesium ions**. The expression of the MexXY-OprM efflux pump responsible for multidrug resistance is regulated by several factors including ribosome-targeting antibiotics and the TCSs AmgRS and ParRS. AmgRS is activated by envelope stress, and ParRS is activated by the cationic antimicrobial peptide colistin as well as zinc ions. Moreover, colistin activates other TCSs including CprRS and PmrAB. The latter TCS is also affected by the TCS PhoPQ, which is activated by low extracellular magnesium ion concentrations, and extracellular DNA which sequesters magnesium ions. Extracellular DNA is a central component of *Pseudomonas aeruginosa* biofilms. The TCSs ParRS, CprRS, and PmrAB induce the translation of the *arnBCADTEF* operon which is responsible for the L-Ara4N modification of LPS, resulting in resistance to colistin. Additionally, LPS is modified by palmitoylation through PhoPQ-mediated upregulation of *pagP*, and by PEtN attachments regulated by the ColRS TCS.

**Figure 3 microorganisms-10-01239-f003:**
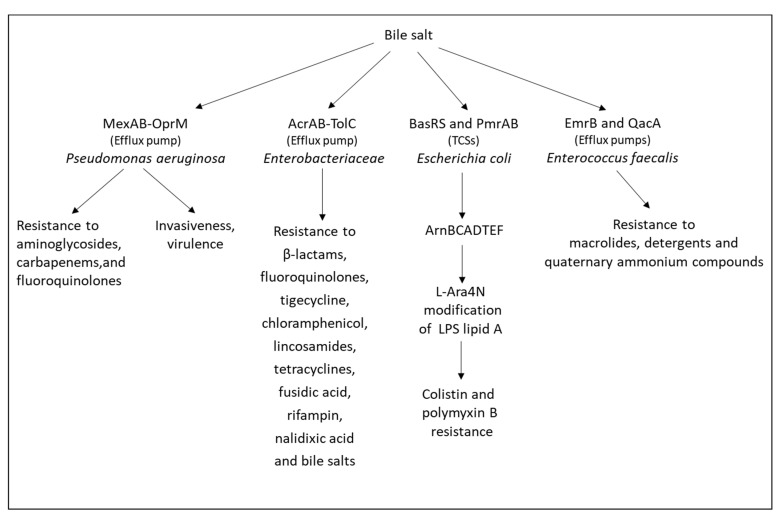
**Examples of antibiotic resistance mechanisms induced by bile acids.** Bile salts which are secreted into the duodenum, induce the expression of various genes in bacteria that confers antibiotic resistance. Outstanding is the upregulation of efflux pumps such as MexAB-OprM in *Pseudomonas aeruginosa*, AcrAB-TolC in *Enterobacteriaceae* and EmrB/QacA in *Enterococcus faecalis* that confer resistance to multiple antibiotics and antiseptics, as well as to the bile salts themselves. Additionally, the TCSs BasRS and PmrAB are induced in *Escherichia coli*, resulting in the upregulation of the *arnBCADTEF* operon responsible for the L-Ara4N modification of LPS. This modification reduces the affinity of colistin/polymyxin B to LPS, with consequent resistance to these drugs.

**Figure 4 microorganisms-10-01239-f004:**
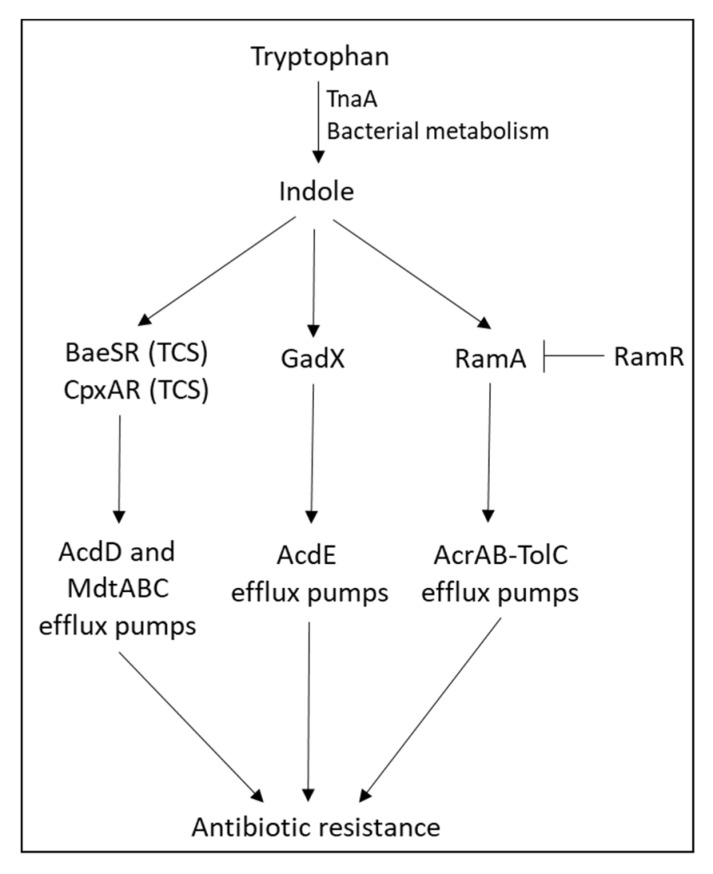
**Induction of antibiotic resistance by the tryptophan metabolite indole.** Indole, which is produced by various gut bacteria including *Escherichia coli*, induces the expression of various efflux pumps (e.g., AcdD, MdtABC, AcdE, AcrAB-TolC) through activation of TCSs (e.g., BaeSR, CpxAR) or transcriptional regulators (GadX, RamA). The activity of RamA is negatively regulated by RamR.

**Figure 5 microorganisms-10-01239-f005:**
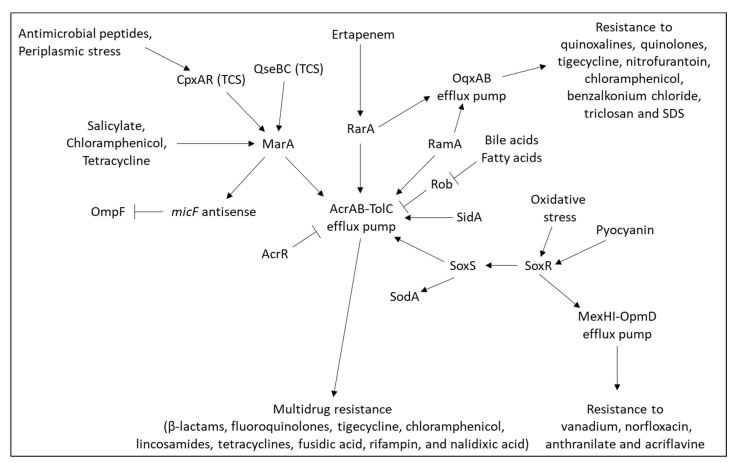
**Regulation of AcrAB-TolC efflux pump expression in Gram-negative bacteria.** The expression of AcrAB-TolC, which confers multidrug resistance, is regulated by several transcriptional regulators including MarA, RarA, RamA, Rob, AcrR, SidA, and SoxS. MarA, in turn, is regulated by the TCSs CpxAR and QseBC, as well as various antibiotics. MarA reduces the expression of the OmpF porin which is required for the penetration of several antibiotics into the bacteria. RarA is activated by the antibiotic ertapenem. Besides upregulating AcrAB-TolC, RarA increases the expression of the OqxAB multidrug efflux pump. Bile acids and fatty acids increase AcrAB-TolC expression through repression of the Rob transcriptional regulator. SoxS is regulated by SoxR whose activity is influenced by oxidative stress, as well as by the *Pseudomonas aeruginosa*-produced pyocyanin pigment. In turn, SoxS increases the expression of SodA superoxide dismutase, which is a mechanism to protect the bacteria from oxidative stress. Additionally, SoxR induces the expression of the MexHI-OpmD efflux pump, thus conferring resistance to additional compounds.

**Figure 6 microorganisms-10-01239-f006:**
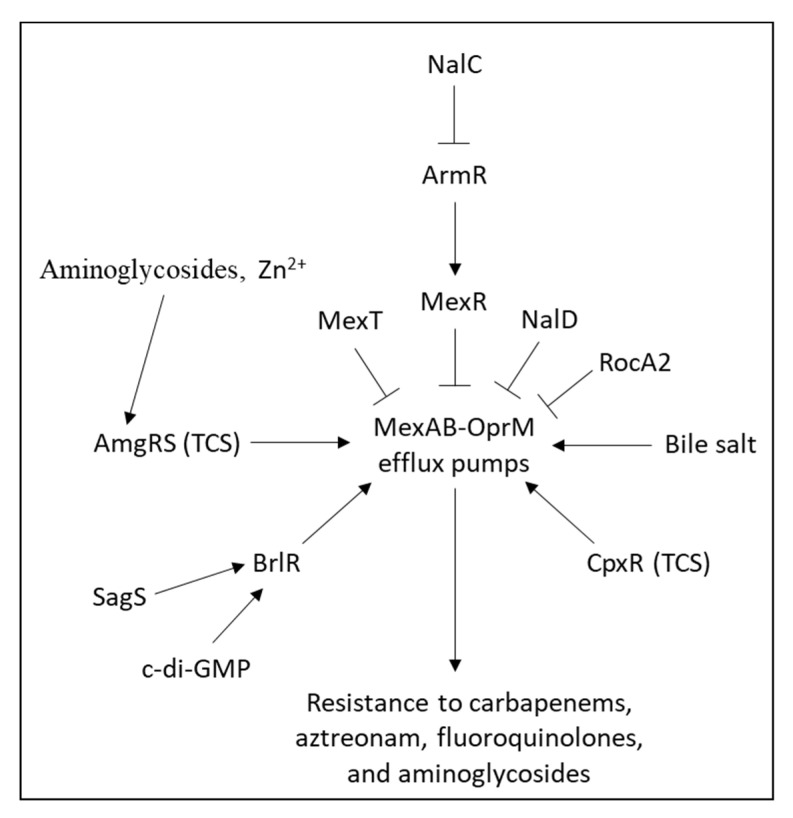
**Regulation of MexAB-OprM multidrug efflux pump expression in *Pseudomonas aeruginosa*.** The expression of the MexAB-OprM efflux pump is positively and negatively regulated by a whole range of transcriptional regulators. Its expression is also affected by bile salts and induced by the AmgRS TCS and CpxR, which is the cognate response regulator of the CpxAR TCS.

**Figure 7 microorganisms-10-01239-f007:**
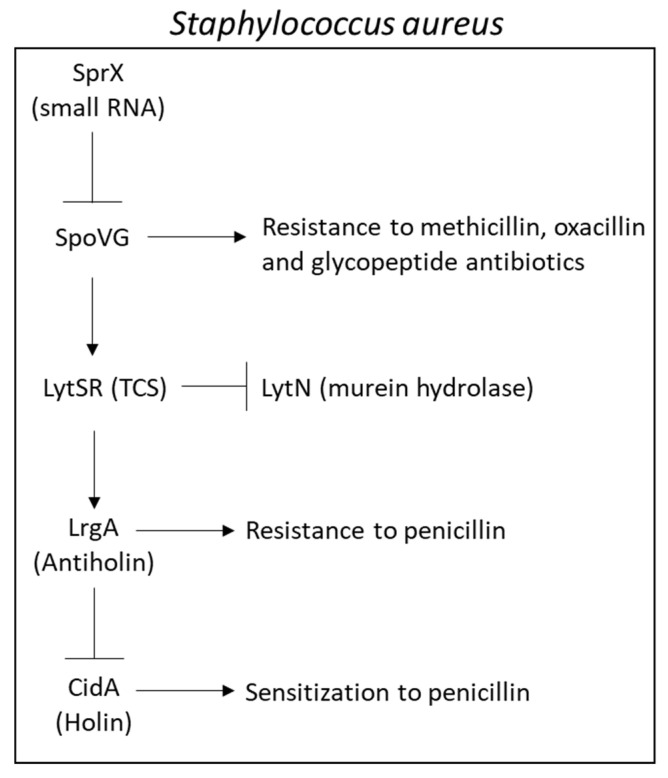
**Regulation of the antiholin LrgA–holin CidA system by the small SprX RNA.** SprX inhibits the expression of the RNA-binding protein SpoVG, which positively regulates the TCS LytSR. LytSR positively regulates the antiholin LrgA which antagonizes the activity of the holin CidA. SpoVG induces methicillin and oxacillin resistance by promoting cell wall synthesis and inhibiting cell wall degradation. LrgA and CidA affect the response to penicillin by respectively inhibiting or activating murein hydrolase activities. Altogether, the overexpression of SprX results in increased susceptibility to β-lactams.

**Figure 8 microorganisms-10-01239-f008:**
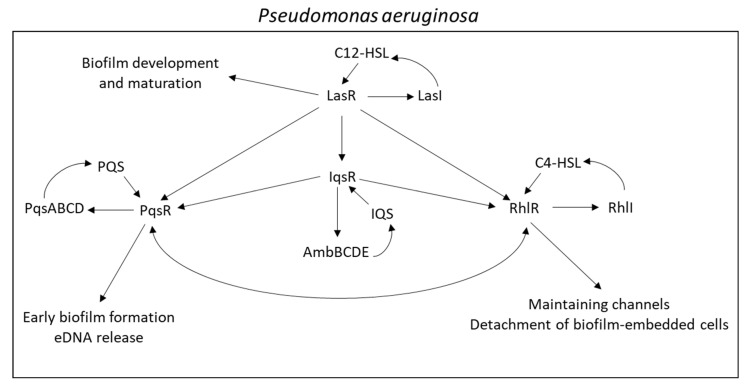
**Crosstalk of the four QS systems Las, Rhl, Pqs, and Iqs in *Pseudomonas aeruginosa* and their relationship to biofilm formation**. Each of the four TCS systems produces its own autoinducer (C12-HSL, PQS, IqsR, or C4-HSL) that acts on the corresponding receptor (LasR, PqsR, IqsR, or RhlR). The receptors, in turn, elicit phosphorelay cascades resulting in altered expression of a large number of genes. In addition, there is crosstalk between these TCSs. LasR affects PqsR, IqsR, and RhlR. There is also mutual communication between PqsR and RhlR. LasR is involved in biofilm development and maturation. PqsR is involved in early biofilm formation and the release of extracellular DNA. RhlR is involved in the maintenance of biofilm channels and the detachment of biofilm-embedded cells. IqsR promotes biofilm formation by acting on PqsR and RhlR.

**Figure 9 microorganisms-10-01239-f009:**
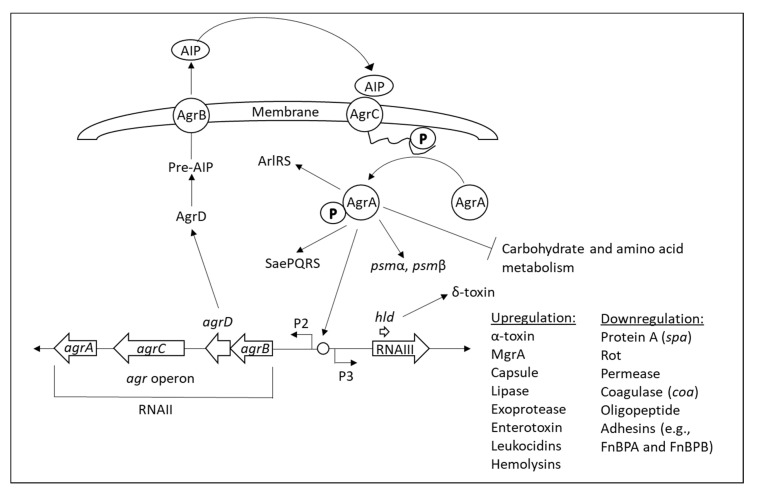
**The Agr QS system of *Staphylococcus aureus*.** The *arg* operon consists of the 4 genes: *agrB*, *agrD*, *agrC*, and *agrA*. AgrD encodes for a precursor of the autoinducer peptide AIP that is processed and transported through the cell membrane by AgrB. The mature AIP interacts with its receptor AgrC eliciting a phosphorelay, resulting in the phosphorylation and activation of the response regulator AgrA that affects the expression of multiple genes including the genes of the *agr* operon by binding to the P2 promoter, and the RNAIII regulatory RNA by binding to the P3 promoter. RNAIII affects the expression of a large number of genes, thereby promoting the virulence of *Staphylococcus aureus*. A small part of the RNAIII transcript encodes for the δ-toxin (delta-hemolysin*; hld*). Additionally, AgrA activates the TCSs ArlRS and SaePQRS, as well as upregulating the expression of the virulence factors phenol-soluble modulins (PSMs) alpha and beta. As shown in Figure 1B, ArlRS modulates the expression of the NorA efflux channel responsible for fluoroquinolone resistance and regulates virulence factors through the induction of MgrA. The SaePQRS TCS regulates the expression of various virulence factors.

**Figure 10 microorganisms-10-01239-f010:**
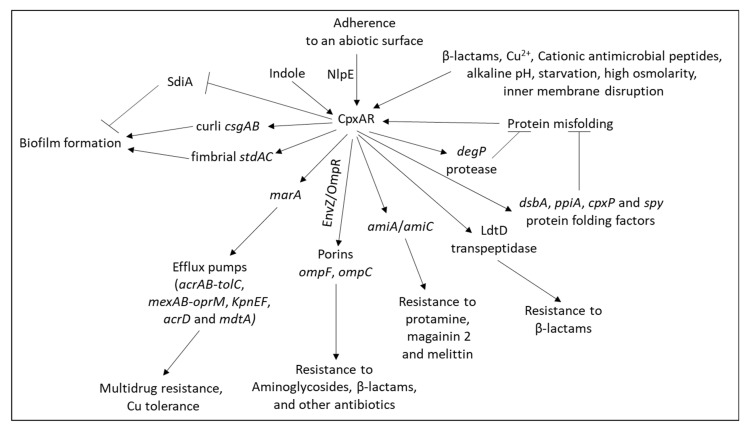
**The Cpx envelope stress-response system.** The CpxAR TCS is activated by various stress stimuli including protein misfolding, certain antibiotics, inner membrane disruption, alkaline pH, starvation, high osmolarity, and adherence to an abiotic surface. The activation of CpxAR leads to large alterations in gene expression in an attempt to protect the bacteria from the environmental stressor. Among others, it induces the expression of genes assisting in removing misfolded proteins and genes involved in biofilm formation. Additionally, CpxAR increases the expression of various genes (e.g., efflux pumps, porins, amidases, and the *ldtD* transpeptidase) that confer multidrug resistance and copper tolerance.

**Figure 11 microorganisms-10-01239-f011:**
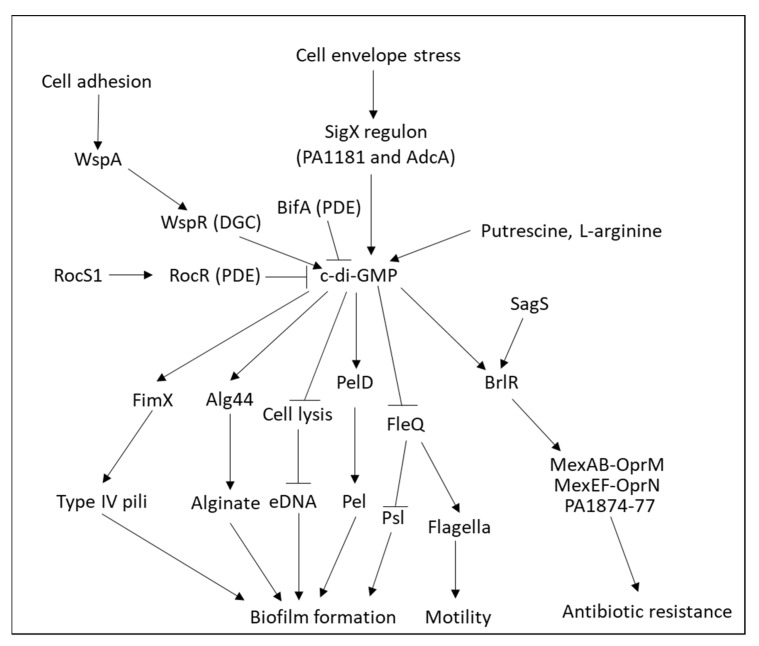
**The role of c-di-GMP in biofilm formation and antibiotic resistance.** The secondary metabolite c-di-GMP is a central mediator of biofilm formation in Gram-negative bacteria. It modulates the expression of various genes (e.g., *fimX*, *alg44*, *pelD*, *fleQ*) involved in the production of pili, flagella, alginate, and exopolysaccharides (Pel and Psl) that contribute to the development of the biofilm. Moreover, it induces the transcriptional responser BrlR that regulates the expression of efflux pumps involved in antibiotic resistance. The activity of BrlR is affected by SagS, which is involved in the molecular switch from a planktonic to a biofilm lifestyle (see Figure 12 below). The c-di-GMP level is affected by various factors including cell adhesion and cell envelope stress. Its synthesis is mediated by diguanylate cyclases (DGCs) and degraded by c-di-GMP phosphodiesterases (PDEs).

**Figure 12 microorganisms-10-01239-f012:**
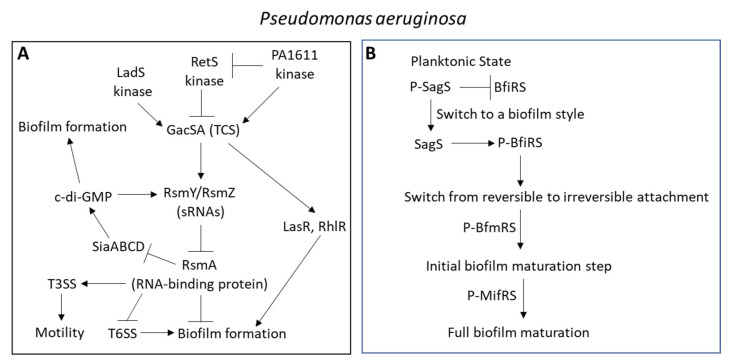
(**A**). **Regulation of RsmY/RsmZ small RNAs by the GacSA TCS and c-di-GMP in *Pseudomonas aeruginosa* and the impact on biofilm formation**. RsmY and RsmZ are small regulatory RNAs involved in the regulation of biofilm formation. Their expression is regulated by the TCS GacSA, whose activity is, in turn, regulated by the LadS, RetS, and P1611 kinases. RsmY and RsmZ sequester the translational repressor RsmA, thereby relieving its inhibitory action on c-di-GMP production and biofilm formation. Since RsmA increases motility through the type III secretion system (T3SS), an increase in the RsmY and RsmZ levels would result in reduced motility. (**B**). **Additional TCS components regulating biofilm formation**. In the planktonic state, SagS is phosphorylated and prevents the activity of BfiSR. Dephosphorylation of SagS leads to phosphorylation of BfiSR resulting in the switch from a reversible attachment to an irreversible attachment. This is followed by phosphorylation of BfmSR, which initiates biofilm maturation, and phosphorylation of MifSR, which is required for full biofilm maturation.

**Table 2 microorganisms-10-01239-t002:** Examples of two-component systems (TCSs) involved in antibiotic resistance.

TCS Involved in Antibiotic Resistance	Function	Bacterial Species	References
**AdeRS**	- AdeRS induces the expression of *adeABC* multidrug efflux pump that confers resistance to various antibiotics including tigecycline, aminoglycosides, chlorhexidine, and pentamidine. - Inactivation of AdeS leads to aminoglycoside sensitivity. - Mutation of AdeR leads to constitutive expression of the *adeABC* efflux pump and antibiotic resistance.	*Acinetobacter baumannii*	[226,389,390,391,392]
**AirSR (YhcSR)**	- An *airSR* mutant showed reduced autolysis rates and higher susceptibility to vancomycin. - Mutation in *airR* caused increased resistance to hydrogen peroxide, vancomycin, norfloxacin, and ciprofloxacin under anaerobic conditions. - AirSR regulates the nitrate respiratory pathway. - AirSR senses oxygen tension and regulates redox signaling. - AirSR regulates the expression of genes that function in cell wall metabolism (*cap*, *pbp1*, and *ddl*) and autolysis (*lytM*).	*Staphylococcus aureus*	[393,394]
**AmgRS**	- AmgRS is a membrane stress-responsive TCS that causes intrinsic tobramycin resistance and increases virulence. - AmgRS is activated by aminoglycosides such as paromomycin. - AmgRS induces the expression of various genes that contribute to aminoglycoside resistance including the *mexXY* and *mexAB*-*oprM* efflux pumps; a modulator of FtsH protease (YccA); a membrane protease (HtpX); and the membrane protein PA5528. - AmgRS is activated by zinc ions and protects the bacteria from zinc-mediated membrane damage.	*Pseudomonas aeruginosa*	[52,197,395,396,397]
**ApsRS**	- ApsRS confers resistance to antimicrobial peptides by inducing the expression of the *dlt* operon that adds D-alanine to teichoic acid in the cell wall, and *mprF* (*fmtC*) that adds lysine to phosphatidylglycerol in cell membranes.	*Staphylococcus aureus*	[381,382,383]
**ArlRS**	- ArlRS modulates *norA* gene expression. - *arlS* mutation leads to increased *norA* expression, with consequent resistance to fluoroquinolones. - Deletion of *arlRS* in MRSA USA300 strain resulted in increased susceptibility to oxacillin, which was attributed to a downregulation of *spx*, a global regulator that impacts stress tolerance. Overexpression of *spx* restored oxacillin resistance in the Δ*arlRS* mutant. - ArlRS affects autolysis, biofilm formation, capsule synthesis, and virulence. - ArlRS activates the expression of the global regulator MgrA.	*Staphylococcus aureus*	[182,398,399,400]
**BaeSR**	- BaeSR functions as a stress responder under high osmotic conditions. - BaeR confers resistance to novobiocin and deoxycholate by upregulating the expression of the *yegMNOB* (*mdtABCD*)–*baeSR* operon by binding to the yegM promoter in *Escherichia coli*. The MdtB/MdtC efflux pump confers resistance to bile salts in *Escherichia coli*. The YegO and YegN RND transporters, but not the YegB MFS transporter, confer antibiotic resistance to novobiocin. - Overexpression of *baeR* conferred resistance to carbenicillin, aztreonam, and carumonam in *Escherichia coli*. - In *Salmonella typhimurium*, ciprofloxacin induces *mdtA* expression via activation of BaeRS. MdtA pumps out novobiocin and deoxycholate. - BaeR regulates tigecycline susceptibility in *Acinetobacter baumannii* through positively regulating the efflux pump genes *adeA*, *adeB*, *adeIJK*, and *macAB-tolC*.	*Escherichia coli*, *Salmonella typhimurium*, *Acinetobacter baumannii*	[401,402,403,404,405,406]
**BasSR**	- BasSR regulates susceptibility to polymyxin, deoxycholic acid, sodium dodecyl sulfate, fluoroquinolones, tetracyclines, erythromycin, clindamycin, and lincomycin. - BasSR regulates the expression of the multidrug efflux pump EmrD.	*Escherichia coli*	[407]
**BqsRS**	- BqsRS induces the expression of *arnBCADTEF*, which confers resistance to polymyxins and other antimicrobial peptides. - BqsRS senses extracellular Fe^2+^ ions.	*Pseudomonas aeruginosa*	[408]
**BraRS**	- BraRS induces resistance to bacitracin and nisin by upregulating the ABC transporters *braAB*, *braDE*, and *vraDE*. - Low concentrations of bacitracin and nisin induce BraRS-dependent transcription of *braDE* and *vraDE*. BraDE is involved in bacitracin sensing and signaling through BraRS. VraDE acts as a detoxification module.	*Staphylococcus aureus*	[409,410]
**ChtRS**	- ChtRS contributes to chlorhexidine and bacitracin tolerance.	*Enterococcus faecium*	[411]
**CiaRH**	- CiaRH is a TCS induced by cell wall stress caused by cycloserine, bacitracin, and vancomycin. - CiaRH is required for survival under various lysis-inducing conditions and confers resistance to cell wall inhibitors including β-lactams, cycloserine, bacitracin, and vancomycin. - CiaRH promotes intracellular survival within neutrophils and macrophages and protects against antimicrobial peptides and reactive oxygen species.	*Streptococcus pneumoniae*	[412,413,414]
**ColRS**	- ColRS regulates the expression of a pEtN transferase that modifies LPS by transferring phosphoethanolamine to lipid A, thus resulting in polymyxin resistance. - ColRS is activated by zinc ions.	*Pseudomonas aeruginosa*	[415]
**CopRS**	- In the presence of copper or zinc ions, CopR represses the *oprD* porin, resulting in carbapenem resistance.	*Pseudomonas aeruginosa*	[416,417]
**CprRS**	- CprRS induces the expression of *arnBCADTEF*, which adds arabinosamine to LPS, thereby causing resistance to antimicrobial peptides.	*Pseudomonas aeruginosa*	[418,419]
**CpxAR**	- CpxAR is a membrane stress-response TCS that is associated with the resistance to several cell-envelope-targeting drugs. - CpxAR induces the expression of the KpnEF efflux pump. - CpxAR induces the expression of N-acetylmuramoyl-L-alanine amidases *amiA* and *amiC*, which confer drug resistance to protamine. - CpxAR activates the expression of the *marRAB* operon that facilitates multidrug resistance by enhancing the expression of TolC tripartite multidrug transporters. - CpxAR regulates the expression of the porins *ompC* and *ompF*. - It contributes to Cu tolerance. - CpxAR is induced by antimicrobial peptides including polymyxin B, melittin, and LL-37, as well as copper ions. - CpxAR responds to periplasmic stress including misfolded proteins, inner membrane disruption, alkaline pH, starvation, and high osmolarity.	*Escherichia coli*, *Klebsiella pneumoniae*, *Salmonella typhimurium* and other Gram-negative bacteria	[420,421,422,423,424,425]
**CreBC**	- CreBC is a TCS activated by low concentrations of the β-lactam cefoxitin that inhibits PBP4, resulting in resistance to β-lactam due to induction of the AmpC cephalosporinase.	*Pseudomonas aeruginosa*	[426]
**CroRS**	- CroRS drives resistance to ampicillin and cephalosporins, among others, by upregulating PBP5 with low affinity to β-lactams. - The IreK kinase positively affects CroR-dependent gene expression through threonine phosphorylation of CroS, resulting in antibiotic resistance. - CroRS is activated by antibiotic-induced cell wall damage.	*Enterococcus faecium*	[427,428,429,430]
**CzcRS**	- Activation of CzcRS by zinc, cadmium, and cobalt ions leads to the expression of the CzcCBA efflux pump which expels these ions. - CzcR confers carbapenem resistance by repressing the expression of *oprD* porin, which is required for the uptake of these antibiotics.	*Pseudomonas aeruginosa*	[416,431,432,433]
**EvgAS**	- EvgAS confers resistance to bile salts by upregulating the *emrKY* and *yhiUV* efflux pump genes.	*Escherichia coli*	[434,435,436,437,438]
**GraRS (ApsRS)**	- GraRS regulates the expression of the ABC transporter genes *vraFG*, which confer resistance to vancomycin; the lysyl-phosphatidylglycerol synthase *mprF*, which adds positively charged lysine residues to phosphatidylglycerol; and *dltABCD*, which contributes to the net positive surface charge by covalently attaching D-alanine to the cell wall teichoic acids. - Besides vancomycin, GraRS confers resistance to cationic antimicrobial peptides such as cathelicidin LL-37, nisin A, and class I bacteriocins, which is supposedly due to its induction of the *dlt* operon involved in D-alanylation of teichoic acid. - GraRS is induced by polymyxin B and the host defense tPMP-1 (Thrombin-induced platelet microbicidal proteins) and the synthetic RP-1. - Δ*graRS* MRSA strains showed increased susceptibility to ampicillin, oxacillin, vancomycin, gentamicin, antimicrobial peptides, and neutrophil defense mechanisms.	*Staphylococcus aureus*	[410,439,440,441,442,443]
**LiaFSR**	- LiaFSR confers resistance to bile salts, daptomycin, and telavancin.	*Enterococcus faecium*, *Enterococcus faecalis*	[210,444]
**LrgAB**	- LrgAB has an inhibitory effect on murein hydrolase activity. - A *lrgAB* mutant showed increased extracellular murein hydrolase activity compared to that of the wild-type strain. - LrgAB promotes antibiotic tolerance to penicillin. - LrgAB is regulated by the LytSR TCS.	*Staphylococcus aureus*	[289]
**LytSR**	- LytSR is a sensor of altered membrane potential that confers resistance to antimicrobial peptides. - LytSR affects the murein hydrolase activities by upregulating *lrgA* and *lrgB*. LrgA is an antiholin-like protein that inhibits autolysis. - *lytSR* mutants are more sensitive to cationic antimicrobial peptides.	*Staphylococcus aureus*	[445,446]
**NsaRS**	- NsaRS confers resistance to the lantibiotic nisin. - NsaRS induces the expression of the ABC transporters *braDE* and *vraDE*. BraDE senses nisin and physically interacts with NsaSR. VraDE is important for the detoxification of nisin. - NsaRS is activated by various cell wall damaging antibiotics including phosphomycin, β-lactams, nisin, and gramicidin.	*Staphylococcus aureus*	[447,448]
**ParRS**	- ParRS upregulates the expression of the *mexXY* multidrug efflux operon. - ParRS represses the expression of the carbapenem-selective porin *oprD*. - ParRS induces the expression of *arnBCADTEF* in response to indolicidin and polymyxin, resulting in resistance to polymyxins and other antimicrobial peptides.	*Pseudomonas aeruginosa*	[66,189,419,449,450]
**PhoBR**	- PhoBR increases the expression of the *kpnO* porin, which affects the susceptibility to tetracycline, nalidixic acid, tobramycin, streptomycin, and spectinomycin. - Both *phoB* mutant and Δ*kpnO* mutant show decreased susceptibility to antibiotics. - The Δ*kpnO* mutant showed reduced virulence.	*Klebsiella pneumoniae*	[451]
**PhoPQ and PmrAB**	- Both PhoPQ and PmrAB induce the expression of *arnBCADTEF* in response to low Mg^2+^ concentration, resulting in resistance to cationic antimicrobial peptides, polymyxins, and aminoglycosides. - PhoPQ regulates the enzyme *pagP*, which catalyzes the transfer of palmitate from outer membrane phospholipids to lipid A, thus conferring resistance to some antimicrobial peptides. - The PhoPQ signaling cascade is induced under conditions of low extracellular Mg^2+^ conditions, resulting in the induction of the *pmrB*/*pmrA* TCS, which confers polymyxin resistance, and the small outer membrane protein *oprH*, which facilitates the uptake of divalent cations. - PhoPQ is activated by sub-MIC concentrations of polymyxin in *Salmonella enterica*. - In *Salmonella enterica*, PhoP can activate *pmrAB*. - PmrAB is activated by Fe^3+^ and low pH. - In *Klebsiella pneumoniae*, PhoP activates PmrD that in turn binds to and activates PmrA responsible for polymyxin B resistance.	*Pseudomonas aeruginosa*, *Salmonella enterica*, *Klebsiella pneumoniae*, *Acinetobacter baumannii*	[91,419,452,453,454,455,456,457,458,459,460]
**QseBC**	- QseBC increases the transcription of efflux pump-associated genes *marA*, *acrA*, *acrB*, *acrD*, *emrD*, and *mdtH.*	*Escherichia coli*	[249]
**TarRS**	- TarRS confer multidrug resistance through interaction with the chaperonin GroEL. - A Δ*tar*-Δ*tas* strain shows high susceptibility to kanamycin, gentamycin, tetracycline, and imipenem. - The regulator FleQ indirectly activates *tarR*-*tarS* transcription. - c-di-GMP antagonizes FleQ activation and thus inhibits *tarR*-*tarS* transcription. - The sigma factor FliA directly activates *tarR*-*tarS* transcription.	*Pseudomonas aeruginosa*	[461]
**VanRS**	- Binding of vancomycin and teicoplanin to VanS induced a long-lived phosphorylation state of VanS, resulting in the phosphorylation of VanR, and the consequent regulation of *vanA* responsible for vancomycin resistance.	*Enterococcus faecium*	[462]
**VraTSR**	- VraTSR is a three-component system that senses and responds to cell wall stress. - VraTSR regulates genes associated with cell wall synthesis including the transpeptidase *pbp2*, the transglycosylase *sgtB*, and the UDP-N-acetylglucosamine enolpyruvyl transferase *murZ*. - VraTSR confers resistance to cell wall antibiotics including methicillin, vancomycin, cycloserine, teicoplanin, bacitracin, and daptomycin. - Inhibition of VraTSR by small molecules sensitized MRSA to oxacillin.	*Staphylococcus aureus*	[463,464,465,466,467]
**WalKR (YycFG, VicRK, MicAB)**	- WalKR is involved in the regulation of cell wall synthesis and cell division through the regulation of autolysin production such as *lytM*, *atlA*, *sle1*, *ssaA*, and *sceD*. - Single nucleotide substitutions within either *walK* or *walR* led to resistance to vancomycin and daptomycin.	*Staphylococcus aureus*	[468,469,470]

**Table 4 microorganisms-10-01239-t004:** Examples of QS inhibitors/QS quenchers.

Compound	Effects on Bacteria	References
**Ajoene**	- Ajoene represses quorum sensing, virulence, and biofilm formation by reducing the sRNAs RsmY and RsmZ in *Pseudomonas aeruginosa* and reduces the regulatory RNAIII in *Staphylococcus aureus*. - Ajoene acts synergistically with tobramycin to kill *Pseudomonas aeruginosa* in biofilms.	[823,824]
**Allicin**	- Allicin inhibits the production of *Pseudomonas aeruginosa* virulence-associated factors, such as elastase, pyocyanin, pyoverdine, and rhamnolipids, by inhibiting the Rhl and Pqs QS systems.	[825]
**Baicalin (5,6,7-** **trihydroxyflavone) and 3,5,7-** **Trihydroxyflavone**	- Baicalin is an AHL-based QS inhibitor. - Sub-MIC concentrations of baicalin inhibit *Pseudomonas aeruginosa* biofilm formation, reduce virulence phenotypes (e.g., LasA protease, LasB elastase, pyocyanin, and rhamnolipid) regulated by QS, reduce the expression of QS regulatory genes (*lasI*/*lasR*, *rhlI*/*rhlR*, *pqsR*/*pqsA*) and enhance the antibacterial effects of amikacin, tobramycin, and ceftazidime. - Baicalin reduces the expression of the Type III secretion system (T3SS) in *Pseudomonas aeruginosa* by a process depending on PqsR. - In a fruit fly infection model and a rat pulmonary infection model of *Pseudomonas aeruginosa*, baicalin reduced inflammatory responses and enhanced bacterial clearance. - In avian pathogenic *Escherichia coli* (APEC), baicalin inhibited AI-2 secretion, biofilm formation, and the expression of virulence genes such as *lsrB*, *lsrK*, *luxS*, *pfs*, *csgA*, *csgB*, and *rpoS*. - 3,5,7-Trihydroxyflavone interferes with the LasR TCS and prevents biofilm formation of *Pseudomonas aeruginosa*. - Baicalin acts in synergy with linezolid against MRSA biofilms.	[531,826,827,828,829,830]
**Berberine**	- Berberine inhibits QS and biofilm formation of *Escherichia coli* and downregulates the QS-related genes: *luxS*, *pfS*, *sdiA*, *hflX*, *motA*, and *fliA*. - Berberine prevents *Salmonella* Typhimurium biofilm formation by preventing type I fimbria formation.	[831,832]
**Betulin and** **betulinic acid**	- Betulin and betulinic acid attenuate the production of QS-regulated virulence factors and biofilm formation in *Pseudomonas aeruginosa*. - Betulin and betulinic acid interfere with the initial stages of biofilm development by decreasing the exopolysaccharide production and cell-surface hydrophobicity. - Betulin and betulinic acid act as competitive inhibitors of the QS regulators LasR and RhlR.	[833]
** *meta* ** **-** **Bromothiolactone**	- *meta*-Bromothiolactone inhibits LasR and RhlR via competition with the natural autoinducers for occupancy of the ligand binding sites. - It inhibits both the production of the virulence factor pyocyanin and biofilm formation. - It protects *Caenorhabditis elegans* and human lung epithelial cells from killing by *Pseudomonas aeruginosa*.	[834]
**Cajaninstilbene** **acid analogs**	- Cajaninstilbene acid analogs have potent anti-biofilm activity against *Pseudomonas aeruginosa* through suppression of *lasB* and *pqsA* and their corresponding virulence factors.	[835]
**Cannabigerol (CBG) and** **Cannabidiol (CBD)**	- CBG reduces the QS-regulated bioluminescence and biofilm formation of *Vibrio harveyi* without affecting the planktonic bacterial growth. - CBG reduces the motility of *Vibrio harveyi* in a dose-dependent manner. - CBG increases LuxO expression and activity, with a concomitant 80% downregulation of the LuxR gene in *Vibrio harveyi*. - CBG has antibacterial and anti-biofilm activities against *Streptococcus mutans*, MSSA, and MRSA. - CBD has antibacterial and anti-biofilm activities against MSSA and MRSA.	[803,804,805,836,837,838,839]
**Carvacrol**	- Carvacrol inhibits acyl-homoserine lactone synthesis by inhibiting LasI of *Pseudomonas aeruginosa*. - It reduces the expression of *cviI* encoding the N-acyl-L-homoserine lactone synthase and the production of violacein in *Chromobacterium violaceum*. - Carvacrol inhibits biofilm formation of *Chromobacterium violaceum*, *Pseudomonas aeruginosa*, *Salmonella enterica subsp. typhimurium*, and *Staphylococcus aureus*. - Carvacrol targets SarA and CrtM of MRSA, resulting in reduced biofilm formation and staphyloxanthin synthesis. - Carvacrol synergizes with the efflux pump inhibitor phenylalanine-arginine β-naphthylamide (PAβN) to kill MexAB-OprM-overexpressing *Pseudomonas aeruginosa*.	[819,840,841,842]
**Cinnamic acid**	- Cinnamic acid inhibits the production of QS-dependent virulence factors and biofilm formation in *Pseudomonas aeruginosa*. - In silico analysis suggests that cinnamic acid acts as a competitive inhibitor for the natural ligands to LasR and RhiR. - The cinnamic acid analog 4-methoxybenzalacetone showed stronger anti-QS activities than cinnamic acid and suppressed the production of pyocyanin and rhamnolipids, as well as the swarming motility of *Pseudomonas aeruginosa*.	[843,844]
**Cinnamaldehyde**	- Cinnamaldehyde inhibits the activity of LasB, RhlA, and PqsA and reduces the intracellular level of c-di-GMP in *Pseudomonas aeruginosa*. - It inhibits biofilm formation of *Pseudomonas aeruginosa* and disperses preformed biofilms. - Cinnamaldehyde has a synergistic effect with colistin, but not with carbenicillin, tobramycin, or erythromycin, on *Pseudomonas aeruginosa*. - It prevents biofilm formation and epithelial attachment of uropathogenic *Escherichia coli* (UPEC). - Cinnamaldehyde could detach and kill MRSA and *Staphylococcus epidermidis* in pre-existing biofilms. - Cinnamaldehyde suppresses the Fsr QS pathway and inhibits biofilm formation and EPS production by *Enterococcus faecalis*.	[845,846,847,848,849,850,851]
**Clofoctol**	- Clofoctol is an approved antimicrobial compound that inhibits the PQS quorum sensing system of *Pseudomonas aeruginosa*. - Clofoctol inhibits the expression of PQS-controlled virulence traits, such as pyocyanin production, swarming motility, biofilm formation, and expression of genes involved in siderophore production.	[852]
**Clotrimazole** **and miconazole**	- Clotrimazole and miconazole are antifungal drugs that have been shown to inhibit the PQS quorum sensing system of *Pseudomonas aeruginosa*.	[852]
**Curcumin**	- Curcumin reduces the expression of the LasR and RhlR QS regulators in *Pseudomonas aeruginosa*, inhibits biofilm formation, and acts in synergism with azithromycin, gentamicin, ceftazidime, and ciprofloxacin. - Curcumin inhibits the response regulator BfmR in *Acinetobacter baumannii* resulting in reduced biofilm formation and virulence. - It inhibits pellicle formation and surface motility of *Acinetobacter baumannii*. - Curcumin inhibits Sortase A of *Streptococcus mutans*. - Bisdemethoxycurcumin sensitizes MRSA to gentamicin, and to a lesser extent to methicillin and oxacillin.	[853,854,855,856,857,858,859]
**Domperidone**	- Domperidone has anti-virulence and anti-biofilm activities against *Staphylococcus aureus*. - It reduces the expression levels of *crtM*, *sigB*, *sarA*, *agrA*, *hla*, *fnbA*, and *icaA* genes. - It is a clinically approved drug that acts as a dopamine antagonist in the treatment of nausea and vomiting.	[860]
**Falcarindol**	- Falcarindol is a major constituent of *Notopterygium incisum*. - It inhibits biofilm formation by repressing QS-related genes (*lasB*, *phzH*, *rhlA*, *lasI*, *rhlI*, *pqsA*, and *rhlR*) in *Pseudomonas aeruginosa*.	[861]
**Flavonoids**	- Flavonoids inhibit QS by preventing LasR/RhlR DNA binding in *Pseudomonas aeruginosa*. - The flavonoid catechin interferes with the RhlR TCS and reduces virulence factors such as pyocyanin and elastase, and prevents biofilm formation in *Pseudomonas aeruginosa*. - The flavanones naringenin, eriodictyol, and taxifolin reduce the production of pyocyanin and elastase in *Pseudomonas aeruginosa*. - Naringenin prevents the production of autoinducers driven by LasI and RhlI in *Pseudomonas aeruginosa*. - Naringenin, kaempferol, quercetin, and apigenin suppressed biofilm formation in *Vibrio harveyi* and *Escherichia coli*.	[862,863,864,865]
**4-Fluoro-** **5-hydroxypentane-2,3-dione (F-DPD)**	- 4-Fluoro-5-hydroxypentane-2,3-dione is a fluoro-DPD analog that disrupts autoinducer-2-dependent quorum sensing in *Vibrio harveyi*.	[866]
**Gingerol**	- Gingerol interferes with the LasR, PhzR, and RhlR TCSs resulting in an impact on the production of EPS, biofilm, pyocyanin, and rhamnolipids of *Pseudomonas aeruginosa*. - Gingerol increased the susceptibility of the bacteria to ciprofloxacin.	[867,868]
**Halogenated furanones**	- The quorum-sensing disrupter (5Z)-4-bromo-5-(bromomethylene)-3-butyl-2(5H)-furanone inhibits the swarming motility and biofilm formation of *Escherichia coli* through inhibition of autoinducer-2 signaling. - Furanone C-30 is a brominated derivative of furanone that acts as a quorum sensing inhibitor and together with tobramycin or colistin it reduces *Pseudomonas aeruginosa* biofilm biomass. - Halogenated furanones blocks cell signaling and quorum sensing in *Pseudomonas aeruginosa* biofilms thus affecting the architecture of the biofilm and enhancing the process of bacterial detachment.	[869,870,871,872,873,874]
**Halogenated** **thiophenones**	- Halogenated thiophenones are anti-QS compounds with anti-biofilm activities against *Vibrio* species and *Staphylococcus epidermidis*.	[875,876,877]
**Hamamelitannin**	- Hamamelitannin is a QS inhibitor that increases the susceptibility of biofilm-associated *Staphylococcus aureus* to vancomycin and clindamycin.	[531]
**4-** **Hydroxybenzylidene indolinone**	- 4-Hydroxybenzylidene indolinone downregulates the Agr quorum sensing system in *Staphylococcus aureus*, upregulates the *sceD* peptidoglycan hydrolase, and sensitizes the bacteria to β-lactam antibiotics and vancomycin. - It inhibits c-di-AMP synthase.	[878,879]
**Luteolin**	- Luteolin inhibits biofilm formation and production of virulence factors by *Pseudomonas aeruginosa* through interaction with LasR.	[880]
**Methyl anthranilate**	- Methyl anthranilate attenuates QS-related virulence production and biofilm formation of *Pseudomonas aeruginosa*. - One mechanism could be by disrupting autoinducer PQS biosynthesis.	[881,882]
**2-[(Methylamino)** **methyl]phenol**	- It targets the quorum regulator SarA in *Staphylococcus aureus* and inhibits biofilm and downregulates virulence genes such as *fnbA*, *hla*, and *hld*.	[883]
**MHY1383 and MHY1387**	- MHY1383 and MHY1387 show anti-QS and anti-virulence activity towards *Pseudomonas aeruginosa* at a low concentration of 100 pM. - They reduce protease production. - They show anti-biofilm activity toward *Pseudomonas aeruginosa* at the low concentration of 1–10 pM by reducing intracellular c-di-GMP levels.	[884]
**MomL**	- MomL is an N-acyl homoserine lactonase from *Muricauda olearia* that degrades AHL autoinducers. - MomL attenuated the virulence of *Pseudomonas aeruginosa* in a *Caenorhabditis elegans* infection model. - A recombinant MomL produced in *Bacillus brevis* reduced the secretion of pathogenic factors and dampened the pathogenicity of *Pseudomonas aeruginosa*.	[885,886]
**Mosloflavone**	- Mosloflavone from the herb *Mosla soochouensis Matsuda* inhibits QS-regulated virulence and biofilm formation of *Pseudomonas aeruginosa* by downregulating the expression of *lasI*, *lasR*, *rhlI*, *chiC*, *phzM*, *exoS*, *algD*, *pelA*, and pyocyanin production. - It binds to the QS regulatory proteins LasR and RhlR by competitively inhibiting the binding of the natural autoinducers.	[887]
**Niclosamide**	- Niclosamide is an anthelmintic drug that inhibits the QS in *Pseudomonas aeruginosa* and prevents the production of autoinducers, resulting in the suppression of motility, biofilm formation, and the production of virulence factors such as elastase, pyocyanin, and rhamnolipids.	[888]
**Nitrofurazone and erythromycin estolate**	- The two antibiotics nitrofurazone and erythromycin estolate reduce the expression of PqsE-dependent virulence traits and biofilm formation by *Pseudomonas aeruginosa*.	[889]
**Oritavancin**	- Oritavancin inhibits the ATPase activity of the histidine kinase ArlS and increases the sensitivity of MRSA to oxacillin both in the planktonic and biofilm state.	[399]
**An Oxoquinazolin derivate**	- It inhibits PqsR and reduces the level of pyocyanin, and the autoinducers HHQ and PQS in *Pseudomonas aeruginosa*. - It potentiates the antimicrobial effect of ciprofloxacin.	[890]
**Paeonol**	- Paeonol exhibits anti-biofilm activity and interferes with the AHL-mediated QS system in *Pseudomonas aeruginosa*. - It downregulates the transcription level of the QS-related genes *lasI/R*, *rhlI/R*, *pqs/mvfR*, as well as the virulence factors *lasA*, *lasB*, *rhlA*, *rhlC*, *phzA*, *phzM*, *phzH*, and *phzS*.	[891]
**Palmitoyl-** **DL-carnitine**	- Palmitoyl-DL-carnitine impairs *Pseudomonas aeruginosa* and *Escherichia coli* biofilm formation. - It inhibits the Las QS system and prevents the biofilm-promoting effects of subminimal inhibitory concentrations of aminoglycosides.	[892]
**Parthenolide**	- Parthenolide has anti-biofilm and anti-QS activities against *Pseudomonas aeruginosa*. - It downregulates *lasI/lasR* and *rhlI/rhlR* as well as reduces EPS production.	[893]
**Phenyl lactic acid**	- Phenyl lactic acid attenuates virulence and pathogenicity of *Pseudomonas aeruginosa* and *Staphylococcus aureus*. - It decreases the adherence of the bacteria to biotic and abiotic surfaces. - It increases the sensitivity of *Pseudomonas aeruginosa* to antibiotics.	[894]
**N-(2-Pyrimidyl)** **butanamide**	- N-(2-Pyrimidyl)butanamide inhibits the Las and Rhl TCSs of *Pseudomonas aeruginosa*, reduces biofilm formation, and sensitizes the bacteria to ciprofloxacin, tobramycin, and colistin.	[895]
**Quinazolinone** **analogs**	- Quinazolinone analogs are antagonists of the PqsR QS receptor of *Pseudomonas aeruginosa*. - They attenuate pyocyanin production.	[351,896]
**Quercetin**	- Quercetin interferes with the LasR, PhzR, and RhlR TCSs of *Pseudomonas aeruginosa*. - A synergistic effect was observed between quercetin and amoxicillin against *Staphylococcus epidermidis* and *Staphylococcus aureus*. - Quercetin in combination with tetracycline was bactericidal for *Escherichia coli*.	[862,867,897,898,899,900]
**Resveratrol(3,5,4′-** **trihydroxystilbene)**	- Resveratrol inhibits the expression of the QS-related *lasI* and *rhlI* in *Pseudomonas aeruginosa*. - Resveratrol enhances the antibacterial effects of aminoglycosides on *Pseudomonas aeruginosa* biofilms. - It is proposed that the antioxidant property of resveratrol inhibits the QS system by relieving oxidative stress. - Resveratrol inhibits biofilm formation in MRSA by interfering with genes related to QS and capsular polysaccharides. - It inhibits the hemolytic activity of *Staphylococcus aureus*.	[901,902,903,904]
**Rifampicin**	- Rifampicin was found to target the AmgRS envelope stress responsive TCS that is responsible for the upregulation of the *mexXY* efflux pump. - Rifampicin could potentiate the activity of aminoglycosides against various clinical isolates of *Pseudomonas aeruginosa*.	[905,906]
**Sesamin and** **Sesamolin lignans**	- Sesamin and Sesamolin lignans inhibit the LasR TCS of *Pseudomonas aeruginosa* and prevent biofilm formation by reducing the production of alginate, exopolysaccharides, and rhamnolipids. - The two lignans rescued pre-infected worms of *Caenorhabditis elegans* and reduced the colonization of bacteria inside the intestine.	[818]
**Sinefungin**	- Sinefungin is a nucleoside analog of S-adenosylmethionine that prevents biofilm formation by *Streptococcus epidermidis*. - It reduces the expression of LuxS and inhibits the production of autoinducer-2.	[907]
**Staquorsin**	- Staquorsin is an AgrA inhibitor that impairs QS and virulence in *Staphylococcus aureus*.	[908]
**Tryptanthrin**	- Tryptanthrin has anti-biofilm activity against *Vibrio cholerae* by targeting LuxO. - It has a synergistic effect with ciprofloxacin.	[909]
**Tryptophan-containing** **antibacterial peptides**	- Tryptophan-containing peptides downregulate the gene expression of both the *las* and *rhl* TCSs in *Pseudomonas aeruginosa*. - Inhibition of biofilm formation by these peptides was associated with reduced expression of *pelA*, *algD*, and *pslA* in *Pseudomonas aeruginosa*. - Tryptophan-containing peptides showed a synergistic effect with ceftazidime and piperacillin. This was related to reduced expression of the efflux pump genes *oprM*, *mexX*, and *mexA* in *Pseudomonas aeruginosa*. - These peptides increased the susceptibility of multidrug-resistant *Staphylococcus epidermidis* to erythromycin and β-lactam antibiotics.	[820,910]
**Unsaturated** **fatty acids**	- The mono-unsaturated fatty acids palmitoleic and myristoleic acids decrease biofilm formation and motility of *Acinetobacter baumannii*. These fatty acids decrease the expression of the regulator *abaR* of the AbaIR TCS and consequently reduce the production of the AHL autoinducer. - Linoleic acid induces biofilm dispersion of *Pseudomonas aeruginosa* by stimulating the phosphodiesterase activity thereby reducing the intracellular cyclic diguanylate concentration. - Linoleic acid inhibits biofilm formation and synergizes with gentamicin to kill *Staphylococcus aureus*. - Palmitoleate inhibits the growth of *Staphylococcus aureus*. - cis-2-Decenoic acid is a diffusible signal factor of *Pseudomonas aeruginosa* that at 2.5 nM causes biofilm dispersion. It also causes dispersion of several other bacterial biofilms including *Escherichia coli*, *Klebsiella pneumoniae*, *Proteus mirabilis*, *Streptococcus pyogenes*, and *Staphylococcus aureus*. - The omega fatty acids cis-4,7,10,13,16,19-docosahexaenoic acid (DHA) and cis-5,8,11,14,17-eicosapentaenoic acid (EPA) inhibit *Staphylococcus aureus* biofilm formation and decreases hemolysis. - Caprylic acid prevents and eradicates *Klebsiella pneumoniae* biofilm. - The polyunsaturated arachidonic acid increases the susceptibility of MSSA, MRSA, and *Staphylococcus aureus* small colony variants to aminoglycosides. - Arachidonic acid increases the membrane fluidity of *Staphylococcus aureus*. - Arachidonic acid and docosahexaenoic acid increase the membrane permeability of *Escherichia coli* and make the bacteria less susceptible to colistin.	[911,912,913,914,915,916,917,918,919,920,921]
**Zingerone**	- Zingerone interferes with the LasR, PhzR, PqsR, TraR, and RhlR TCSs of *Pseudomonas aeruginosa* resulting in attenuation of virulence and inhibition of biofilm formation. - Zingerone increases the susceptibility of *Pseudomonas aeruginosa* to ciprofloxacin.	[867,922,923]
**Walkmycin B**	- Walkmycin B, which is produced by *Streptomyces* species, inhibits the histidine kinase activity of WalK in *Staphylococcus aureus* leading to bacterial growth inhibition.	[924]

**Table 5 microorganisms-10-01239-t005:** Examples of biofilm inhibitors.

Compound	Effects on Bacteria	References
**Astilbin**	- Astilbin is a plant-derived flavanone compound that inhibits SrtA and prevents biofilm formation by *Streptococcus mutans*.	[945]
**Trans-Chalcone**	- Trans-chalcone is a natural plant product that inhibits SrtA and prevents biofilm formation by *Streptococcus mutans*.	[944]
**Chitosan**	- The anti-biofilm activity of chitosan has been attributed to its polycationic nature that binds to negatively charged biofilm constituents such as EPS, eDNA, proteins, and lipids. This leads to changes in membrane permeability and biofilm dispersion. - It also affects biological processes through the chelation of essential metal ions. - Chitosan has been used as a carrier for various antimicrobial and anti-biofilm drugs for synergistic action against biofilms. - Conjugation of streptomycin with chitosan efficiently damaged established biofilms of *Pseudomonas aeruginosa* and *Listeria monocytogenes*.	[950,951,952]
**Clemastine**	- Clemastine inhibits biofilm formation of *Staphylococcus aureus* and enhances the antibacterial activity of oxacillin. - Clemastine inhibits the release of eDNA during biofilm formation and decreases hemolytic activity. - Clemastine reduces the transcription of biofilm formation-related genes (*fnbB*, *icaA*, and *icaB*) and virulence genes (*hlg*, *hld*, *lukde*, *lukpvl*, *beta-PSM*, *delta-PSM*, and *cap5A*). - One of its targets is the cyclic di-AMP phosphodiesterase GdpP.	[953]
**Compound 62520** **(5-fluoro-1-((1R,3S)-** **3-(hyroxymethyl)-** **1,3-dihydroisobenzofuran-** **1-yl)pyrimidine-** **2,4-(1H,3H)-dione)**	- Compound 62520 inhibits *ompA* expression and biofilm formation in *Acinetobacter baumannii*.	[122]
**DMNP—A diterpene analog**	- 4-(4,7-Dimethyl-1,2,3,4-tetrahydronaphthalene-1-yl)pentanoic acid (DMNP) suppresses persistence and eradicates biofilms of *Mycobacterium smegmatis* by inhibiting Rel (p)ppGpp synthetases.	[954]
**Epigallocatechin-** **3-gallate (EGCG)**	- EGCG inhibits biofilm formation and virulence of *Escherichia coli*, *Pseudomonas aeruginosa*, *Staphylococcus aureus*, and *Streptococcus mutans*. - EGCG prevents curli production by suppressing the expression of curli-related proteins and the sigma factor RpoS in *Escherichia coli*. - EGCG promotes RpoS degradation by the ATP-dependent protease ClpXP in combination with its adaptor RssB. - EGCG is an antioxidant.	[955,956,957]
**Eugenol/carvacrol**	- The combined treatment of eugenol and carvacrol eradicates pre-established biofilms of MSSA and MRSA.	[958]
**Gallium nitrate**	- Gallium nitrate leads to nutritional iron starvation and sensitizes MRSA biofilms to vancomycin. - Gallium nitrate inhibits biofilm formation of *Pseudomonas aeruginosa* by disrupting bacterial iron homeostasis. Together with sub-MIC concentrations of tetracycline, gallium nitrate inhibits pyoverdine and suppresses bacterial growth.	[959,960,961,962,963]
**5-Episinuleptolide**	- 5-Episinuleptolide inhibits biofilm formation of *Acinetobacter baumannii*, and decreases the expression of genes from the *pgaABCD* operon, which encodes the extracellular polysaccharide poly-β-(1,6)-N-acetylglucosamine (PNAG). - 5-Episinuleptolide increases the sensitivity of *Acinetobacter baumannii* to levofloxacin.	[964]
**Flavonoids**	- Various glycone (myricitrin, hesperidin, and phloridzin) and aglycone flavonoids (myricetin, hesperetin, and phloretin) inhibit *Staphylococcus aureus* biofilm formation, without affecting their growth.	[156]
**5-Hydroxymethylfurfural**	- 5-Hydroxymethylfurfural inhibits biofilm formation of *Acinetobacter baumannii*. - It inhibits EPS production and downregulates the expression of *bfmR*, *csuA/B*, *ompA*, and *katE* virulence genes.	[965]
**Kaempferol**	- Kaempferol inhibits biofilm formation of *Staphylococcus aureus* by inhibiting SrtA and reducing the expression of adhesion-related genes. Kaempferol had no effect on planktonic growth. - Kaempferol in combination with azithromycin suppressed biofilm formation and growth of azithromycin-resistant *Staphylococcus aureus*.	[966,967]
**α-Mangostin**	- α-Mangostin inhibits biofilm formation of *Acinetobacter baumannii*. - It downregulates the expression of *bfmR*, *pgaA*, *pgaC*, *csuA/B*, *ompA*, and *katE* virulence genes.	[968]
**Meloxicam**	- Meloxicam inhibits biofilm formation of *Pseudomonas aeruginosa* and enhances the antimicrobial activity of tetracycline, gentamicin, tobramycin, ciprofloxacin, ceftriaxone, ofloxacin, norfloxacin, and ceftazidime. - Docking studies suggest that meloxicam can bind to the active site of the QS proteins LasR and PqsE.	[969,970]
**ML346**	- ML346 is a compound with a barbituric acid and cinnamaldehyde scaffold that functions as an irreversible inhibitor of *Staphylococcus aureus* SrtA and *Streptococcus pyogenes* SrtA. - It reduces the virulence phenotype of *Staphylococcus aureus*.	[971]
**Myrtenol**	- Myrtenol attenuates MRSA biofilm and virulence by suppressing *sarA* expression. - It inhibits the synthesis of slime, lipase, α-hemolysin, staphyloxanthin, and autolysin by MRSA. - Myrtenol sensitizes MRSA to hydrogen peroxide. - It prevents biofilm formation and suppresses the expression of the biofilm-associated genes *bmfR*, *csuA/B*, *bap*, *ompA*, *pgaA*, *pgaC*, and *katE* in *Acinetobacter baumannii*. - It increases the susceptibility of *Acinetobacter baumannii* to amikacin, ciprofloxacin, gentamicin, and trimethoprim.	[972,973]
**Orientin**	- Orientin is a natural compound found in several plants that inhibits SrtA and prevents the binding of *Staphylococcus aureus* to fibrinogen and diminishes biofilm formation and virulence.	[974]
**1,2,4-** **Oxadiazole derivatives**	- 1,2,4-Oxadiazole derivatives belonging to the topsentin family inhibit Sortase A transpeptidase activity resulting in biofilm inhibition of *Staphylococcus aureus* and *Pseudomonas aeruginosa*.	[943]
**Pyrimidinedione**	- Pyrimidinedione is a DNA adenine methyltransferase (Dam) inhibitor that inhibits biofilm formation by *Streptococcus pneumoniae*.	[975]
**Pyrogallol**	- Pyrogallol inhibits biofilm formation of *Acinetobacter baumannii*. - It downregulates the expression of *pgaA*, *pgaC*, *csuA/B*, *ompA*, and *katE* virulence genes.	[976]
**Quercetin and** **tannic acid**	- Quercetin and tannic acid inhibit biofilm formation and hemolytic activity of *Staphylococcus aureus*.	[977,978,979,980]
**Taxifolin**	- Taxifolin is a flavonoid plant compound that inhibits SrtA in MRSA and prevents their virulence, biofilm formation, and binding to lung epithelial tissue.	[949]
**Thiazolidinediones (e.g., ciglitazone, TZD-C8, and thiazolidinedione-8)**	- Ciglitazone, a glucose-lowering drug, reduced the bacterial load of *Streptococcus pneumoniae* in the lungs of mice in an infection model. - TZD-C8 inhibits biofilm formation of *Pseudomonas aeruginosa* by inhibiting the autoinducer synthase LasI and downregulating the genes of the *pqsABCDE* and *arcDABC* operons. - Thiazolidinedione-8 inhibits the biofilm formation of *Candida albicans.* - Thiazolidinediones blocked AI-2 QS in *Vibrio harveyi* by decreasing the DNA-binding ability of LuxR.	[981,982,983,984]
**Ursolic acid** **and Asiatic acid**	- Ursolic acid and asiatic acid are pentacyclic triterpenes that inhibit biofilm formation by *Escherichia coli* and *Pseudomonas aeruginosa* without affecting the QS. - Ursolic acid prevents biofilm formation of MRSA by reducing amino acid metabolism and adhesins expression.	[903,985,986]
**Virstatin (4-** **[N-(1,8-naphthalimide)]-** **n-butyric acid)**	- Virstatin prevents pili biosynthesis in *Acinetobacter baumannii.* - It targets the AnoR regulator in *Acinetobacter nosocomialis* that is involved in the synthesis of the autoinducer N-(3-hydroxy-dodecanoyl)-L-homoserine lactone. - It inhibits the transcriptional regulator ToxT in *Vibrio cholerae*, thereby preventing the expression of cholera toxin encoding by *ctx* genes and the co-expressed pili type IV. - Virstatin protected infant mice from intestinal colonization by *Vibrio cholerae*.	[987,988,989]
**Zerumbone**	- Zerumbone inhibits biofilm formation and disrupts established *Acinetobacter baumannii* biofilms. - It downregulates the expression of *adeA*, *adeB*, *adeC*, and *bap*.	[990]

**Table 6 microorganisms-10-01239-t006:** Examples of efflux pump inhibitors (EPIs) ^1^.

Compound	Effects on Bacteria	References
**Alkylaminoquinolines**	- Alkylaminoquinolines inhibit the AcrAB-TolC efflux pump of *Enterobacter aerogenes*, resulting in increased intracellular levels of norfloxacin, tetracycline, and chloramphenicol. - They reduce the MIC of these antibiotics.	[999]
**2-(2-Aminophenyl)** **indole**	- 2-(2-Aminophenyl) indole inhibits NorA in *Staphylococcus aureus* and potentiates the activity of fluoroquinolones.	[1000]
**6-(Aryl)alkoxypyridine-** **3-boronic acid derivatives**	- They inhibit NorA in *Staphylococcus aureus* and potentiate the activity of fluoroquinolones.	[1001]
**Arylpiperazines such as 1-(1-naphthylmethyl)-piperazine (NMP)**	- Arylpiperazines inhibit the AcrAB-TolC efflux pump by binding to AcrB.	[810]
**Berberine**	- Berberine inhibits the MexXY-OprM efflux pump in *Pseudomonas aeruginosa*. - A combination of a berberine derivative with tobramycin increased the sensitivity of *Pseudomonas aeruginosa* to the antibiotic.	[1002,1003]
**Boeravinone B**	- Boeravinone B inhibits NorA efflux pump in *Staphylococcus aureus*, increases the sensitivity to ciprofloxacin, and prevents biofilm formation and bacterial invasiveness.	[786]
**Boronic acid**	- Boronic acid inhibits NorA in *Staphylococcus aureus* and increases the sensitivity to ciprofloxacin.	[1001]
**Capsaicin**	- Capsaicin inhibits NorA in *Staphylococcus aureus* and increases the sensitivity to ciprofloxacin. - It reduces the invasiveness of *Staphylococcus aureus.*	[1004]
**Carvacrol**	- Carvacrol inhibits NorA in *Staphylococcus aureus* and sensitizes the bacteria to norfloxacin.	[1005]
**Conessine**	- Conessine inhibits the MexAB-OprM efflux pump and likely other efflux pumps too in *Pseudomonas aeruginosa.* - It sensitizes the bacteria to various antibiotics including cefotaxime, erythromycin, levofloxacin, and tetracycline. - Combined treatment of conessine and levofloxacin reduced *Galleria mellonella* larval burden of *Pseudomonas aeruginosa*.	[1006,1007]
**D13-9001(A 4-oxo-** **4H-pyrido [1,2-a]** **pyrimidine derivative)**	- D13-9001 inhibits the AcrAB-TolC of *Escherichia coli* by binding to AcrB. - It inhibits the MexAB-OprM of *Pseudomonas aeruginosa* and sensitizes the bacteria to aztreonam.	[810,1008]
**Doxorubicin**	- Doxorubicin inhibits the AcrAB-TolC efflux pump by binding to AcrB.	[810]
**Endocannabinoids**	- Anandamide inhibits drug efflux in MRSA and MDRSA by altering the membrane properties and dispatching the protein motive force. - Anandamide and its analog arachidonoyl L-serine (AraS) sensitize MRSA and MDRSA to various antibiotics including methicillin, ampicillin, tetracycline, gentamicin, and norfloxacin. - Anandamide and AraS inhibit biofilm formation of MSSA, MRSA, and MDRSA.	[789,1009,1010]
**Eugenol derivatives**	- Eugenol derivatives inhibit NorA in *Staphylococcus aureus* and sensitize the bacteria to norfloxacin.	[1011]
**IITR08027**	- IITR08027 inhibits the AbeM efflux pump of *Acinetobacter baumannii* by inducing membrane depolarization and increases the sensitivity to fluoroquinolones and chlorhexidine.	[995]
**Isoliquiritigenin derivates (e.g., IMRG4)**	- Isoliquiritigenin derivates inhibit NorA in *Staphylococcus aureus* and potentiate the activity of fluoroquinolones.	[1012]
**Ketoconazole**	- Ketoconazole inhibits NorA in *Staphylococcus aureus,* potentiates the activity of fluoroquinolones, and inhibits biofilm formation.	[787]
**MBX2319** **(A pyranopyridine)**	- MBX2319 inhibits the AcrAB-TolC by binding to AcrB. - It increases the susceptibility of *Enterobacteriaceae* to ciprofloxacin, levofloxacin, and piperacillin.	[810,1013]
**Meropenem**	- Meropenem is a carbapenem antibiotic that inhibits the MexXY-OprM efflux pump of *Pseudomonas aeruginosa*. - Meropenem sensitizes *Pseudomonas aeruginosa* to various aminoglycosides including paromomycin.	[1014]
**Metformin**	- Metformin increases the sensitivity of *Staphylococcus aureus* to various antibiotics including levofloxacin, linezolid, ampicillin-sulbactam, vancomycin, and doxycycline.	[1015]
**Minocycline**	- Minocycline inhibits the AcrAB-TolC efflux pump by binding to AcrB.	[810]
**1-(1-Naphthylmethyl)** **piperazine**	- The EPI 1-(1-Naphthylmethyl)piperazine sensitizes *Escherichia coli* and *Klebsiella pneumoniae* to tetracycline.	[153]
**Nilotinib**	- Nilotinib is a tyrosine kinase inhibitor that inhibits NorA efflux channel in *Staphylococcus aureus* and sensitizes the bacteria to ciprofloxacin. - Nilotinib together with ciprofloxacin reduces both biofilm formation and preformed mature biofilms.	[154]
**5-Nitro-2-(3-phenylpropoxy)pyridine**	- 5-Nitro-2-(3-phenylpropoxy)pyridine inhibits NorA in *Staphylococcus aureus* and potentiates the activity of fluoroquinolones.	[1016]
**Phenylalanine arginyl β-naphthylamide (PA** **β** **N)**	- Phenylalanine arginyl β-naphthylamide sensitizes drug-resistant *Acinetobacter baumannii* to imipenem. - Phenylalanine arginyl β-naphthylamide inhibits the efflux activity of AcrAB-TolC by binding to AcrB in Gram-negative bacteria. - Phe-Arg-βNaphtylamide prevents biofilm formation of *Escherichia coli*, *Klebsiella pneumoniae*, *Staphylococcus aureus*, and *Salmonella typhimurium*.	[45,153,155,810,1017]
**Phenothiazines such as chlorpromazine,** **thioridazine and** **amitriptyline**	- Phenothiazines inhibit the AcrAB-TolC efflux pump by binding to AcrB. - Phenothiazines inhibit NorA in *Staphylococcus aureus* and potentiates the activity of fluoroquinolones. - Chlorpromazine affects the potassium flux across the membrane in *Staphylococcus aureus* resulting in reduced proton motive force required for drug efflux. - Thioridazine and chlorpromazine prevent biofilm formation of *Escherichia coli*, *Klebsiella pneumonia*, *Staphylococcus aureus*, *Pseudomonas putida*, *Pseudomonas aeruginosa*, and *Salmonella typhimurium*.	[153,155,774,996,1018,1019,1020]
**4-Phenylbenzylidene** **derivatives**	- 4-Phenylbenzylidene derivatives sensitize MRSA to β-lactams. - Molecular docking studies suggest an interaction with the allosteric site of PBP2a. - 4-Phenylbenzylidene derivatives inhibit AcrAB-TolC efflux pump in *Klebsiella aerogenes.*	[1021]
**Piperazine derivatives**	- Piperazine derivatives inhibit RND efflux pumps such as AcrAB-TolC and AcrEF-TolC and enhance the sensitivity of *Escherichia coli* to levofloxacin.	[1022]
**Piperine analogs**	- Piperine analogs inhibit NorA in *Staphylococcus aureus* and potentiate the activity of fluoroquinolones.	[1023]
**Raloxifene and** **pyrvinium**	- Raloxifene and pyrvinium are approved drugs that were found to inhibit NorA in *Staphylococcus aureus* and sensitize the bacteria to ciprofloxacin. - The combination of these efflux pump inhibitors with ciprofloxacin could reduce the burden of preformed biofilms. - Raloxifene is a selective estrogen receptor modulator that is used to prevent and treat osteoporosis. - Pyrvinium is an anthelmintic drug.	[1024]
**Reserpine**	- Reserpine inhibits NorA and TetK of *Staphylococcus aureus*, thus sensitizing the bacteria to fluoroquinolones. - Reserpine inhibits biofilm formation by *Staphylococcus aureus* and *Pseudomonas aeruginosa*. - Reserpine inhibits virulence factors in *Staphylococcus aureus* supposedly by interacting with the regulatory proteins AgrA, AtlE, Bap, IcaI, SarA, and SasG.	[1025,1026,1027,1028]
**Resveratrol**	- Resveratrol is a phytoalexin found in grapes and other plants. - Resveratrol downregulates the *adeB* efflux pump in *Acinetobacter baumannii* and increases their sensitivity to chlorhexidine. - Resveratrol inhibits the AcrAB-TolC efflux pump of *Escherichia coli*. - Resveratrol inhibits ATP synthase in *Staphylococcus aureus* and *Streptococcus pyogenes*. - Resveratrol sensitizes *Staphylococcus aureus* to the antimicrobial peptide human β-defensin hBD4 and polymyxins.	[1029,1030,1031,1032,1033]
**Thiazolidinedione** **derivatives**	- Thiazolidinedione targets NorA of *Staphylococcus aureus*, thus sensitizing the bacteria to fluoroquinolones. - Thiazolidinedione-8 prevents biofilm formation in a mixed *Streptococcus mutans*-*Candida albicans* culture and *Candida albicans* monoculture. - Thiazolidinedione has anti-QS activity against *Vibrio harveyi* by decreasing the DNA-binding activity of LuxR. - (z)-5-Octylidenethiazolidine-2,4-dione prevents biofilm formation and targets the LasI quorum-sensing signal synthase in *Pseudomonas aeruginosa*. - 13-5-(2,4-dimethoxyphenyl)thiazolidine-2,4-dione shows a high affinity for PhzS, a key enzyme in the biosynthesis of the virulent factor pyocyanin in *Pseudomonas aeruginosa*.	[981,982,984,1034,1035,1036]
**Thymol**	- Thymol inhibits NorA in *Staphylococcus aureus* and sensitizes the bacteria to norfloxacin.	[1005]
**Totarol**	- Totarol is a phenolic diterpene that sensitizes NorA-overexpressing *Staphylococcus aureus* to antibiotics by inhibiting the efflux pump.	[1037]
**Trimethoprim**	- Trimethoprim inhibits the AcrAB-TolC efflux pump and synergizes with ciprofloxacin in *Enterobacteriaceae* and *Pseudomonas aeruginosa.*	[1038]
**Verapamil**	- Verapamil inhibits drug efflux and potentiates the antibacterial effect of bedaquiline against *Mycobacterium tuberculosis*. - Verapamil increases the sensitivity of *Staphylococcus aureus* to various antibiotics including levofloxacin, linezolid, ampicillin-sulbactam, vancomycin, and doxycycline.	[1015,1039]

^1^ In the studies claiming that the compound inhibits NorA, the research was performed on a *Staphylococcus aureus* strain (SA-1199B) that overexpresses NorA, but it could be that these compounds also interfere with other efflux pumps.

**Table 8 microorganisms-10-01239-t008:** Examples of compounds targeting cell division components.

Compound	Effects on Bacteria	References
**A benzofuroquinolinium derivative**	- It inhibits bacterial cell division by interfering with the GTPase activity of FtsZ, thus preventing its polymerization. - It restores the susceptibility of MRSA to β-lactam antibiotics.	[1088]
**Berberine**	- It is an isoquinoline alkaloid from *Berberis* and other plants that inhibits FtsZ. - It sensitized multidrug-resistant *Acinetobacter baumannii* to various antibiotics including tigecycline, sulbactam, meropenem, and ciprofloxacin.	[1089,1090]
**Cinnamaldehyde**	- Cinnamaldehyde is a natural compound of cinnamon oils that inhibits FtsZ. - It sensitizes MRSA to amikacin, gentamicin, and oxacillin with a dependency on the strain studied. - It inhibits biofilm formation.	[1091,1092,1093,1094]
**1-Methylquinolinium iodide derivative**	- 1-Methylquinolinium iodide derivative inhibits the GTPase activity of FtsZ, thus preventing FtsZ polymerization and cell division. - It shows antibacterial activity against MRSA, vancomycin-resistant *Enterococcus*, and NDM-1 *Escherichia coli*. - It sensitizes MRSA to β-lactams.	[1095]
**PC190723**	- PC190723 is a benzamide family analog that inhibits the GTPase activity of FtsZ and sensitizes MRSA to β-lactam antibiotics.	[1083,1096,1097]
**Quinuclidine 1**	- Quinuclidine 1 inhibits bacterial cell division by preventing FtsZ protofilament formation and sensitizes MRSA and vancomycin-resistant *Enterococcus faecium* to β-lactam antibiotics.	[1098]
**TXA707**	- TXA707 is an FtsZ inhibitor that sensitizes MRSA to β-lactams.	[1085]

## Data Availability

Not applicable.

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
