# Peer review of "Targeting the Holy Triangle of Quorum Sensing, Biofilm Formation, and Antibiotic Resistance in Pathogenic Bacteria"

_microorganisms, 2022, doi:10.3390/microorganisms10061239_

Round 1
Author Response
We thank the Reviewer for reading and critically reviewing the manuscript.
It seems like a book chapter rather than a review article. The manuscript is too lengthy to read and it is a mixture of different topics unnecessarly corelated with each other. It should be more specific and concise, maybe half content and only with biofilm or qs or amr.
- We agree with the reviewer that this is a comprehensive review. The three topics of antibiotic resistance, quorum sensing and biofilm formation are highly interconnected, and the aim of the review was to present data showing the intricate connection between these three processes. Understanding the interrelationship between these processes will enable us to target the vicious circuits and thus be able to increase the efficacy of antibiotic therapy. The review was in purpose divided into 4 major sections as described in the Introduction. The first section introduces the readers to various antibiotic resistance mechanisms, while the second section deals with the regulation of antibiotic resistance by the quorum sensing systems. The third section deals with antibiotic resistance due to biofilm formation and the regulation of biofilm development by the quorum sensing system. Thus, these three sections are related to each other and have to be described together. The fourth section describes compounds that target either of the processes and can sensitize the resistant bacteria to antibiotics, or they are sufficient as a single agent to elicit an anti-microbial response.
The author should give a citation to each mechanism or sentence, but it is mixed with Gam-positive and Gram-negative organisms. For example, in table-1; the author cited references 25,28,34,46,47 for antibiotic degrading enzymes but not mentioned which organism and the specific citation to the example given. At every table, this is the same pattern of citations.
- According to the Reviewer's recommendations, we have added examples of the bacterial species to Table 1 and in the text when appropriate. The other tables already state the bacterial species. It is well accepted to add the citations together for each issue in a Table. In the text, each sentence has the specific citation.

Reviewer 2 Report
This review deals with various aspects, such as antibiotic resistance, QS and biofilm with a focus on pathogenic bacteria of the "ESKAPE" group. The first part describes the mechanisms involved in antibiotic resistance. The second part describes quorum sensing. The third part discusses various factors that influence biofilm formation and the impact of biofilms on antibiotic resistance. In the last section, some strategies developed to inhibit communication between quorum sensing, biofilm and antibiotic resistance are described. This review is complete, very detailed and accurate. It is appropriate for the journal audience. Some tips below:
1) Line 348,368: the description of the tables must be formatted.
2) The figures need to be improved, they are too large and complex.
3) Line 465, 492: use name in italics, check all main text.
Author Response
We thank the Reviewer for reading and critically reviewing the manuscript.
This review deals with various aspects, such as antibiotic resistance, QS and biofilm with a focus on pathogenic bacteria of the "ESKAPE" group. The first part describes the mechanisms involved in antibiotic resistance. The second part describes quorum sensing. The third part discusses various factors that influence biofilm formation and the impact of biofilms on antibiotic resistance. In the last section, some strategies developed to inhibit communication between quorum sensing, biofilm and antibiotic resistance are described. This review is complete, very detailed and accurate. It is appropriate for the journal audience. Some tips below:
1) Line 348,368: the description of the tables must be formatted.
We have formatted the Tables.
2) The figures need to be improved, they are too large and complex.
The figures were prepared to integrate and better illustrate some of the highly complex and intricate regulatory systems described in the text. In order to make it easier for the reader to understand these figures, we have now added descriptive text to the figure legends.
3) Line 465, 492: use name in italics, check all main text.
We have put the bacteria names in italics.

Round 2
Reviewer 1 Report
I am satisfied with the author's revision and recommend the manuscript for publication.